# Learning the Optimal Stopping for Early Classification within Finite Horizons via Sequential Probability Ratio Test

**Akinori F. Ebihara**     **Taiki Miyagawa**     **Kazuyuki Sakurai**     **Hitoshi Imaoka**
NEC Corporation
aebihara@nec.com

## Abstract

Time-sensitive machine learning benefits from Sequential Probability Ratio Test (SPRT), which provides an optimal stopping time for early classification of time series. However, in *finite horizon* scenarios, where input lengths are finite, determining the optimal stopping rule becomes computationally intensive due to the need for *backward induction*, limiting practical applicability. We thus introduce FIRMBOUND, an SPRT-based framework that efficiently estimates the solution to backward induction from training data, bridging the gap between optimal stopping theory and real-world deployment. It employs *density ratio estimation* and *convex function learning* to provide statistically consistent estimators for sufficient statistic and conditional expectation, both essential for solving backward induction; consequently, FIRMBOUND minimizes Bayes risk to reach optimality. Additionally, we present a faster alternative using Gaussian process regression, which significantly reduces training time while retaining low deployment overhead, albeit with potential compromise in statistical consistency. Experiments across independent and identically distributed (i.i.d.), non-i.i.d., binary, multiclass, synthetic, and real-world datasets show that FIRMBOUND achieves optimalities in the sense of Bayes risk and speed-accuracy tradeoff. Furthermore, it advances the tradeoff boundary toward optimality when possible and reduces decision-time variance, ensuring reliable decision-making. Code is included in the supplementary materials.

## 1 Introduction

Sequential Probability Ratio Test (SPRT) (Wald, 1945) offers a theoretically optimal framework for early classification of time series (ECTS) (Xing et al., 2009). ECTS is a task to sequentially observe an input time series and classify it as early and accurately as possible, balancing speed and accuracy (Gupta et al., 2020; Mori et al., 2016). This is vital in real-world scenarios with high sampling costs or where delays can have severe implications: e.g., medical diagnosis (Evans et al., 2015; Griffin & Moorman, 2001; Vats & Chan, 2016), stock crisis identification (Ghalwash et al., 2014), and autonomous driving (Doná et al., 2019). While the multi-objective nature of ECTS presents challenges, SPRT, with log class-likelihood ratios (LLRs), is optimal for binary i.i.d. samples and asymptotically optimal for multi-class, non-i.i.d. time series. SPRT's optimality ensures decisions within the shortest possible time with a controlled error rate (Tartakovsky, 1998; 1999).

A key limitation of SPRT in real-world applications is the *finite horizon* (Grinold, 1977; Xiong et al., 2022): the deadline for classification. While the original SPRT assumes an indefinite sampling period to reach its decision threshold (Tartakovsky et al., 2014), practical scenarios often demand earlier decisions. For instance, detecting a face spoofing attack at a biometric checkpoint requires classification before the subject passes through (Labati et al., 2016). This constraint frequently results in suboptimal performance, as early thresholds may cause either delayed or rushed decisions (Fig. 1a).

Fortunately, the optimal decision boundary for the finite horizon can be derived by solving *backward induction*, a recursive formula progressing from the horizon to the start of the time series (Chow et al., 1991; Peskir & Shiryaev, 2006). It optimizes the boundary by minimizing Bayes risk, or average *a*

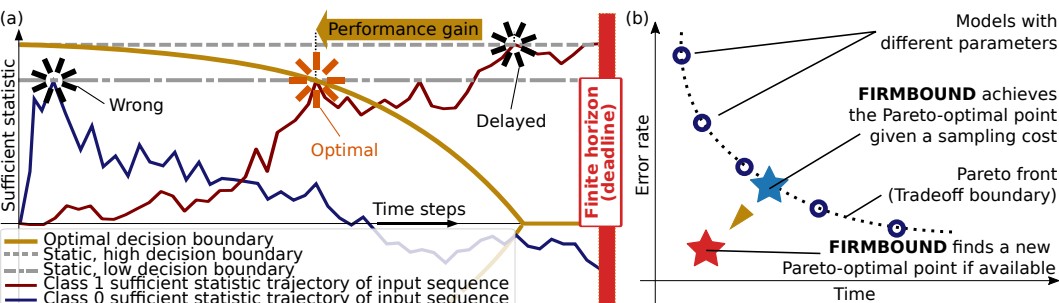

Figure 1: **Visual guide to the optimal stopping under finite horizon.** (a) **Finite horizon SPRT.** Prematurely set decision boundaries lead to suboptimal results. Starbursts mark the stopping times of three decision boundaries for class 1: (Right) a static boundary (upper gray line) leads to delayed decision making; (Center) an optimal decision boundary within a finite horizon (yellow curve) achieves a faster stopping time; and (Left) a lower static boundary (lower gray line) can achieve the same hitting time (center starburst) but increases the risk of classifying another sequence (blue trajectory) to a wrong class. (b) **FIRMBOUND & Pareto front.** FIRMBOUND's goal is to delineate the *Pareto-optimal* point (meaning "optimal in the speed-accuracy multi-objective optimization problem") on the speed-accuracy tradeoff (SAT) curve. It achieves the Pareto-optimal point within the existing front (blue star) or discovers a new Pareto-optimal point (red star) if possible.

*posteriori risk* (AAPR), which accounts for both classification accuracy and sampling costs. The resulting optimal boundary typically tapers monotonically as it nears the finite horizon (Fig. 1a).

Unfortunately, applying the backward induction under real-world conditions is impractical due to its prohibitively high computational costs and the lack of true LLRs (Tartakovsky et al., 2014). To solve the backward induction, numerically calculating the *conditional expectation* of future risks (Eq. 5) is required because no analytical solution has been identified. This calculation necessitates intensive computational resources when applied to large-scale, high-dimensional real-world datasets. For instance, a naive use of sampling-based methods, such as Monte Carlo integration, is ineffective because ECTS demands instantaneous evaluation of the conditional expectation on the fly (Wang & Scott, 2019). Moreover, the lack of true LLRs, which are the *sufficient statistic* required for the backward induction and SPRT, further complicates their practical application within finite horizons.

Thus, we propose the FInite-horizon average a posteriori Risk Minimizer for optimal BOUNDary (FIRMBOUND), a framework designed to estimate the solution to (i) the backward induction and (ii) the sufficient statistic, with theoretical guarantees. For estimating the backward induction, we offer two approaches: first, recognizing the concave nature of the conditional expectation, we formulate its estimation as *convex function learning* (CFL) to provide a statistically consistent estimator. Second, due to the high training costs of CFL (Siahkamari et al., 2022), we explore a faster alternative using Gaussian process (GP) regression (Hensman et al., 2013). Although it can compromise on statistical consistency, GP regression is trained 30 times faster than CFL with comparable performance. Both CFL and GP regression models offer low deployment overhead during the test phase, making them suitable for real-time ECTS. To address the absence of the sufficient statistic (i.e., true LLRs), FIRMBOUND integrates a sequential density ratio estimation (DRE) algorithm (Ebihara et al., 2021) to handle both i.i.d. and non-i.i.d. time series of any class size, producing statistically consistent LLR estimates, on which the optimal decision is learned (Fig. 2b,c).

Our extensive experiments demonstrate that FIRMBOUND effectively approaches Bayes optimality (i.e., minimizes the AAPR) and delineates the *Pareto-optimal* points (meaning "optimal in the speed-accuracy multi-objective optimization problem") of the speed-accuracy tradeoff (SAT). In contrast to most existing ECTS methods, which lack theoretical guarantees (Gupta et al., 2020), FIRMBOUND significantly outperforms these baselines with less parameter sensitivity, substantiating the theoretical predictions across a wide range of synthetic and real-world datasets: two- and three-class sequential i.i.d. Gaussian datasets, non-i.i.d. damped oscillating LLR (DOL) datasets, and real-world datasets such as Spoofing in the Wild (SiW) (Liu et al., 2018b), the human motion database HMDB51 (Kuehne et al., 2011), the action recognition dataset UCF101 (Soomro et al., 2012), and FordA from UCR time series classification archive (Dau et al., 2018). Moreover, FIRMBOUND often achieves a lower

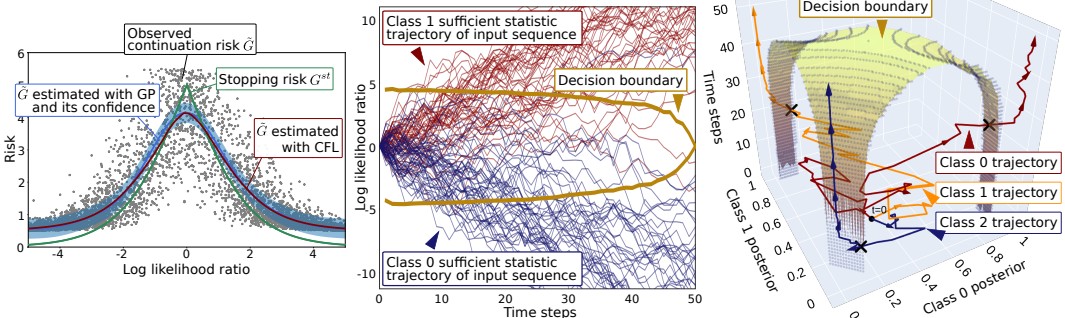

Figure 2: **Learning Decision Boundaries.** (a) **Estimation of the continuation risk function** $\tilde{G}$. Convex Function Learning (CFL) and Gaussian Process (GP) regression on a two-class sequential Gaussian dataset are used. The decision boundary at the current time step ($= 48$) is defined by the intersection of $\tilde{G}$ and the stopping risk function $G^{st}$ (Thm. 2.1) (b, c) **Decision boundaries (thresholds) derived from a two-class (b) and three-class(c) sequential Gaussian dataset.**

error rate than SPRT with static thresholds, illustrating its ability to advance the tradeoff boundary to discover new *Pareto fronts* (see Fig. 1b). Importantly, we empirically show that FIRMBOUND reduces decision-time variance even when it does not advance the Pareto front, contributing to reliable decision making, which is crucial for practical applications. In summary, our contribution is threefold:

1. Statistically consistent and computationally efficient estimation alternatives of the optimal decision boundaries of SPRT for ECTS within finite horizons.

2. Comprehensive data handling under real-world scenarios, being capable of processing both i.i.d. and non-i.i.d. data series, and both binary to multiple, large class datasets.

3. Pareto-optimal decision making with an ability to identify potential new Pareto fronts and reduce variance of decision making time.

A comprehensive literature review can be found in App. E.

## 2 PRELIMINARIES: NOTATIONS AND SPRT

We provide informal definitions here due to page limitations. Detailed mathematical foundations are provided in App. A and Tartakovsky et al. (2014). Let $X^{(1,t)} := \{x^{(t')}\}_{t'=1}^{t}$ and $y \in [K] := \{1, \ldots K\}$ be random variables that represent an input sequence with length $t \in [T]$ and its class label, respectively, where $x^{(t')} \in \mathbb{R}^{d_{\text{feat}}}$ is a feature vector, and $T \in \mathbb{N}$ is the fixed maximum length of sequences, or the *finite horizon*. $X^{(1,t)}$ and $y$ follow the joint density $p(X^{(1,t)}, y)$. Their samples denoted by $X_m^{(1,t)} := \{x_m^{(t')}\}_{t'=1}^{t}$ and $y_m \in [K] := \{1, \ldots K\}$ consist of a dataset, where $m \in [M]$, and $M \in \mathbb{N}$ is the dataset size. The log-likelihood ratio (LLR) contrasting class $k \in [K]$ and $l \in [K]$ is defined as $\lambda_{kl}(T) := \lambda_{kl}(X^{(1,T)}) := \log(p(X^{(1,T)}|y = k)/p(X^{(1,T)}|y = l))$. The posterior of class $k \in [K]$ given $X^{(1,t)}$ is denoted by $\pi_k(X^{(1,t)}) := p(y = k|X^{(1,t)})$. Let $d_t : X^{(1,t)} \mapsto d_t(X^{(1,t)}) \in [K]$ and $\tau : X^{(1,T)} \mapsto \tau(X^{(1,T)}) \in [T]$ denote the *terminal decision rule* (i.e., a class predictor) and *stopping time* (i.e., decision time or hitting time) of the input sequence, respectively. The terminal decision rule may not depend on $t$, in which case we omit the subscript $t$. The stopping time may not require the whole sequence $X^{(1,T)}$ and may be able to calculate from the first $t$ samples $X^{(1,t)}$, depending its algorithm. Our task is to construct the terminal decision rule $\{d_t\}_{t \in [T]}$, which will turn out to be time-independent, and the stopping time $\tau$ that are "optimal" and can be computed efficiently.

**Sequential probability ratio test (SPRT).** Our model is based on SPRT:

**Definition 2.1 (SPRT).** *Given the thresholds $a_k^{(t)} \in \mathbb{R}$ ($k \in [K]$ and $t \in [T]$) for LLRs of input sequences, SPRT is defined as a tuple of a time-independent terminal decision rule and stopping time,*

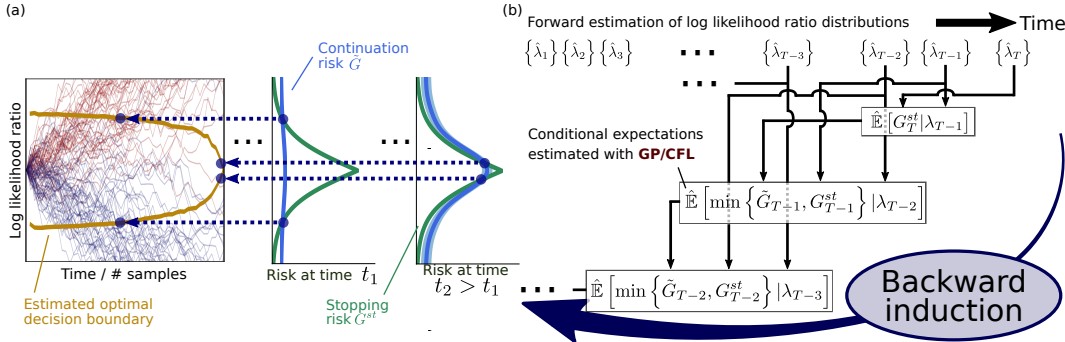

Figure 3: **Conceptual figure of FIRMBOUND.** (a) The intersections of $\tilde{G}$ and $G^{\text{st}}$ delineates the decision boundary. (b) FIRMBOUND estimates conditional expectations using either GP or CFL, based on available sufficient statistic such as (estimated) posterior probabilities $\pi$ or LLRs $\lambda$.

denoted by $\delta^* := (d^*, \tau^*)$, such that

$$d^*(X^{(1,T)}) = d^*(X^{(1,\tau^*)}) \in \arg\max_{k \in [K]}\{\min_{l(\neq k)\in[K]} \lambda_{kl}(X^{(1,t)}) - a_k^{(t)} \mid t = \tau^*(X^{(1,T)})\}, \quad (1)$$

$$\tau^*(X^{(1,T)}) = \tau^*(X^{(1,\tau^*)}) := \min\{t \in [T] \mid \max_{k \in [K]}\{\min_{l(\neq k)\in[K]} \lambda_{kl}(X^{(1,t)}) - a_k^{(t)}\} \geq 0\}. \quad (2)$$

This algebraic definition may seem complex, but the graphical descriptions are given in Figs. 1a & 2b. Note that the terminal decision rule (Eq. 1) is equivalent to choosing the argmax of the gaps between the class posteriors and the thresholds w.r.t. $k \in [K]$ at the stopping time, i.e., choosing the most likely class.

A key feature of SPRT is its various optimalities—asymptotic, non-asymptotic, and Bayes—which theoretically establish SPRT as the best model for ECTS. In this paper, we exploit the Bayes optimality, with the other optimalities summarized in App. B.

To define the Bayes optimality of SPRT, we introduce the *sufficient statistic*, *a posterior risk* (APR), and *average APR* (AAPR). The sufficient statistic for sequential tests here means $\mathscr{S}_t := (\lambda_{kl}(X^{(1,t)}))_{k,l\in[K]}$ (or equivalently, we can use $\mathscr{S}_t := (\pi_k(X^{(1,t)}))_{k\in[K]}$ interchangeably), providing all necessary information for decision at $t$ and serving as the fundamental variable in SPRT's optimality instead of $X^{(1,t)}$ (see also App. C for the formal definition). Then, for a given $d_t$, APR at $t \in [T]$ is defined as

$$\text{APR}_t(\mathscr{S}_t, d_t(X^{(1,t)}) = k) := \bar{L}_k(1 - \pi_k(X^{(1,t)})) + ct, \quad (3)$$

where $c \in \mathbb{R}_{\geq 0}$ is a sampling cost, and $\bar{L}_k$ is the $k$-th element of a penalty vector $\bar{L} \in \mathbb{R}_{\geq 0}^K$ with $k \in [K]$, which penalizes incorrect classifications of $d_t$. The average APR (AAPR), or the Bayes risk, for $\{d_t\}_{t\in[T]}$ and $\tau$ is defined as

$$\text{AAPR}(\{d_t\}_{t\in[T]}, \tau) := \mathbb{E}\left[\text{APR}_\tau(\mathscr{S}_\tau, d_\tau)\right]. \quad (4)$$

SPRT is *Bayes optimal* in the sense that *it can provide a terminal decision rule and stopping time that minimize AAPR if the thresholds for LLRs are properly chosen.* The following theorem provides how to compute the optimal thresholds to achieve the Bayes optimality (Arrow et al., 1949; Tartakovsky et al., 2014).

**Theorem 2.1** (**Backward induction equation**). *Let $\mathscr{S}_t$ be $(\pi_1(X^{(1,t)}), \ldots, \pi_K(X^{(1,t)}))$ w.l.o.g. SPRT $\delta^*$ is Bayes optimal if time-dependent thresholds $a_k^{(t)}$ in Eqs. 1 & 2 are given by the intersections of the continuation risk function $\tilde{G}_t(\mathscr{S}_t)$ and the stopping risk function $G_t^{\text{st}}(\mathscr{S}_t)$, which are defined for a pre-defined density $p$, not for each sample of $X^{(1,T)}$, and satisfy the following backward induction equation:*

$$\tilde{G}_t(\mathscr{S}_t) = \mathbb{E}\left[G_{t+1}^{\min}(\mathscr{S}_{t+1})|\mathscr{S}_t\right] + c \quad (5)$$

$$G_t^{\text{st}}(\mathscr{S}_t) = \min_k\left\{\bar{L}_k(1 - \pi_k(X^{(1,t)}))\right\}, \quad (6)$$

*where $G_t^{\min}(\mathscr{S}_t)$ is referred to as the minimum risk function:*

$$G_t^{\min}(\mathscr{S}_t) := \begin{cases} G^{\mathrm{st}}(\mathscr{S}_t) & (t = T) \\ \min\left\{ G^{\mathrm{st}}(\mathscr{S}_t), \tilde{G}_t(\mathscr{S}_t) \right\} & (1 \le t < T). \end{cases} \tag{7}$$

*Therefore, the optimal stopping region is $\{(\pi_1(X^{(1,t)}), \ldots, \pi_K(X^{(1,t)})) \mid G^{\mathrm{st}}(\mathscr{S}_t) = \tilde{G}_t(\mathscr{S}_t)\}_{t \in [T]} \subset \mathbb{R}^{K \times T}$. A similar theorem holds for $\mathscr{S}_t = (\lambda_{kl}(X^{(1,t)}))_{k,l \in [K]}$, rewriting $\{\pi_k(X^{(1,t)})\}_{k \in [K]}$ by $\{\lambda_{kl}(X^{(1,t)})\}_{k,l \in [K]}$.*

The formal proof is provided in Tartakovsky et al. (2014). For an intuitive explanation, see App. F.

This theorem indicates that the optimal stopping time is given by $\tau^*(X^{(1,T)}) = \tau^*(X^{(1,\tau^*)}) = \min\{t \in [T] \mid G_t^{\mathrm{st}}(\mathscr{S}_t) \le \tilde{G}_t(\mathscr{S}_t)\}$ and that the optimal terminal decision rule simplifies to $d^*(X^{(1,T)}) = d^*(X^{(1,\tau^*)}) \in \arg\min_{k \in [K]}\{\bar{L}_k(1 - \pi_k(X^{(1,t)})) \mid t = \tau^*(X^{(1,T)})\}$. Note that an explicit formula of the dynamic threshold $a_k^{(t)}$ as a function of the sufficient statistic $\mathscr{S}_t$ is unnecessary to compute $d^*$ and $\tau^*$ (Figs. 2a, b, and 3a serve only for visualization). Note also that once the optimal stopping region is determined, calculating $\tau^*$ no longer require a backward computation each time a new sequence arrives because $G_t^{\mathrm{st}}$ and $\tilde{G}_t$ are defined for the underlying density, not for individual sample sequences.

## 3 FIRMBOUND

Unfortunately, solving and deploying Eqs.5–7 in real-world scenarios presents significant challenges. First, these equations lack closed-form solutions. A naive numerical computation, such as Monte Carlo integration, would be possible, but it suffers from the curse of dimensionality ($K$ can be $> 100$, requiring an exponentially large number of samples for convergence) (see also App. D). Second, obtaining a well-calibrated sufficient statistic $\mathscr{S}_t$ is challenging. Although computing softmax logits as class posteriors is common in classification problems (He et al., 2016a;b; Krizhevsky et al., 2012; LeCun et al., 1998), high-dimensional classifiers often produce overconfident or miscalibrated outputs (Guo et al., 2017; Melotti et al., 2022; Müller et al., 2019; Mukhoti et al., 2020).

We address these challenges by transforming the backward induction into a pair of estimation problems and providing statistically consistent estimators (Secs. 3.1 & 3.2). Our proposed model, `FIRMBOUND`, is then proved to be statistically consistent with the Bayes optimal solution (Thm. 3.2).

### 3.1 ESTIMATING THE CONDITIONAL EXPECTATION

The first key idea is to transform the computation of the conditional expectation in the backward induction equation into a regression problem. An important observation is that the conditional expectation function in Eq. 5 is concave (Jarrett & van der Schaar, 2020; Tartakovsky et al., 2014):

**Theorem 3.1.** *$\tilde{G}_t$ and $G_t^{\min}$ are concave functions of vector $(\pi_1(X^{(1,t)}), \ldots, \pi_K(X^{(1,t)}))$ for all $t \in [T]$.*

Equipped with the concavity, we propose to build a consistent estimator of the Eq. 5 though convex function learning (CFL).

**CFL.** CFL aims to build a statistically consistent estimator of a convex function from noisy data points, assuming the target function is inherently convex (Argyriou et al., 2008; Bach, 2010; Bartlett et al., 2005; Boyd & Vandenberghe, 2010; Mendelson, 2004). Assume that we have estimates of the sufficient statistic $\mathscr{S}_t$ for all $t \in [T]$ estimated on a given training dataset $\{(X_m^{(1,T)}, y_m)\}_{m=1}^M$ via the algorithm given in Sec. 3.2. Then, our task toward solving the backward induction equation (Eq. 5–7) is to estimate $G_t^{\mathrm{st}}(\mathscr{S}_t)$ and $\tilde{G}_t(\mathscr{S}_t)$ for all $t \in [T]$. $G_t^{\mathrm{st}}$ can be computed from the estimated sufficient statistic via Eq. 6. Thus, we focus on the continuation risk $\tilde{G}_t(\mathscr{S}_t) = \mathbb{E}\left[G_{t+1}^{\min}(\mathscr{S}_{t+1})|\mathscr{S}_t\right] + c$. To

estimate it from the estimated sufficient statistic, we first rewrite $\tilde{G}_t$ as

$$\tilde{G}_t(\mathscr{S}_t(X_m^{(1,t)})) = \mathbb{E}_{X^{(t+1)}}[G_{t+1}^{\min}(\mathscr{S}_{t+1}(X^{(1,t+1)}))|\mathscr{S}_t(X^{(1,t)} = X_m^{(1,t)})] + c \qquad (8)$$

$$= \int dP(X^{(t+1)}|X^{(1,t)} = X_m^{(1,t)}) G_{t+1}^{\min}(\mathscr{S}_{t+1}(X^{(1,t+1)})) + c \qquad (9)$$

$$= G_{t+1}^{\min}(\mathscr{S}_{t+1}(X_m^{(1,t+1)})) - \epsilon_m^{(t)} + c \quad =: \mathscr{G}_m^{(t+1)} - \epsilon_m^{(t)} + c, \qquad (10)$$

where $\mathscr{G}_m^{(t+1)} := G_{t+1}^{\min}(\mathscr{S}_{t+1}(X_m^{(1,t+1)}))$, $P$ is a properly defined probability measure, and $\epsilon_m^{(t)}$ is a random variable representing the deviation of $\mathscr{G}_m^{(t+1)}$ from the expectation integral in Eq. 9. Suppose that the backward induction equation is solved for $T, T-1, \ldots, t+1$; i.e., $G_{t+1}^{\min}$ is given. Then, $\mathscr{G}_m^{(t+1)}$ is computable from the estimated sufficient statistic by definition. Therefore, we regard $\{\mathscr{G}_m^{(t)}\}_{m,t}$ as a given dataset henceforth, leading to the idea that *the dataset $\{\mathscr{G}_m^{(t)}\}_{m,t}$ can be regarded as a set of noisy observations of the ground truth continuation risk $\tilde{G}_t(\mathscr{S}_t(X_m^{(1,t)}))$ of $X_m^{(1,t)}$* (up to a constant $c$) because $\tilde{G}_t(\mathscr{S}_t(X_m^{(1,t)})) = \mathscr{G}_m^{(t+1)} - \epsilon_m^{(t)} + c$ (Eq.10) $\Leftrightarrow$

$$\mathscr{G}_m^{(t+1)} = \tilde{G}_t(\mathscr{S}_t(X_m^{(1,t)})) + \epsilon_m^{(t)} - c. \qquad (11)$$

This change of view, together with the fact that $\tilde{G}_t(\mathscr{S}_t)$ is concave w.r.t. $\mathscr{S}_t = (\pi_1, \ldots, \pi_K)$, leads to the following noisy convex regression problem:

$$\hat{\tilde{G}}_t(\{X_m^{(1,T)}\}_{m=1}^M) \in \operatorname*{arg\,min}_{f:\text{ concave}}\left\{\frac{1}{M}\sum_{m=1}^M \left(f(\mathscr{S}_t(X_m^{(1,t)})) - \mathscr{G}_m^{(t)}\right)^2 + \lambda\|f\|\right\} + c, \qquad (12)$$

where $\|f\|$ is a regularizer, $\lambda$ is a hyperparameter, and $\hat{\tilde{G}}_t(\{\mathscr{G}_m^{(t)}\}_{m,t})$ denotes the continuation risk estimated on $\{\mathscr{G}_m^{(t)}\}_{m,t}$ or, equivalently, $\{X_m^{(1,T)}\}_{m=1}^M$. With this novel reformulation, we employ an efficient solver, the 2-block Alternating Direction Method of Multipliers (ADMM) algorithm integrated with the augmented Lagrangian method with a concavity constraint (Siahkamari et al., 2022). Specifically, $f$ in Eq. 12 is represented as a piecewise linear function, and $\|f\|$ is defined as the $L^1$ penalty terms (see App. G for the complete algorithm). This algorithm is known to converge to the ground truth function as $M \to \infty$; i.e., it is consistent (Siahkamari et al., 2022).

Consequently, given the estimates of the sufficient statistic, we now have the estimates of the continuation risks $\tilde{G}_t$ and the stopping risks $G_t^{\text{st}}$ as functions of $\mathscr{S}_t$ for all $t \in [T]$. Therefore, in the test phase, we can compute the optimal stopping region given in Thm. 2.1 for any $\mathscr{S}_t$ and $t \in [T]$ without re-solving the backward induction equation.

**Gaussian process (GP) regression.** Although the aforementioned CFL algorithm is theoretically sound and computationally tractable, we further propose a more computationally efficient estimator using GP regression. GP regression is a Bayesian approach to regression used for probabilistic predictions, assuming the objective function values follow a Gaussian distribution defined by a covariance kernel (Wang, 2020).

Evaluating the conditional expectation, or the continuation risk, $\mathbb{E}[G_{t+1}^{\min}(\mathscr{S}_{t+1})|\mathscr{S}_t]$ at any $\mathscr{S}_t$ and $t \in [T]$ is formulated below (we omit inducing points here for brevity). Suppose that a set of estimated sufficient statistics at any $t \in [T]$, denoted by $\{\mathscr{S}_{t,m} := \mathscr{S}_t(X_m^{(1,t)})\}_{m\in[M]}^{t\in[T]}$, is given. We begin with our reformulation discussed above (Eq. 11):

$$\mathscr{G}_m^{(t+1)} + c = \tilde{G}_t(\mathscr{S}_{t,m}) + \epsilon_m^{(t)}. \qquad (11)$$

We make the following fundamental assumptions of GP regression. First, the observation noise $\epsilon_m^{(t)}$ for any $t \in [T]$ and $m \in [M]$ follows a Gaussian distribution. Second, $\{\tilde{G}_t(\mathscr{S}_{t,m})\}_{m\in[M]}$ for any $t \in [T]$ forms a Gaussian process. Under these assumptions, Eq. 11 can be regarded as a GP regression problem with the latent function $\tilde{G}_t$, the explanatory variable $\mathscr{S}_{t,m}$, and the response variable $\mathscr{G}_m^{(t+1)} + c$. Therefore, the predictive distribution of the continuation risk can be calculated, using the standard methods for the evidence lower bound (ELBO) maximization. Specifically, we use the variational GP with an inducing point method (Hensman et al., 2015; Matthews, 2017) via minibatch training to maximize the ELBO. This algorithm uses standard functions from GPyTorch

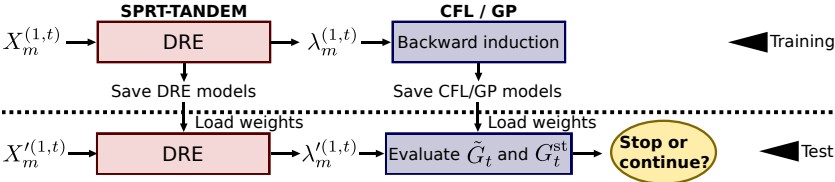

Figure 4: **Training and Testing.** (Top) In the training phase, the sequential DRE algorithm SPRT-TANDEM is trained, followed by the training of CFL or GP models using the backward induction. (Bottom) In the testing phase, the trained DRE model is loaded to sequentially update the LLRs, with which the trained CFL/GP model calculates $\tilde{G}_t$ and compares it with $G_t^{\text{st}}$ to make decisions at time $t$.

(Gardner et al., 2018). For implementation details, see our code. For further detailed mathematical foundations of GP regression, see App. H.

Together with the estimated sufficient statistics, the predictive distribution thus obtained provide $G_t^{\text{st}}$ and $\tilde{G}_t$ for any $\mathscr{S}_t$ and $t \in [T]$. Therefore, in the test phase, we can compute the optimal stopping region given in Thm. 2.1 for any $\mathscr{S}_t$ and $t \in [T]$ without re-solving the backward induction equation. We empirically validate that the training (sometimes referred to as *inference* in the Bayesian context) with GP regression is 30 times faster than the CFL training .

## 3.2 Density Ratio Estimation (DRE) for ECTS

Our remaining task is to estimate the sufficient statistic $\mathscr{S}_t$ for all $t \in [T]$, the second estimation problem mentioned at the beginning of Sec. 3. A simple approach to this end is to estimate LLRs via a sequential density ratio estimation algorithm. It enhances precision by estimating the ratio of probabilities directly, rather than estimating each probability independently, thus reducing degrees of freedom (Belghazi et al., 2018; Gutmann & Hyvärinen, 2012; Hjelm et al., 2019; Liu et al., 2018a; Moustakides & Basioti, 2019; Oord et al., 2018; Sugiyama et al., 2010; 2008; 2012). Specifically, we employ SPRT-TANDEM algorithm (Ebihara et al., 2021; Miyagawa & Ebihara, 2021; Ebihara et al., 2023), which involves a consistent loss function, named *Log-Sum-Exp Loss* (LSEL):

$$\hat{L}_{\text{LSEL}}(\boldsymbol{w}; \{(X_m^{(1,T)}), y_m\}_{m \in [M]}) := \frac{1}{KM} \sum_{k \in [K]} \sum_{t \in T} \frac{1}{M_k} \sum_{i \in I_k} \log(1 + \sum_{l(\neq k) \in [K]} e^{-\hat{\lambda}_{kl}(\boldsymbol{w}, X^{(1,t)})}) \tag{13}$$

where $\boldsymbol{w} \in \mathbb{R}^d$ is the trainable parameters, e.g., the weights of a neural network, $I_k := \{i \in [M] \mid y_i = k\}$ is the index set of class $k$, $M_k := |I_k|$ is the size of $I_k$, and $\hat{\lambda}_{kl}(\boldsymbol{w}, X^{(1,t)})$ is the estimated LLR parameterized by $\boldsymbol{w}$. By minimizing LSEL, the estimated LLRs approaches the true LLRs as $M \to \infty$ (Miyagawa & Ebihara, 2021); i.e, LSEL is consistent. We provide further details of SPRT-TANDEM in App. I.

Finally, integrating CFL and LSEL and solving the backward induction equation, we establish that `FIRMBOUND` is statistically consistent.

**Theorem 3.2** (Informal). *Under several technical assumptions, `FIRMBOUND` with CFL is statistically consistent with the Bayes optimal algorithm for ECTS; i.e., it minimizes AAPR as $M \to \infty$.*

Main assumptions are (i) a sufficiently large dataset size, (ii) a sufficiently large number of iterations of the ADMM algorithm in CFL, and (iii) a sufficiently large neural network for LSEL. The complete set of assumptions, the formal statement, and the proof are provided in App. J, as they are technical and lengthy. In the following, we empirically validate Thm. 3.2, demonstrating that `FIRMBOUND` minimizes AAPR, and highlight its practical strengths of `FIRMBOUND` across various datasets.

## 4 Experiments and Results

These experiments are designed for a *fair* comparison with baseline models without exploring all possible configurations, as such variations would not alter our study's conclusion. To ensure fairness, the same feature extractor and feature vector size $d_{\text{feat}}$ are used across all models. All

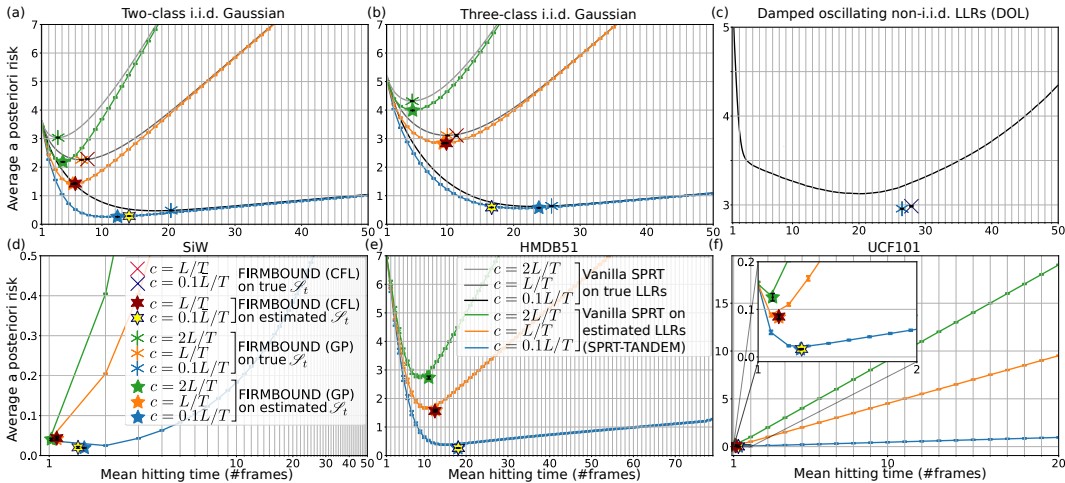

Figure 5: **Averaged a posteriori risk (AAPR) curves.** AAPRs of `FIRMBOUND` are compared with static-threshold SPRTs. Horizontal and vertical axes are mean hitting time and AAPR, respectively. Note that we only show models with well-calibrated sufficient statistic here, as ill-calibrated statistic does not necessarily correlate with ECTS performance by definition and thus not meaningful discussing its minima (but see App. M for AAPR of other baseline models). Error bars represent the standard error of the mean.

hyperparameters, including those for the baseline models, are optimized using Optuna (Akiba et al., 2019) with the Tree-structured Parzen Estimator (Bergstra et al., 2011). Details on parameter selection can be found in App. K. Additional parameter sensitivity test on GP models can be found in App. L, showing robustness against kernel choice. Fig. 4 shows The training and testing pipeline.

**Baselines.** We evaluate the performance of `FIRMBOUND` by comparing it against SPRT with static thresholds and four ECTS models. To conduct SPRT on real-world datasets lacking true LLRs, we utilize SPRT-TANDEM (Ebihara et al., 2021; Miyagawa & Ebihara, 2021; Ebihara et al., 2023) to estimate LLRs. ECTS baseline models include: LSTMms, which enhances monotonic score growth (Ma et al., 2016), the reinforcement learning algorithm EARLIEST (with two fixed hyperparameters lambda=$10^{-1}$ and $10^{-10}$) (Hartvigsen et al., 2019), the convolutional neural network-transformer hybrid, TCN-Transformer (TCNT, with two fixed hyperparameters $\alpha = 0.3$ and $0.5$) (Chen et al., 2022), and Calibrated eArLy tIMe sERies clAsifier (CALIMERA, with fixed hyperparameters, delay penalty= $0.1, 0.5.1.0$) (Bilski & Jastrzębska, 2023).

**Evaluation criteria.** Our evaluation metrics are AAPR and SAT curve. We compute APR at the decision time using softmax probabilities as class posteriors, with a fixed $\bar{L}_k = L = 10$ for all $k \in [K]$, and up to three variations of $c \in \{L/T, 2L/T, 0.1L/T\}$(Fig. 5), where $c = L/T$ is set such that the two terms in APR (Eq. 3) are of comparable magnitude. We do not vary $L$ because decision boundaries are invariant to the scaling of $L$ and $c$ (see App. N for the proof). The SAT curve is derived from the averaged per-class error rate (i.e., macro-averaged recall) measured at the stopping time (Fig. 6).

**CFL.** CFL model is trained at each time step to estimate the conditional expectation (Eq. 7), utilizing a custom training routine adapted from (Siahkamari et al., 2022). We optimize hyperparameters by randomly sampling 1,000 sequential data points. This process is repeated 30 times to identify the hyperparameters that minimize mean squared error using Optuna (Akiba et al., 2019). Once hyperparameter is set, we sample 5,000 sequential data points to model the conditional expectation curve with 5 epochs of training. The requirement of a convex function over a finite input space lets us use posteriors $\pi_k(X_m^{(1,t)})$ as the sufficient statistic $\mathscr{S}_t$. Training on a two-class sequential Gaussian dataset (details provided below) takes approximately 10 hours on NVIDIA RTX 2080Ti.

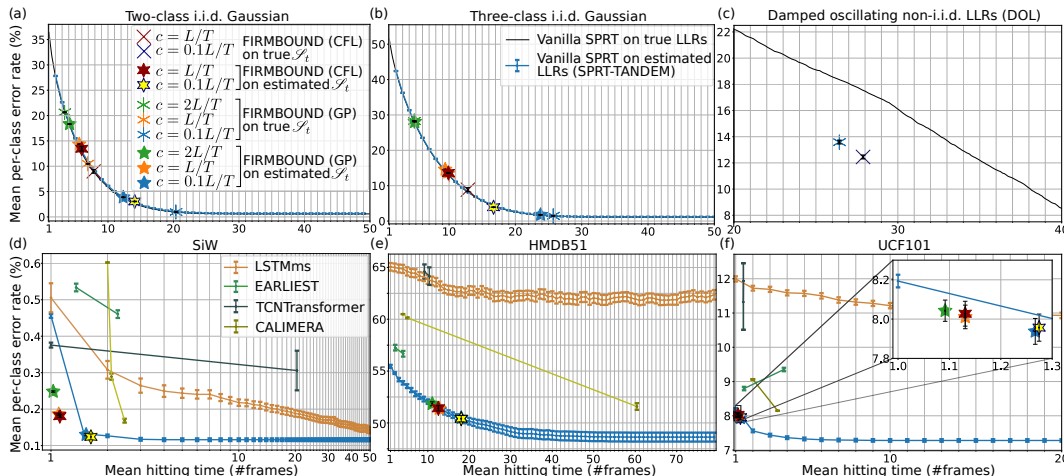

Figure 6: **Speed-Accuracy Tradeoff (SAT) Curves.** The performance of ECTS is evaluated through SAT curves. The horizontal axis represents the mean hitting time, while the vertical axis shows the averaged per-class error rate, equivalent to macro-averaged recall. Thus, models closest to the bottom-left corner perform best. Error bars represent the standard error of the mean.

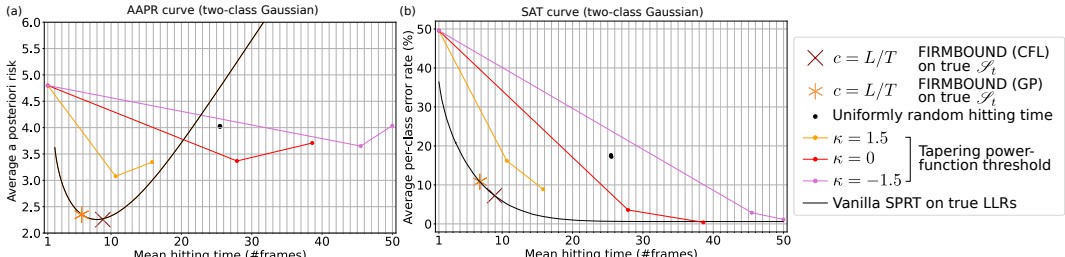

Figure 7: **Additional ablation tests.** Random hitting time and artificial tapering thresholds evaluating (a) AAPR curve and (b) SAT curve. The artificial thresholds start with three different magnitude at $t = 1$, gradually tapering to zero as approaching the horizon $t = T$. See App. O for details.

**GP regression.** GP model is trained at each time step same as CFL. Models are trained for 30 epochs with a batch size of 2,000 with 200 randomly selected inducing points. Our empirical comparisons indicate that either LLRs or posteriors can serve as the sufficient statistic $\mathscr{S}_t$, yielding similar results (App. C). In subsequent analyses, we use LLRs for synthetic data and opt for posteriors for real-world data because of their lower dimensionality. Training on a two-class sequential Gaussian dataset (details provided below) typically requires approximately 20 minutes on NVIDIA RTX 2080Ti.

**Ablation tests.** To assess the impact of different stopping rules, we conducted ablation tests using LLRs. The primary baseline is SPRT with static thresholds on estimated LLRs, as shown in Figs. 5 and 6. Additional tests are random stopping times to establish a chance level and monotonically descending decision boundaries generated using a power function (see App. O for details). Neither variant surpassed FIRMBOUND in terms of AAPR or SAT (Fig. 7). The datasets used in these experiments are detailed below.

**Dataset: sequential i.i.d. Gaussian datasets with known LLRs.** Preliminary assessments are conducted on sequential Gaussian datasets to find that FIRMBOUND can minimize APR to achieve Pareto-optimal both with ground-truth and estimated LLRs. Let $p_0(x)$, $p_1(x)$, and $p_2(x)$ be the 128-dimensional Gaussian densities with an identity covariance matrix. The mean vectors are defined as $(0.5, 0, 0, ..., 0)$, $(0, 0.5, 0, ..., 0)$, and $(0, 0, 0.5, ..., 0)$ for $p_0(x)$, $p_1(x)$, $p_2(x)$, respectively. Only $p_0(x)$ and $p_1(x)$ are used for the two-class dataset. We randomly sampled sequences of length $T = 50$ from these Gaussian distributions to form the datasets. The sizes of the training, validation, and test datasets are as follows: for the two-class dataset, 80,000, 2,000, and 80,000 samples respectively; for

the three-class dataset, 60,000, 6,000, and 120,000 samples respectively. SPRT-TANDEM is trained with the sampled vectors to provide estimated LLRs. Fig. 5a, b and 6a, b shows that `FIRMBOUND` effectively minimize AAPR to reach the best speed-accuracy tradeoff given a sampling cost.

**Dataset: sequential non-i.i.d. Damped-Oscillating LLRs (DOLs).**    We explore the potential for mitigating early inadvertent error to find a new Pareto-front (Fig. 1a). Nonlinear two-class LLRs $\Lambda(t)$ of length $T = 50$ are generated as $\Lambda(t) = \gamma(1 - (1 - t/T)^{\exp(\kappa)}) + A\exp(-\beta t)\sin(\omega t) + \mathcal{N}(0, \sigma)$, where $\gamma \in \{-1, 1\}$ denotes the class label value towards which the first term converges, the second term introduces a damped oscillation, and the third term represents Gaussian noise. The dataset is generated using parameters $\kappa$, $A$, $\beta$, $\omega$, and $\sigma$, chosen from a predefined parameter space. This results in 20,000 training samples, 2,000 validation samples, and 80,000 test samples. For additional details on LLR trajectories and dataset parameters, see App. P. Fig. 6c demonstrate that `FIRMBOUND` achieves notably low errors, effectively advancing the Pareto-front to a new optimal level.

**Dataset: real-world datasets for ECTS.**    Four datasets are used: SiW (two-class) (Liu et al., 2018b), HMDB51 (51-class) (Kuehne et al., 2011), UCF101 (101-class) (Soomro et al., 2012), and FordA dataset (two-class) (Dau et al., 2018). For the SiW dataset, a ResNet-152 (He et al., 2016a;b) is trained as a feature extractor to generate 512-dimensional feature vectors for each frame. The pretrained Microsoft Vision Model ResNet50[1], without fine-tuning, is used to extract 2048-dimensional feature vectors from the HMDB51 datasets. The pretrained vision transformer DINOv2 (the largest model without distillation) with registers (Dosovitskiy et al., 2021; Oquab et al., 2024; Darcet et al., 2024) is used to extract 1,538-dimensional feature vectors from UCF101 datasets. The dataset sizes and sequence lengths are as follows: SiW comprises 46,729 training, 4,968 validation, and 43,878 test samples, all with a sequence length of $T = 50$; HMDB51 includes 5,277 training, 519 validation, and 2,434 test samples with $T = 79$; UCF101 consists of 35,996 training, 4,454 validation, and 15,807 test samples, each with $T = 20$. Figs. 5d–f and 6d–f shows that `FIRMBOUND` reaches Pareto-front by minimizing AAPR, and extend the frontier in a few datasets. FordA is used as an experiment on a non-vision modality dataset, presented in App. Q, showing the same trend.

**Reducing the variance of hitting time.**    Although `FIRMBOUND` consistently identifies the Pareto-optimal point, it does not always show performance gains compared with the vanilla SPRT. However, the *variance* of the hitting time is *statistically significantly* smaller across the database (Tab. 1, Wilcoxon signed-rank test, $p = 2.00 \times 10^{-8} \ll 0.001$), demonstrating `FIRMBOUND`'s advantage in reducing the variance of hitting time to enable reliable decision making across data.

Table 1: **Mean variance of hitting time (MVHT)**. `FIRMBOUND` with CFL provides smaller variance of hitting times than vanilla SPRT when evaluated at the same mean hitting time and corresponding macro averaged recall. The variance reduction is statistically significant (see main text).

| Dataset | Gauss2est. | Gauss3est. | DOL | SiW | HMDB | UCF101 | FordA |
|---|---|---|---|---|---|---|---|
| Trial repeats | 5 | 3 | 3 | 5 | 6 | 10 | 2 |
| ↓MVHT, vanilla SPRT with static threshold | 10.47 | 44.02 | 489.89 | 2.87 | 199.31 | 0.55 | 32.15 |
| ↓MVHT, `FIRMBOUND` with CFL | 9.01 | 42.78 | 405.78 | 1.97 | 195.35 | 0.53 | 23.39 |
| **↑Difference in MVHT (positive is better)** | **1.45** | **1.24** | **84.11** | **0.90** | **3.97** | **0.017** | **8.76** |

## 5    CONCLUSION

With two statistically consistent estimators for backward induction and the sufficient statistic, `FIRMBOUND` delineates stable Pareto fronts across diverse datasets. Unlike existing ECTS models, which lack theoretical guarantees and are sensitive to hyperparameters and datasets (Fig.6d–f), `FIRMBOUND` consistently achieves optimal performance with reduced hitting time variance, approaching optimal performance for ECTS in real-world scenarios. For further discussion, see App.F.

---

[1]https://pypi.org/project/microsoftvision/

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

## ACKNOWLEDGEMENTS

We sincerely thank the anonymous reviewers for their constructive comments and the area chair for a thorough, fair, and objective evaluation of our work. The authors also extend their gratitude to Yuka Fujii for reading an early draft of the manuscript and providing invaluable feedback.

## AUTHOR CONTRIBUTIONS

A.F.E. and T.M. conceived the overall research direction, developed the theoretical framework, and wrote the manuscript. A.F.E. formulated the methodology, ran a pilot study to establish feasibility, conducted the experiments, and organized the code for release. K.S. and H.I. supervised the project.

APPENDIX

CONTENTS

# A MATHEMATICAL FOUNDATIONS

In the main text, we introduced concise notations to avoid delving into unnecessarily technical details. Here, we provide more rigorous definitions. See Tartakovsky et al. (2014) for details.

## A.1 PROBABILITY MEASURE AND DATA RANDOMNESS

We consider a standard probability space $(\Omega, \mathcal{F}, P)$, where $\Omega$ is a sample space, $\mathcal{F} \subset \mathcal{P}(\Omega)$ is a $\sigma$-algebra of $\Omega$, where $\mathcal{P}(\Omega)$ denotes the power set of $\Omega$, and $P$ is a probability measure satisfying Kolmogorov's axioms:

- $P(\Omega) = 1$,
- $P(A) \geq 0$ for any $A \in \mathcal{F}$,
- $P\left(\bigcup_{i=1}^{\infty} A_i\right) = \sum_{i=1}^{\infty} P(A_i)$ for any countable collection $\{A_i\}_{i=1}^{\infty} \subset \mathcal{F}$ of pairwise disjoint sets (i.e., $A_i \cap A_j = \emptyset$ for $i \neq j$).

A function $X = X(\omega)$ defined on the space $(\Omega, \mathcal{F})$ (with values in $\mathbb{R}^{d_{feat}}$ ($d_{feat} \in \mathbb{N}$) in our paper) is called random variable if it is $\mathcal{F}$-measurable. The probability that a random variable $X$ takes values in a set $B \subset \mathbb{R}^{d_{feat}}$ is defined as $P(X \in B) := P(X^{-1}(B))$, where $X^{-1}$ is the preimage of $X$.

Let $\{\mathcal{F}_t\}_{t \geq 0}$ be a filtration, which is a non-decreasing sequence of sub-$\sigma$-algebras of $\mathcal{F}$; i.e., $\mathcal{F}_s \subset \mathcal{F}_t \subset \mathcal{F}$ for all $0 \leq s \leq t$. Each element of the filtration can be interpreted as the available information at a given point $t$. The tuple $(\Omega, \mathcal{F}, \{\mathcal{F}_t\}_{t \geq 0}, P)$ is called a filtered probability space.

In our problem setting, $X_m^{(1,T)}$ in the dataset $S = \{X_m^{(1,T)}\}_{m=1}^M$ represents a sequence of observations for the $m$-th sample, which is treated as a stochastic process or as a realization of the stochastic process $X^{(1,T)}$ interchangeably in our paper. $y_m$ is the fixed class label associated with $X_m^{(1,T)}$.

## A.2 DECISION RULE, TERMINAL DECISION, AND STOPPING RULE

The decision rule $\delta$ is defined as the pair $(d_t, \tau)$, where $d_t$ is the terminal decision rule at time $\tau = t$ ($t \in \{1, ...T\}$) and $\tau \in \{1, ..., T\}$ is the stopping time. We provide their definitions below.

The task of hypothesis testing as a time series classification involves identifying which one of the densities $p_1, \ldots p_K$ the sequence $X^{(1,T)}$ is sampled from. Formally, this tests the hypotheses $H_1 : y = 1, \ldots H_K : y = K$.

The decision function or test for a stochastic process $X^{(1,T)}$ is denoted by $d_t(X^{(1,T)}) : \Omega \rightarrow \{1, \ldots, K\}$. For each realization of $X^{(1,T)}$, we identify $d_T$ as a map $d_t : \mathbb{R}^{d_{feat} \times T} \rightarrow \{1, \ldots, K\}$, i.e., $X^{(1,T)}(\omega) \mapsto y$, where $y \in \{1, \ldots, K\}$. For simplicity, we write $d_t$ instead of $d_t(X^{(1,T)})$.

The stopping time $\tau$ of $X^{(1,T)}$ with respect to a filtration $\{\mathcal{F}_t\}_{t \geq 1}$ is defined as $\tau := \tau(X^{(1,T)}) : \Omega \rightarrow \mathbb{R}_{\geq 0}$ such that $\{\omega \in \Omega | \tau(\omega) \leq t\} \in \mathcal{F}_t$.

Accordingly, for a fixed $T \in \mathbb{N}$ and $y \in \{1, \ldots, K\}$, the set $\{d_t = y\}$ represents the time-series data for which the decision function accepts the hypothesis $H_i (i \in \{1, \ldots, K\})$ with a finite stopping time. Specifically, $\{d_t = y\} = \{\omega \in \Omega | d_t(X^{(1,T)})(\omega) = y, \tau(X^{(1,T)})(\omega) < \infty\}$.

## B  SEQUENTIAL PROBABILITY RATIO TEST AND ITS OPTIMALITY

Our work centers around the optimality of Wald's SPRT. Below, we briefly review the optimality statements for both i.i.d. and non-i.i.d., multiclass classification scenarios. Note that the assumption of increasing LLRs is not applicable under the finite horizon setting discussed in the main manuscript.

**SPRT's optimality with i.i.d., binary class data series.**

**Theorem B.1.  I.I.D. Optimality** *Let the time-series data points $x^{(t)}$, $t = 1, 2, ...$ be i.i.d. with density $f_0$ under $H_0$ and with density $f_1$ under $H_1$, where $f_0 \not\equiv f_1$. Let $\alpha_0 > 0$ and $\alpha_1 > 0$ be fixed constants such that $\alpha_0 + \alpha_1 < 1$. If the thresholds $-a_o$ and $a_1$ satisfies $\alpha_0^*(a_0, a_1) = \alpha_0$ and $\alpha_1^*(a_0, a_1) = \alpha_1$, then SPRT $\delta^* = (d^*, \tau^*)$ satisfies*

$$\inf_{\delta=(d,\tau)\in C(\alpha_0,\alpha_1)} \left\{ \mathbb{E}[\tau|H_0] \right\} = \mathbb{E}[\tau^*|H_0] \quad and \quad \inf_{\delta=(d,\tau)\in C(\alpha_0,\alpha_1)} \left\{ \mathbb{E}[\tau|H_1] \right\} = \mathbb{E}[\tau^*|H_1] \quad (14)$$

A similar optimality also holds for continuous-time processes (Irle & Schmitz, 1984). Thus, SPRT terminates at the earliest expected stopping time compared to any other decision rule achieving the same or lower error rates—establishing the optimality of SPRT.

Thm. B.1 demonstrates that, given user-defined thresholds, SPRT achieves the optimal mean hitting time. Additionally, these thresholds determine the error rates (Wald, 1947). Therefore, SPRT can minimize the required number of samples while maintaining desired upper bounds on false positive and false negative rates.

**SPRT's Asymptotic Optimality with Non-I.i.d., Multiclass Data Series.**  Intuitively, Thm. B.2 (Tartakovsky et al., 2014) suggests that if the LLRs $\lambda_{kl}$ increase as samples accumulate, SPRT algorithm achieves asymptotic optimality. In this condition, the moments of the stopping time are minimized up to order $r$ for a specified classification error rate.

**Theorem B.2** (**Asymptotic optimality of SPRT under a multiclass, non-i.i.d. case**). *Assume that a non-negative increasing function $\psi(t)$ ($\psi(t) \xrightarrow{t\to\infty} \infty$) and positive finite constants $I_{kl}$ ($k, l \in [K]$, $k \neq l$) exist, such that for some $r > 0$, $\lambda_{kl}(t)/\psi(t) \xrightarrow[t\to\infty]{P_k\text{-}r\text{-}quickly} I_{kl}$. Then for all $m \in (0, r]$ and $k \in [K]$, $\inf_{\delta} \mathbb{E}_k[\tau]^m \approx \mathbb{E}_k[\tau^*]^m$ as $\max_{k,l} a_{kl} \to \infty$.*

The precise definition of $r$-quick convergence and a more detailed discussion can be found, e.g., in Tartakovsky et al. (2014). Fig. 8 shows a graphical guide to the multiclass-SPRT decision rule.

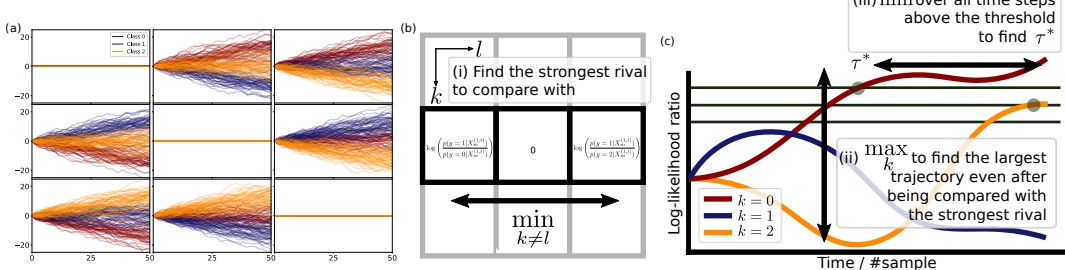

Figure 8: **Procedure of multiclass SPRT with static thresholds.** (a) Example LLR trajectories for a three-class sequential Gaussian dataset, represented as an LLR matrix. (b, c) The two minimum operations defined in Def. 2.1 to determine the stopping time $\tau^*$.

## C   LLRs and Posteriors as Sufficient Statistic

The backward induction equation (Eq.7) depends on a sufficient statistic, which encapsulates all necessary information for decision-making. In hypothesis testing, true LLRs or posterior probabilities suffice to make decisions with a predefined error rate (Wald, 1947), thus both LLRs and posteriors qualify as sufficient statistics. The conversion is expressed by $\pi_k(X^{(1,t)}) = 1/(1 + \sum_{i \neq k} \chi_{ik} \exp(\lambda_{ik}(X^{(1,t)})))$, where $\chi_{kl} := p(y = k)/p(y = l)$ represents the prior ratio. A formal definition of a sufficient statistic is available in Tartakovsky et al. (2014), as follows:

**Definition C.1 (Sufficient Statistic).** *A sequence $\{\mathscr{S}_t\}_{t \geq 1}$ is defined to be sufficient statistic for the sequential decision problem if it satisfies the following conditions:*

1. ***Transitivity****: The sequence is transitive, meaning there exists a function $\phi_n(\cdot)$ such that*

$$\mathscr{S}_{t+1} = \phi_t(\mathscr{S}_t, x^{(t+1)}), \quad \text{almost surely, for } n \geq 1.$$

2. ***Equality of Conditional Probability Density Function (pdf)****: The conditional pdf of $X_{t+1}$ given the past observations $X^t$ can be expressed solely in terms of $\mathscr{S}_t$:*

$$p_{t+1}(x^{(t+1)} \mid X^{(1,t)}) = p_{t+1}(x^{(t+1)} \mid \mathscr{S}_t), \quad \text{almost surely, for } t \geq 1.$$

3. ***Equality of Risks****: The A Posteriori Risk (APR) when using the sufficient statistic $\mathscr{S}_t$ equals the APR calculated directly from the observations:*

$$\text{APR}(X^{(1,t)}) = \text{APR}(\mathscr{S}_t), \quad \text{almost surely, for } n \geq 1.$$

Note that the online DRE algorithm SPRT-TANDEM is transitive, providing consistent estimation of the sufficient statistic.

**Which statistic to use, LLRs or posteriors?**   In principle, the CFL algorithm can handle either LLRs or posteriors as the sufficient statistic for calculating the conditional expectation. Our experiments confirm that both LLRs and posteriors yield equivalent results; however, we opt to use posteriors to reduce input dimensionality.

Conversely, our use of GP regression is predicated on the assumption that the risk distribution is jointly Gaussian, which motivates us to use LLRs as the sufficient statistic. Nonetheless, an experiment with the two-class Gaussian dataset confirms that GP regression provides equivalent results regardless of the type of statistic used (Fig. 9).

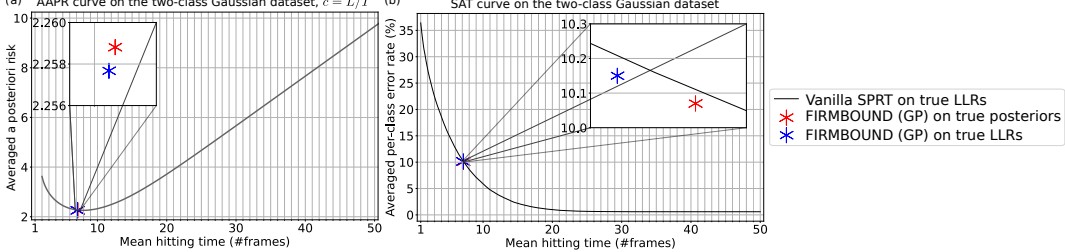

Figure 9: **Comparing LLRs and posteriors as sufficient statistics for GP Regression.** LLRs and posteriors are used as sufficient statistics to evaluate (a) the AAPR curve and (b) the SAT curve. The two-class Gaussian dataset provides the ground-truth LLRs and the corresponding converted posteriors.

## D    COMPUTATIONAL COMPLEXITY OF FIRMBOUND AND SAMPLING METHOD

Here, we provide a detailed comparison of the computational complexity for the inference stage of both the direct estimation approach and the Monte Carlo Integration with Kernel Density Estimation (KDE) approach.

### D.1    DIRECT ESTIMATION APPROACH (FIRMBOUND)

The direct estimation approach uses the following function to evaluate the conditional expectation:

```
@torch.no_grad()
def predict(self, X: Tensor, *args, **kwargs) -> Tensor:
    pred, _ = torch.max(
        torch.matmul(X, self.a.T) + self.y_hat.reshape(1, -1), dim=1
    )
    return pred
```

In this function:

- $X$ is the input tensor of size $[B, K]$, where $B$ is the batch size and $K$ is the number of classes.
- `self.a` and `self.y_hat` are parameter tensors of size $[I, K]$ and $[I]$, respectively, where $I \ll M$ is the subset data number.

The computational complexity for each step in the inference stage is as follows:

1. **Matrix Multiplication**: The operation `torch.matmul(X, self.a.T)` has a complexity of $\mathcal{O}(B \cdot K \cdot I)$. **Broadcasting and Addition**: The operation `torch.matmul(X, self.a.T) + self.y_hat.reshape(1, -1)` involves broadcasting and addition, which has a complexity of $\mathcal{O}(B \cdot I)$.

2. **Maximum Value Selection**: The operation `torch.max(..., dim=1)` finds the maximum value along the specified dimension, which has a complexity of $\mathcal{O}(B \cdot I)$.

Thus, the total computational complexity for the inference stage of the direct estimation approach is dominated by the matrix multiplication step, resulting in:

$$\mathcal{O}(B \cdot K \cdot I)$$

### D.2    MONTE CARLO INTEGRATION WITH KDE APPROACH

The Monte Carlo Integration with KDE approach involves the following steps for the inference stage:

1. Generate $S$ samples from the conditional density $p(\mathscr{S}_{t+1} \mid \mathscr{S}_t)$ using KDE.
2. Evaluate the function $G_{t+1}(\mathscr{S}_{t+1})$ for each sample.
3. Compute the average to estimate the conditional expectation.

Assuming:

- $B$ is the batch size (number of input samples in $\mathscr{S}_t$).
- $K$ is the dimensionality (number of classes).
- $M$ is the total number of data points.
- $S$ is the number of Monte Carlo samples.

In the Monte Carlo Integration with KDE approach, the dimensionality $K$ affects the number of Monte Carlo samples $S$ required for convergence. Let $S(K)$ denote the number of samples as a function of $K$, typically increasing with $K$. The computational complexity for each step is as follows:

1. **Sampling from KDE**: Generating $S(K)$ samples for each of the $B$ input samples, each requiring $\mathcal{O}(M \cdot K)$ operations, resulting in a complexity of $\mathcal{O}(B \cdot S(K) \cdot M \cdot K)$.

2. **Function Evaluation**: Evaluating $G_{t+1}(\mathscr{S}_{t+1})$ for each sample with complexity $\mathcal{O}(K)$, resulting in a total complexity of $\mathcal{O}(B \cdot S(K) \cdot K)$.

3. **Monte Carlo Integration**: The averaging step has a complexity of $\mathcal{O}(B \cdot S(K) \cdot K)$.

Thus, the total computational complexity for the inference stage of the Monte Carlo Integration with KDE approach is dominated by the sampling step, resulting in:

$$\mathcal{O}(B \cdot S(K) \cdot M \cdot K)$$

### D.3 COMPARISON

The direct estimation approach has a computational complexity of $\mathcal{O}(B \cdot K \cdot I)$ for the inference stage, while the Monte Carlo Integration with KDE approach has a complexity of $\mathcal{O}(B \cdot S(K) \cdot M \cdot K)$. Given that $I \ll M$ and considering that higher dimensionality ($K$) increases the number of samples required for convergence ($S(K)$), the direct estimation approach is significantly more efficient in terms of computational complexity during inference. This efficiency is particularly advantageous for real-time applications and large-scale datasets.

Table 2: Comparison of Inference Stage Computational Complexity

| Approach | Inference Stage Complexity |
|---|---|
| Direct Estimation with `FIRMBOUND` | $\mathcal{O}(B \cdot K \cdot I)$ |
| Monte Carlo Integration with KDE | $\mathcal{O}(B \cdot S(K) \cdot M \cdot K)$ |

# E SUPPLEMENTARY RELATED WORK

## E.1 SPRT AND ITS OPTIMALITY

SPRT is Bayes optimal in binary classification with i.i.d. samples and is also known to require the minimal sample size to achieve a predefined error rate (Wald & Wolfowitz, 1948; 1950). The properties of SPRT under multiclass scenarios (Armitage, 1950; Baum & Veeravalli, 1994; Chernoff, 1959; Dragalin, 1987; Dragalin & Novikov, 1999; Kiefer & Sacks, 1963; Lorden, 1977; Paulson, 1963; Pavlov, 1991; 1984; Simons, 1967; Sobel & Wald, 1949), and with non-i.i.d. samples (Dragalin et al., 1999; Lai, 1981; Tartakovsky, 1998), have also been studied (App. B). Several algorithms employ SPRT with estimated density ratio with kernel method (Teng & Ertin, 2016) or boosting (Sochman & Matas, 2005) approach. However, they often assume i.i.d. samples and limited to binary classification, without considering the finite horizon.

## E.2 OPTIMAL STOPPING THEORY

Optimal stopping theory helps decide the best time to act, minimizing expected cost. It applies to various settings like the secretary problem, parking problem, one-armed bandit, change-point detection, and sequential statistical decision problems.

Among these, finite-horizon problems are particularly relevant to our study, which involve a known upper bound on the length of the sequence. Discrete-time, finite-horizon problems are typically solved using dynamic programming techniques like backward induction, a type of Bellman equations. However, backward induction poses significant computational challenges. It requires storing and computing all possible histories, leading to high computational costs and analytical intractability unless the underlying distribution is known and simple (Ferguson, 2006; Tec et al., 2023). Several approximation methods, such as $k$-step and $k$-time look-ahead rules, have been proposed, but they fall short of optimality unless the problem is monotonic, which is often not the case with real-world data.

## E.3 SPRT'S BACKWARD INDUCTION AND CONDITIONAL EXPECTATION

Applying the backward induction under real-world conditions is impractical (Tartakovsky et al., 2014). No analytical solution has been identified, and although numerical computation on simulated datasets is feasible (Jarrett & van der Schaar, 2020), calculating the *conditional expectation of future risks*—a critical component of backward induction—is computationally intensive. This often necessitates approximations such as assuming conditional independence of temporal evidence (Ahmad & Yu, 2013; Naghshvar & Javidi, 2013), discretizing continuous variables (Frazier & Yu, 2007), or adopting a one-step look-ahead approach (Kleinegesse et al., 2020; Najemnik & Geisler, 2005). Moreover, ECTS demands instantaneous evaluation of the conditional expectation, precluding the use of sampling-based methods of the conditional expectation on the fly (Wang & Scott, 2019). The lack of true LLRs, which are the *sufficient statistic* required for SPRT and backward induction, further complicates their practical application within finite horizons.

## E.4 SEQUENTIAL DESIGN

Sequential design, particularly simulation-based Bayesian sequential design, offers a practical approach to these challenges. This method, grounded in statistical decision theory, approximates the objective function (e.g., minimum risk) using simulated trajectories on finite grid points rather than exhaustive computation of all possible histories (Brockwell & Kadane, 2003; Müller et al., 2007; Kadane & Vlachos, 2002). Notable approaches within this framework include constrained backward induction and sequential design with optimizing decision boundaries. Constrained backward induction iteratively approximates expected utility using simulated trajectories, while sequential design with optimizing decision boundaries transforms the sequential decision problem into a non-sequential optimization of parametric decision boundaries (Rossell et al., 2007). Both methods rely on simulated trajectories, unlike our model, which utilizes real-world data trajectories and avoids the tradeoff between precision and computational cost associated with grid-based methods.

### E.5 Reinforcement Learning (RL)

RL is another domain where backward induction, often referred to as the Bellman equation, is extensively applied. In RL, algorithms like Q-learning and policy gradient methods can be viewed as constrained backward induction and sequential decision-making with optimizing boundaries, respectively. However, RL faces significant challenges, including poor sample efficiency and training instability, often leading to catastrophic forgetting and high variance in policy gradient estimates (Atkinson et al., 2018; Bjorck et al., 2021; Cetin et al., 2022; Kumar et al., 2020; Nikishin et al., 2018; Sullivan et al., 2022). RL approach is often combined with the sequential design (Asano, 2022; Blau et al., 2022).

### E.6 Active Learning

Active learning is a machine learning paradigm aimed at achieving high accuracy with minimal labeled data by strategically querying the most informative samples. It encompasses several strategies, including active sensing and active hypothesis testing. In active sensing, the system optimizes sensor placements and parameters to gather the most relevant data, while in active hypothesis testing, the goal is to identify the correct hypothesis as efficiently as possible. One foundational work by (Cohn et al., 1996) demonstrated the effectiveness of active learners over passive learners by querying the most informative data points. (Lewis & Gale, 1994) introduced uncertainty sampling, where instances with the highest uncertainty are selected for labeling. Another key method, query-by-committee (QBC) by (Seung et al., 1992), selects instances based on the disagreement among multiple models. More recently, approaches like Bayesian active learning by disagreement (Houlsby et al., 2011) and core-set approaches (Sener & Savarese, 2017) have been developed to handle the complexity of neural networks. Jarrett & van der Schaar (2020) developed a framework for timely decision-making under context-dependent time pressure.

### E.7 Convex Function Learning (CFL)

CFL aims to infer a convex function from data points, assuming the target function is inherently convex. This assumption ensures that any local minimum is also a global minimum (Argyriou et al., 2008; Bach, 2010; Bartlett et al., 2005; Boyd & Vandenberghe, 2010), thereby simplifying the optimization landscape and enhancing the efficiency of solving optimization problems (Mendelson, 2004). Within this framework, the Alternating Direction Method of Multipliers (ADMM) has proven to be particularly effective (Amos et al., 2016), allowing for the decomposition of complex optimization tasks into smaller, more manageable subproblems that are solved iteratively (Eckstein, 2012; Gabay & Mercier, 1976; Glowinski & Marroco, 1975). ADMM's capability extends to solving the augmented Lagrangean equation on a piecewise linear function, optimizing each segment effectively (Siahkamari et al., 2020). However, the standard ADMM can be slow to converge. To address this, enhancements such as the 2-block ADMM have been developed to accelerate convergence, thus improving the overall performance of CFL applications (Siahkamari et al., 2022).

### E.8 Gaussian Process (GP) Regression

GP regression is a Bayesian approach that makes probabilistic predictions (Wang, 2020). Unlike traditional regression methods that presuppose a specific form for the regression function, GP regression treats observed function values as jointly Gaussian, with a mean function and a covariance defined by a kernel function. The kernel encapsulates assumptions about the function's smoothness and the nature of correlation between function values at different points in the input space. The inherent flexibility of GP regression, which does not require the explicit specification of the function form, renders GP regression widely applicable in diverse fields including geostatistics—often referred to as Kriging (Huang, 2020; Tao et al., 2022; Richter & Toledano-Ayala, 2015), financial modeling (Gonzalvez et al., 2019; Herfurth, 2020; Petelin et al., 2011), to robotics (Cheng et al., 2022; Jakkala & Akella, 2023; Xu et al., 2022).

Traditional GP model, however, faces significant computational challenges when applied to large datasets due to the $\mathcal{O}(M^3)$ scaling with respect to the number of data points $M$. To making it infeasible for large-scale applications. To mitigate this, inducing point methods have been developed to approximate the full GP, substantially reducing the computational load while largely retaining

the model's expressive power (Candela & Rasmussen, 2005). By summarizing the dataset with a smaller set of $m$ inducing points, the complexity is reduced to $\mathcal{O}(m^2 M)$. Additionally, stochastic variational inference (SVI) optimizes variational parameters using minibatches of data, which significantly decreases the computational demands to $\mathcal{O}(m^3)$ per update, independent of the full dataset size (Hensman et al., 2014). This approach not only makes GP regression scalable but also adapts well to modern computational infrastructures, such as GPUs, enabling the handling of extensive datasets within constrained resource settings (Deisenroth & Ng, 2015; Wilson & Nickisch, 2015).

### E.9 OTHER ECTS ALGORITHMS

ECTS is pivotal in scenarios requiring prompt and accurate classification decisions from incomplete data streams. Applications of ECTS includes, but not limited to, medical diagnosis (Evans et al., 2015; Griffin & Moorman, 2001; Vats & Chan, 2016), stock crisis identification (Ghalwash et al., 2014), autonomous driving (Doná et al., 2019), action recognition (Weng et al., 2020), and e-commerce user profiling (Duan et al., 2024). Delays in classification can have critical consequences, positioning ECTS as a key area of research within time series analysis. This field inherently presents a multi-objective optimization challenge aimed at maximizing classification accuracy while minimizing decision time (Mori et al., 2018; Mori et al., 2015; Xing et al., 2012). Recent advancements have integrated deep learning techniques due to their robust representational capacities (Dennis et al., 2018; Ismail Fawaz et al., 2019; Lv et al., 2023; Sun et al., 2023; Suzuki et al., 2018; Hartvigsen et al., 2021). For example, LSTM-s and LSTM-m, have been developed to impose monotonicity on classification scores and enhance inter-class margins, respectively, thereby accelerating action detection (Ma et al., 2016). The Early and Adaptive Recurrent Label ESTimator (EARLIEST) leverages a combination of reinforcement learning and recurrent neural networks to dynamically decide the timing and classification of data (Hartvigsen et al., 2019). Moreover, the incorporation of transformer technologies, as seen in TCN-Transformer, merges temporal convolution with transformer architecture to prioritize early classification through specialized loss functions (Chen et al., 2022). Several algorithms empirically predict future risk to decide when to halt the sampling (Martinez et al., 2020; Wang et al., 2024; Zafar et al., 2021). For example, Calibrated eArLy tIMe sERies clAsifier (CALIMERA) (Bilski & Jastrzębska, 2023) predicts the minima of risk function where the decision making should be made.

### E.10 NEUROPHYSIOLOGICAL UNDERPINNINGS OF SPRT.

SPRT has been identified as a neural decision-making algorithm within the primate brain's lateral intraparietal cortex (LIP, (Roitman & Shadlen, 2002)). During alternative-choice tasks, LIP neurons gradually accumulate sensory evidence, represented by an increasing firing rate of single neurons (Latimer et al., 2015), the average population activity (Shadlen et al., 2016), or a high-dimensional manifold of neural populations (Okazawa et al., 2021). Since neural activities are proportional to LLRs, the behavior of LIP neurons and primates' decision strategies can be best explained by SPRT Kira et al. (2015). For more information, readers are directed to review articles such as (Doya, 2008; Gallivan et al., 2018; Gold & Shadlen, 2007).

Multiple studies investigate decision-making under time pressure (Churchland et al., 2008; Drugowitsch et al., 2012; Hanks et al., 2014). Some papers report closing boundaries under such conditions(Kira et al., 2024), reminiscent of the optimal decision boundary computed with the backward induction (Fig. 2b), while others identify an *urgency signal*, a linearly increasing offset added to the ramping neural activity (i.e., corresponding to the sufficient statistic in optimal stopping theory), which accelerates decisions as the deadline approaches. Both the closing boundary and the urgency signal have psychophysically equivalent effects, compelling quicker decisions with less confidence as the finite horizon approaches.

# F SUPPLEMENTARY DISCUSSION

## F.1 INTUITIVE UNDERSTANDING OF FIRMBOUND

**APR.** The first term of Eq. 3 imposes a heavy penalty if the terminal decision $d_t = k$ corresponds to a low posterior probability $\pi_k$. The second term accumulates the sampling costs up to the current time step $t$.

**Theorem 2.1 (backward induction).** Theorem 2.1 can be interpreted as a recursive decision-making process that minimizes Bayes risk at each time step. The Sequential Probability Ratio Test (SPRT) at each step must either (i) continue sampling to refine the sufficient statistic, or (ii) make a final classification decision with the current sufficient statistic. Each choice—(i) continuing or (ii) stopping—incurs a form of risk: the continuation risk, $\tilde{G}_t$ (Eq. (5)), and the stopping risk, $G_t^{\text{st}}$ (Eq. (6)). At each time step, the lower of these two values defines the minimum risk, $G_t^{\min}$ (Eq. (7)), up to the classification deadline or finite horizon.

**Initiation of backward induction.** At the finite horizon ($t = T$), the minimum risk $G_T^{\min}$ is always equal to the stopping risk $G^{\text{st}}$, setting the initial condition for the risk distribution at time $T$. Subsequently, $G_t^{\min}$ for earlier time steps is recursively calculated using the risk distribution of the subsequent time step (Fig. 3b).

**Risk comparison during deployment.** Initially, the sufficient statistic is small, leading to a higher $G^{\text{st}}$ than $\tilde{G}$. As more samples are collected, the statistic increases, enhancing decision confidence and reducing $G^{\text{st}}$ below $\tilde{G}$. The decision is made when $G^{\text{st}}$ is less than or equal to $\tilde{G}$, with the decision boundary at $\tau^*$ being the intersection of $\tilde{G}$ and $G^{\text{st}}$ (Fig. 3a).

## F.2 CHALLENGES IN ESTIMATING CONDITIONAL EXPECTATIONS USING MONTE CARLO AND KERNEL DENSITY ESTIMATION

Estimating conditional expectations such as $\mathbb{E}\left[G_{t+1}(\mathscr{S}_{t+1}) \mid \mathscr{S}_t\right]$ is a common problem in various scientific and engineering disciplines. One approach to achieve this estimation is by employing Monte Carlo integration techniques (Kroese et al., 2011; Robert & Casella, 2004) in conjunction with Kernel Density Estimation (KDE, (Scott, 1992; Silverman, 1986)) and its application to conditional density, Kernel Conditional Density Estimation (Rosenblatt, 1969) to approximate $p(\mathscr{S}_{t+1} \mid \mathscr{S}_t)$. While this method is theoretically sound and flexible, it comes with several significant challenges, particularly in high-dimensional settings. This section outlines the primary difficulties associated with this estimation method, including issues related to the curse of dimensionality, computational complexity, bandwidth selection, and sampling efficiency.

### F.2.1 CURSE OF DIMENSIONALITY

**Sparsity of data** In high-dimensional spaces, data points tend to become sparse (Botev et al., 2010). The volume of the space increases exponentially with the number of dimensions, which means that even large datasets may not provide sufficient coverage of the space. This sparsity makes it difficult to accurately estimate the conditional density $p(\mathscr{S}_{t+1} \mid \mathscr{S}_t)$ using KCDE because the kernel functions may have to cover large regions with very few data points, leading to high variance in the density estimates. Indeed, Wang & Scott (2019) defines high-dimensional data for the kernel density method at most 50-dimensional, indicating the difficulty of modeling probability distributions over sufficient statistics $\mathscr{S}$, given that $\mathscr{S}$ is at least $K$-dimensional where $K$ is the class number.

**Bandwidth selection.** Selecting an appropriate bandwidth for the kernel is critical for accurate density estimation (Bashtannyk & Hyndman, 2001; Wang & Wang, 2007). In high-dimensional settings, a single bandwidth parameter is often insufficient, and a multidimensional bandwidth matrix is required. However, selecting and optimizing such a bandwidth matrix is computationally intensive and challenging. If the bandwidth is too large, the estimate will be overly smooth, missing important details. Conversely, if it is too small, the estimate will be too noisy, capturing random fluctuations rather than the true underlying structure.

### F.2.2 COMPUTATIONAL COMPLEXITY

**High computational cost.** Kernel density estimation involves computing distances between data points and evaluating kernel functions. In high dimensions, these computations become increasingly expensive. The number of operations required grows with both the number of data points $M$ and the dimensionality of the space. For KCDE, which requires estimating the joint and marginal densities, the computational cost is even higher. Given that our application is online ECTS, waiting for the estimation to converge at each time step is very impractical. Rather, FIRMBOUND provides a function that is readily be evaluated with convergence guarantee, enabling deployment under real-world scenarios.

### F.2.3 BANDWIDTH SELECTION

**Data-dependent bandwidth.** Adaptive methods, where the bandwidth varies locally depending on the density of data points, can provide better estimates but add another layer of complexity. These methods require careful tuning and can be computationally demanding, especially in high-dimensional spaces.

### F.2.4 SAMPLING EFFICIENCY

**Efficient sampling techniques.** Even with an accurate estimate of the conditional density $p(\mathscr{S}_{t+1} \mid \mathscr{S}_t)$, efficiently sampling from this distribution can be challenging. Techniques such as rejection sampling or Metropolis-Hastings may be necessary, but these can be computationally intensive and may not scale well with dimensionality.

### F.2.5 MITIGATION STRATEGIES

To address these challenges, several strategies can be employed, each with its own assumptions and potential sources of error:

**Dimensionality Reduction:** Techniques such as Principal Component Analysis (PCA) or autoencoders can be used to reduce the dimensionality of $\mathscr{S}$ while preserving important structures in the data. This can help alleviate the curse of dimensionality and improve the efficiency of density estimation. However, this assumes that the reduced dimensions adequately capture the necessary information, which may not always be true.

**Sparse Kernel Methods:** Utilizing a subset of the data points (e.g., random sampling or clustering-based methods) can reduce the computational burden. The assumption here is that the subset is representative of the full dataset, which might not hold in all cases, potentially leading to biased estimates.

**Localized Methods:** Adaptive kernel methods, where the bandwidth varies depending on the local density of data points, can provide more accurate estimates in high-dimensional spaces. These methods assume that local adaptation can adequately capture the density variations, but improper tuning can introduce significant errors.

**Grid-Based Methods:** For moderate-dimensional cases, grid-based methods can approximate the density on a discretized grid, reducing computational complexity. The main assumption is that the grid resolution is fine enough to capture the density details, but this can lead to high memory and computation costs if the dimensionality is still relatively high.

### F.2.6 ADVANTAGE OF HAVING A DIRECT ESTIMATOR FIRMBOUND

The development of a direct estimator for the conditional expectation $\mathbb{E}\left[G_{t+1}(\mathscr{S}_{t+1}) \mid \mathscr{S}_t\right]$ presents significant advantages over the traditional Monte Carlo Integration approach combined with Kernel Density Estimation. Firstly, a direct estimator offers computational efficiency by providing instantaneous evaluations, which is crucial for real-time applications and large-scale datasets. This efficiency eliminates the need for extensive sampling and repeated function evaluations inherent in Monte Carlo methods, thus reducing computational overhead. Additionally, the direct estimator ensures statistical consistency, guaranteeing that as the sample size increases, the estimator converges to the true conditional expectation, thereby enhancing the reliability and accuracy of the estimates. In contrast,

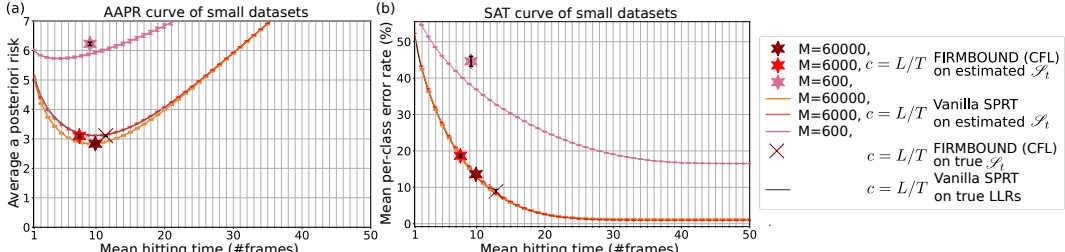

Figure 10: **Performance comparison across datasets of different sizes.** The three-class Gaussian dataset is reduced from the original size ($M = 60000$) to $M = 6000$ and $M = 600$ for training FIRMBOUND, while using the same test dataset. Hyperparameter settings remain fixed, and experiments are repeated five times to compute error bars. (a) The AAPR curve and (b) the SAT curve demonstrate that FIRMBOUND maintains competitive performance with datasets one order of magnitude smaller ($M = 6000$).

Monte Carlo methods are prone to sampling errors and require careful tuning of kernel functions and bandwidth parameters, adding complexity and potential sources of error. Furthermore, the direct estimator simplifies the implementation process by obviating the need for density estimation, which can be particularly challenging in high-dimensional spaces due to the curse of dimensionality. This simplicity, coupled with reduced memory requirements, makes the direct estimator more robust and scalable, offering a clear advantage in handling high-dimensional data and real-time decision-making scenarios.

### F.3 PERFORMANCE UNDER SMALL DATASETS

While FIRMBOUND ensures statistical consistency, its practicality could be questioned if performance degrades significantly with reduced dataset sizes. To address this concern, we conduct an additional experiment. Using the three-class Gaussian dataset, we train FIRMBOUND with datasets up to two orders of magnitude smaller than the original size while keeping the same test dataset. Specifically, dataset sizes of $M = 600$ and $M = 6000$ are compared to the original size of $M = 60000$. Hyperparameter settings are kept fixed, and experiments are repeated five times to compute error bars.

The results show that even with a dataset one order of magnitude smaller ($M = 6000$), FIRMBOUND demonstrates almost negligible differences in performance compared to the original dataset in terms of mean hitting time, AAPR, and mean per-class error rate. Notably, the real-world HMDB51 dataset has a similar order of magnitude ($M = 5277$), further showcasing FIRMBOUND's robustness in real-world scenarios. However, with the smallest dataset ($M = 600$), AAPR and mean per-class error rates increase significantly, indicating that the dataset size is insufficient to accurately estimate the sufficient statistics. Despite this, the mean hitting time remains close to the original, highlighting FIRMBOUND's stability/ and its ability to make reliable best-effort decisions even under highly limited data conditions.

### F.4 BROADER IMPACT

FIRMBOUND enhances the performance of ETCS in real-world settings and prompts further research across both machine learning and neuroscience. It facilitates the backward induction on real-world datasets, effectively removing constraints associated with i.i.d. or non-i.i.d. data, thereby expanding its utility for time-sensitive tasks. Furthermore, backward induction is instrumental in fields like active sensing and sequential design, where unlike ECTS, an agent proactively selects actions to gather informative evidence. FIRMBOUND is ideally suited for such applications, enabling its deployment in dynamic environments. Additionally, the tapering optimal threshold is reminiscent of decision-making processes observed in humans, providing a potential bridge to understanding neural thresholding mechanisms within finite horizon, which, despite extensive study (Churchland et al., 2008; Drugowitsch et al., 2012; Gold & Shadlen, 2007; Latimer et al., 2015; Okazawa et al., 2021), remain elusive.

Our method is designed to optimize the speed-accuracy tradeoff in real-world applications, which is expected to lead to positive societal impacts. The potential for negative effects is minimal and mainly confined to instances where models are intentionally trained to prioritize speed or accuracy by using an extreme value of sampling cost, which could compromise decision speed or quality.

### F.5 LIMITATIONS AND FUTURE WORK

**Domain gap.**    While `FIRMBOUND` provides a theoretical guarantee to minimize the AAPR, it is important to acknowledge that it may not achieve the global minimum on test data when there is a domain gap between the training and test sets. A domain gap occurs when the distribution of the test data differs from that of the training data, which can lead to suboptimal performance of the model, even with the optimal stopping rule. `FIRMBOUND`, as proposed, assumes that the test data follows the same distribution as the training data, and thus, its effectiveness may be compromised in the presence of such domain discrepancies.

We recognize the importance of addressing domain gaps in machine learning research. However, it is important to note that handling domain gaps is beyond the current scope of our paper, which focuses on developing an optimal stopping rule for early classification within finite horizons. Employing a domain adaptation algorithm or foundation models would require a different methodological approach and additional research efforts. Potential directions include incorporating domain adaptation techniques and robustifying `FIRMBOUND` against such discrepancies.

**Future theoretical directions.**    While `FIRMBOUND` is "doubly consistent" estimator of both the backward induction and log-likelihood ratio, several theoretical directions are yet to be investigated. One example is the convergence rate of the algorithm. While it is presumably given by the sum of LSEL's and CFL's. The latter is given in the paper, but the former requires an additional extensive analysis because Lemma J.4 (consistency of LSEL), on which our consistency proof relies, is an asymptotics of the probability that the estimated parameters deviates from the optimal parameter set. Similarly, minimax bound cannot be derived straightforwardly, warrant a separate, focused study.

**Potential density chasm problem.**    We observe that Fig. 5a (for the two-class Gaussian dataset) shows a discrepancy in the minimal averaged posterior risk (AAPR) locations between true and estimated LLRs. Interestingly, this trend is negligible in Fig. 5b (for the three-class Gaussian dataset), where AAPR locations for true and estimated LLRs align more closely. One possible explanation is the density chasm problem, a known issue specific to density ratio estimation on "easy" problems, which increases the absolute value of the density ratio and could contribute to these errors (Rhodes et al., 2020). A countermeasure to the density chasm problem, telescoping density ratio estimation, was proposed in (Rhodes et al., 2020), and employing this approach may help mitigate the error on simple datasets.

These issues are likely specific to simple datasets and are less prevalent in complex real-world datasets with arbitrary class counts. It is important to note that while statistical consistency guarantees minimization of estimation error, it does not address approximation or optimization errors, which may be the main contributing factors here.

**Performance gain under dynamic environments.**    `FIRMBOUND` often shows limited performance on datasets with monotnic trajectory of sufficient statistics (see also App. F.6 for more detailed discussion). This minimal performance gain does not diminish FIRMBOUND's practical value. The real-world datasets examined (SiW, HMDB51, UCF101, and FordA) involve relatively small domain gaps and fewer fluctuations, producing stable, monotonic trajectories that limit opportunities for improvement over static thresholds. However, in more adversarial real-world scenarios, such as those with dim or variable lighting, we would expect the trajectory to fluctuate similarly to the DOL dataset, where FIRMBOUND shows robust performance gains. Furthermore, as noted in Section 4, FIRMBOUND consistently reduces the variance of hitting times across all datasets. Along with its capability to handle i.i.d., non-i.i.d., and multiclass data, this variance reduction demonstrates FIRMBOUND's potential for reliable decision making. In future work, we aim to explore FIRMBOUND's performance under more dynamically adversarial conditions, which we anticipate will reveal greater gains similar to those seen in our DOL experiments.

**Retraining requirement at cost change.** FIRMBOUND effectively delineates the Pareto front on the SAT. A potential limitation arises if a user is unsatisfied with the resulting speed or accuracy and wants to select a different point on the Pareto front; in such cases, retraining FIRMBOUND with a new cost parameter $c$ is required. The training process can be computationally intensive, especially if the CFL algorithm is used to estimate the conditional expectation. However, there is a remedy for this issue. Without incurring additional computational costs, users can re-evaluate the AAPR curve on the current sufficient statistic $\mathscr{S}$ using different values of $c$. This allows them to efficiently identify the optimal $c$ that yields the desired mean hitting time and corresponding error rate on the SAT curve by finding the AAPR curve whose minimum is closest to the desired mean hitting time. This strategy effectively avoids the need for retraining FIRMBOUND when a specific error rate $\alpha$ must be achieved on the SAT curve, which is often crucial in high-security applications.

**Extremely large LLRs' magnitude.** Large LLRs, which can occur when the classification task is relatively easy, can significantly hinder the training of FIRMBOUND in the following ways. When LLRs are extremely large, the corresponding posterior probabilities derived from them often degenerate to either zero or one, making the training data less informative. This forces FIRMBOUND to learn from a dataset with extreme and non-informative posterior probabilities, potentially leading to overfitting and reduced generalization performance. An alternative approach is to train FIRMBOUND directly on the LLRs instead of the posteriors. However, in multiclass classification with $K$ classes, the number of pairwise LLRs required is at least $K(K-1)/2$, which can be extremely large. This approach can be prohibitively memory-intensive, especially when $K$ is large, making training on standard devices challenging. Additionally, the unbounded nature of LLRs can introduce instability into the training process. However, it is important to note that in cases where posterior probabilities degenerate, time-series analysis may not be necessary. The fact that classification can be resolved entirely within the first (or first few) steps suggests that FIRMBOUND may not need to be employed in such scenarios, indicating that this limitation is not a direct weakness of the method.

## F.6 FREQUENTLY ASKED QUESTIONS

**How would you justify FIRMBOUND's two-component framework, given that it introduces additional complexity and potential error propagation?** The doubly consistent estimation —of both the conditional expectation in backward induction and the LLR— required by FIRMBOUND necessitates a multi-component framework. While this design may introduce additional complexity and the potential for error propagation across components, FIRMBOUND guarantees the minimization of estimation errors, particularly in large datasets.

Consistent estimation itself represents a significant advancement in ECTS. Most existing ECTS methods rely on empirical heuristics, whether they follow a two-component approach (e.g., CALIMERA) or a one-component, end-to-end framework (e.g., LSTMms, TCN-Transformer, EARLIEST). Our experiments demonstrate that these heuristic-based approaches are consistently outperformed by FIRMBOUND, emphasizing the practical benefits derived from our theoretically grounded, multi-component framework.

**When is a *new* Pareto-front available?** FIRMBOUND extends the Pareto front on some datasets, depending on LLR monotonicity. In Gaussian datasets with monotonically increasing LLRs, it expedites decisions without increasing error rates (Fig. 1a, 6a, b). Importantly, even without a new Pareto-front, FIRMBOUND ensures reliability by reducing hitting time variance (Tab. 1), crucial for safe deployment in diverse scenarios. Conversely, in non-monotonic DOL datasets with initial noise followed by stabilization, it effectively achieves new Pareto-fronts (Fig. 6c).

**Is it possible to establish consistency with GP? When should I use GP?** Yes, it is possible to construct a consistent estimator using Gaussian Processes (GP) under certain conditions. The GP regressor can be a consistent estimator if, with an appropriate choice of the kernel, $\epsilon_m^{(t)}$ follows a Gaussian distribution for all $t \in [T]$ and $m \in [M]$, and $\{\tilde{G}_t(\mathscr{S}_{t,m})\}_{m=1}^M$ is a Gaussian process for all $t \in [T]$. However, these conditions are difficult to guarantee under arbitrary circumstances. Thus, we introduced CFL as a more general and robust solution.

**When should I use GP? What's your recommendation?**   Albeit the consistency loss, our experiments show that the GP regressor performs competitively with CFL, which is the theoretically optimal approach. This makes GP a practical choice in scenarios with limited computational resources. It is worth noting that GP regression is particularly efficient during the *training* stage. During testing or deployment, both CFL and GP regression are sufficiently fast to support real-time decision-making. Therefore, GP regression is an effective option in environments with constrained training resources, such as edge computing settings.

**Why some models are excluded from Fig. 5?**   Effective risk minimization relies on well-calibrated statistics. As shown in Fig. 14 in App. M shows that model rankings based on AAPR don't always align with their SAT performance due to overconfidence or miscalibration, where predicted confidence levels don't match actual accuracy Guo et al. (2017); Melotti et al. (2022); Müller et al. (2019); Mukhoti et al. (2020). While it is feasible to minimize AAPR on the miscalibrated outputs, it can lead to suboptimal decision-making. By employing well-calibrated Log-Likelihood Ratios (LLRs), FIRMBOUND effectively minimizes risk, achieving Pareto-optimality at a given sampling cost. This calibration ensures that the model's confidence levels are more aligned with the true probabilities, thereby enhancing the reliability of the decision boundaries used in the stopping rule.

**What is the tackled problem?**   Finite horizon Early Classification of Time Series (ECTS) aims to minimize the decision time $\tau < T$ while maintaining a desired error rate $\alpha$, where $T$ is the maximum possible time step. As mentioned in Sec. 2, SPRT optimally solves ECTS under an infinite horizon, detailed in App. B. For finite horizons (including infinite as a special case, l. 108), the average a posteriori risk (AAPR) must be defined and minimized via backward induction to find the optimal stopping boundary (Sec. 3).

**Why does computing the backward induction equation yield minimal AAPR?**   The minimal AAPR is defined as the expected optimal risk at the initial time step, computed recursively from the finite horizon back to the start using the backward equation. For a detailed explanation, see (Tartakovsky et al., 2014).

**Why is FIRMBOUND necessary, given the availability of simulation studies?**   FIRMBOUND accommodates a broad range of time series data, including i.i.d., non-i.i.d., binary, and multiclass series. In contrast, many existing simulation studies focus on artificial datasets with limited classes and do not reflect real-world complexities. Additionally, optimal threshold searching through numerical simulation might require intensive grid sampling, which becomes computationally impractical with large classes—for example, assigning unique posterior probabilities to 101 classes with 0.1 steps could result in up to 47 trillion combinations. FIRMBOUND facilitates the application of backward induction to real-world datasets without restrictions related to the data distribution, thereby extending its applicability to time-sensitive tasks.

**Why are some experimental results not state-of-the-art (SOTA)?**   As discussed in the Experiments and Results section, we do not claim to achieve state-of-the-art results. For example, the pretrained feature extractor ResNet50 was not fine-tuned for HMDB51 and UCF101 datasets. Our focus is on conducting a fair comparison rather than achieving the highest performance, as reaching state-of-the-art would not alter the conclusions of our study.

**Why don't you include a classification penalty in APR?**   While it is feasible to incorporate a classification penalty into the APR to potentially reduce errors, we aim to maintain a simple APR definition as stated in Sec. 3.2. Adding a classification penalty is redundant since the estimation of sufficient statistics inherently addresses error reduction. We utilize a density ratio estimation (DRE) algorithm to ensure statistically consistent estimation of LLRs, thereby reducing errors without the need for an additional penalty term in the APR.

## G  LAGRANGIAN FUNCTION FOR CONVEX FUNCTION LEARNING

In this section, we review the 2-block ADMM algorithm (Siahkamari et al., 2022) used for solving the convex regression problem (Eq. 12). We solve the following noisy convex regression problem with regularization:

$$\hat{f} \triangleq \arg\min_f \frac{1}{n} \sum_{i=1}^n (y_i - f(\boldsymbol{x}_i))^2 + \lambda \|f\|, \tag{15}$$

where $\boldsymbol{x}_i \in \mathbb{R}^d$, and $\lambda$ is a hyperparameter affecting convergence. Note that regression labels $y_i$ are noisy; i.e., they have bounded random discrepancies from the true label.

The 2-block ADMM solves this problem by using piecewise linear functions:

$$\min_{\hat{y}_i, a_i} \frac{1}{n} \sum_{i=1}^n (\hat{y}_i - y_i)^2 + \lambda \sum_{l=1}^d \max_{i=1}^n |a_{i,l}| \tag{16}$$

$$\text{s.t. } \hat{y}_i - \hat{y}_j - \langle a_i, \boldsymbol{x}_i - \boldsymbol{x}_j \rangle \leq 0 \quad i,j \in [n] \times [n]. \tag{17}$$

Then, we estimate $f(\boldsymbol{x})$ via

$$\hat{f}(\boldsymbol{x}) \triangleq \max_i \langle a_i, \boldsymbol{x} - \boldsymbol{x}_i \rangle + \hat{y}_i. \tag{18}$$

The 2-block ADMM is summarized in Algorithms 1 & 2 and Updates 1–4 below. The algorithm uses the augmented Lagrange method and leverages the decomposition of the optimization problem into two blocks that are updated iteratively, focusing on the primal and dual variables. In Siahkamari et al. (2022), it is proven that the 2-block ADMM converges to the ground truth function $f$ when $\mathscr{T} \to \infty$ and $n \to \infty$ if $\lambda$ is in an appropriate region (e.g., $\lambda \geq \frac{3}{\sqrt{2nd}}$ is necessary). The convergence rate is also derived. The algorithm is implemented in Python class `ConvexRegressionModel` in our code.

**Update 1.**

$$\boldsymbol{a}_i = \boldsymbol{\lambda}_i \left( \boldsymbol{\theta}_i + \hat{y}_i \boldsymbol{x}_i + \frac{1}{n} \sum_k \hat{y}_k \boldsymbol{x}_k \right), \tag{19}$$

where

$$\boldsymbol{\lambda}_i \triangleq \left( \boldsymbol{x}_i \boldsymbol{x}_i^{\mathscr{T}} + \frac{1}{n} I + \frac{1}{n} \sum_j \boldsymbol{x}_j \boldsymbol{x}_j^{\mathscr{T}} \right)^{-1},$$

$$\boldsymbol{\theta}_i \triangleq \frac{1}{n} \left( \boldsymbol{p}_i^+ - \boldsymbol{p}_i^- - \boldsymbol{\eta}_i + \sum_j (\alpha_{i,j} + s_{i,j})(\boldsymbol{x}_i - \boldsymbol{x}_j) \right).$$

**Update 2.**

$$\hat{\boldsymbol{y}} = \boldsymbol{\Omega}^{-1} \left( \frac{2\boldsymbol{y}}{n^2 \rho} + \boldsymbol{v} - \boldsymbol{\beta} \right) \tag{20}$$

where $\boldsymbol{y} = [y_1, \ldots, y_n]^{\mathscr{T}}$, $\hat{\boldsymbol{y}} = [\hat{y}_1, \ldots, \hat{y}_n]^{\mathscr{T}}$, and

$$\boldsymbol{\beta}_i \triangleq \frac{1}{n} \sum_j \alpha_{i,j} - \alpha_{j,i} + s_{i,j} - s_{j,i},$$

$$v_i \triangleq \boldsymbol{x}_i^{\mathscr{T}} \boldsymbol{\lambda}_i \boldsymbol{\theta}_i + \boldsymbol{x}_i^{\mathscr{T}} \frac{1}{n} \sum_j \boldsymbol{\lambda}_j \boldsymbol{\theta}_j - \frac{1}{n} \sum_j \boldsymbol{x}_j^{\mathscr{T}} \boldsymbol{\lambda}_j \boldsymbol{\theta}_j,$$

$$\boldsymbol{\Omega}_{i,j} \triangleq \left( \frac{2}{n^2 \rho} + 2 - \boldsymbol{x}_i^{\mathscr{T}} \boldsymbol{\lambda}_i \boldsymbol{x}_i \right) \mathbf{1}(i=j) - \frac{1}{n} D_{i,j},$$

$$D_{i,j} \triangleq \boldsymbol{x}_i^{\mathscr{T}} \left( \boldsymbol{\lambda}_i + \boldsymbol{\lambda}_j + \frac{1}{n} \sum_k \boldsymbol{\lambda}_k \right) \boldsymbol{x}_j - \boldsymbol{x}_j^{\mathscr{T}} \boldsymbol{\lambda}_j \boldsymbol{x}_j - \frac{1}{n} \sum_k \boldsymbol{x}_k \boldsymbol{\lambda}_k \boldsymbol{x}_j.$$

---

**Algorithm 1** L-update

---

**Require:** $\{\gamma_i, c_i\}_{i=1}^n$, and $\rho/\lambda$
 1: $knot_{2n}, \ldots, knot_1 \leftarrow \text{sort}\{\gamma_i + c_i, \gamma_i - c_i\}_{i=1}^n$
 2: $f \leftarrow \lambda/\rho$
 3: $f' \leftarrow 0$
 4: **for** $j = 2$ to $2n$ **do**
 5:     $f' \leftarrow f' + \frac{1}{2}$
 6:     $f \leftarrow f + f' \cdot (knot_j - knot_{j-1})$
 7:     **if** $f \leq 0$ **then**
 8:         **return** $(knot_j - \frac{f}{f'})^+$
 9:     **end if**
10: **end for**
11: **return** $(knot_{2n} - \frac{f}{n})^+$

---

**Algorithm 2** Convex regression

---

**Require:** $\{(\boldsymbol{x}_i, y_i)\}_{i=1}^n, \rho, \lambda$, and $\mathscr{T}$
 1: $\hat{y}_i = s_{i,j} = \alpha_{i,j} \leftarrow 0$
 2: $\boldsymbol{L} = \boldsymbol{a}_i = \boldsymbol{p}_i = \boldsymbol{u}_i = \boldsymbol{\eta}_i = \boldsymbol{\gamma}_i \leftarrow \boldsymbol{0}_{d \times 1}$
 3: **for** $t = 1$ to $\mathscr{T}$ **do**
 4:     **Update** $\hat{y}$ **by Eq. 20**
 5:     **Update** $\boldsymbol{a}_i$ **by Eq. 19**
 6:     $L_l \leftarrow \textbf{L\_update}(\{\gamma_{i,l}, |\eta_{i,l} + a_{i,l}|\}_{i \in [n]}, \lambda/\rho)$
 7:     **Update** $u_{i,l}, p_{i,l}^+, p_{i,l}^-, s_{i,j}$ **by Eq. 21**
 8:     **Update** $\alpha_{i,j}, \gamma_{i,l}, \eta_{i,l}$ **by Eq. 22**
 9: **end for**
10: **return** $f(\cdot) \triangleq \max_{i=1}^n (\langle \boldsymbol{a}_i, \cdot - \boldsymbol{x}_i \rangle + \hat{y}_i)$

---

**Update 3.**

$$
\begin{aligned}
s_{i,j} &= (-\alpha_{i,j} - \hat{y}_i + \hat{y}_j + \langle a_i, x_i - x_j \rangle)^+ , \\
u_{i,l} &= (L_l - \gamma_{i,l} - |\eta_{i,l} + a_{i,l}|)^+ , \\
p_{i,l}^+ &= \frac{1}{2}(L_l - \gamma_{i,l} - u_{i,l} + \eta_{i,l} + a_{i,l})^+ , \\
p_{i,l}^- &= \frac{1}{2}(L_l - \gamma_{i,l} - u_{i,l} - \eta_{i,l} - a_{i,l})^- .
\end{aligned}
\tag{21}
$$

**Update 4.**

$$
\begin{aligned}
\alpha_{i,j} &= \alpha_{i,j} + s_{i,j} \\
&\quad \hat{y}_i - \hat{y}_j - \langle \boldsymbol{a}_i, \boldsymbol{x}_i - \boldsymbol{x}_j \rangle \quad i, j \in [n] \times [n] \\
\gamma_{i,l} &= \gamma_{i,l} + u_{i,l} + p_{i,l}^+ + p_{i,l}^- - L_l \quad i, l \in [n] \times [d] \\
\eta_{i,l} &= \eta_{i,l} + a_{i,l} - p_{i,l}^+ + p_{i,l}^- \quad i, l \in [n] \times [d]
\end{aligned}
\tag{22}
$$

# H  STOCHASTIC VARIATIONAL ELBO MAXIMIZATION

The problem of evaluating $\mathbb{E}[G_{t+1}^{\min}|\mathscr{S}_t]$ at each time step is formulated below. Given any set of $M$ values at time $t$ $\{\mathscr{S}_t(X_m^{(t)})\}_{m=1}^M$, we assume that the joint distribution of the random variables $\{f(\mathscr{S}_t(X_m^{(t)}))\}_{m=1}^M$ are multivariate Gaussian distributions. *Inducing points* $Z = \{z_i\}_{i=1}^I$ with $I \ll M$ are randomly sampled from the training dataset $\{\mathscr{S}_t(X_m^{(t)})\}_{m=1}^M$. The *prior distribution* of the function is defined as:

$$\begin{bmatrix} f_{\mathscr{S}} \\ f_Z \end{bmatrix} = \mathcal{N}\left(0 \quad \begin{bmatrix} K_{\mathscr{S}\mathscr{S}} & K_{\mathscr{S}Z} \\ K_{\mathscr{S}Z}^T & K_{ZZ} \end{bmatrix}\right), \tag{23}$$

where $f_{\mathscr{S}}$ and $f_Z$ are latent functions of  and $z$, and $K$ is the covariance matrix defined with the Redial Basis Function (RBF) kernel: $K = k(s, s') = \sigma^2 \exp\left\{\frac{(s-s')^2}{2l^2}\right\}$. Note that $\sigma$ and $l$ are trainable model parameters.

To approximate the posterior distribution $p(f_{\mathscr{S}}, f_z|G_{t+1}^{\min})$, We define a *variational distribution* $q(f_{\mathscr{S}}, f_z) := p(f_{\mathscr{S}}|f_z)q(f_z)$, where the marginal variational distribution is also defined with a Gaussian: $q(f_z) = \mathcal{N}(f_z|\mu, \Sigma)$. The observed $G_{t+1}^{\min} = \{g_{t+1,1}^{\min}, g_{t+1,2}^{\min}, \ldots, g_{t+1,M}^{\min}\}$ are modeled with a *Gaussian likelihood* that assumes a homoskedastic noise is used: $p(G_{t+1}^{\min}|f) \sim N(G_{t+1}^{\min}|f, \eta^2 I)$, we compute the *marginal log likelihood* and its variational evidence lower bound (*ELBO*):

$$\log(p(G_{t+1}^{\min})) = \log \int \int p(G_{t+1}^{\min}|f_{\mathscr{S}}, f_z)p(f_{\mathscr{S}}, f_z)df_{\mathscr{S}}, df_z$$
$$\geq \mathbb{E}_{q(f_{\mathscr{S}})}\left[\log p(G_{t+1}^{\min}|f_{\mathscr{S}})\right] - D_{KL}\left[p(f_z||q(f_z)\right], \tag{24}$$

where $D_{KL}$ is the Kullback-Leibler divergence (KLD). The r.h.s. of Eq. 24 is defined as $\mathcal{L}_{\text{ELBO}}$. Given that the second term of $\mathcal{L}_{\text{ELBO}}$ is independent of training data, an empirical approximation of $\mathcal{L}_{\text{ELBO}}$ for minibatch computation can be found as:

$$\mathcal{L}_{\text{ELBO}} \sim \frac{1}{M'} \sum_{i=1}^{M'} \mathbb{E}_{q(f_{s_{it}})}\left[\log p(G_{t+1,i}^{\min}|f_{s_{ti}})\right] - D_{KL}\left[p(f_z||q(f_z)\right], \tag{25}$$

where $M' \leq M$ is a minibatch size.

After the model training, we can predict the distribution of $f_{new}$ given a new set of $\mathscr{S}_{new}$ by computing the *predictive distribution*:

$$p(f_{new}|G_{t+1}^{\min}) = \int p(f_{new}|f_z)q(f_z)df_z. \tag{26}$$

Given that $p(f_{new}, f_z)$ is a multivariate Gaussian distribution, the solution of Eq. 26 is analytical and also a Gaussian. Using $\mathscr{S}_{new} = \mathscr{S}_t$ as inputs we thus approximate $\mathbb{E}[G_{t+1}^{\min}|\mathscr{S}_t]$ with the mean of the predictive distribution, $K_{newZ}K_{ZZ}^{-1}\mu$.

# I SPRT-TANDEM

SPRT-TANDEM is a sequential DRE algorithm specifically designed for conducting SPRT on real-world sequential datasets (Ebihara et al., 2021). It employs a feature vector extractor followed by a temporal integrator (TI, Fig. 11), utilizing either recurrent networks or transformers as TIs. In our experiments, both LSTM (Hochreiter & Schmidhuber, 1997) and Transformer (Vaswani et al., 2017) are implemented. The TI outputs class posteriors, which are converted to LLRs using the TANDEM formula (Thm. I.1) in a transitive manner. Initially developed for binary-class, SPRT-TANDEM has been adapted for multiclass classification (Miyagawa & Ebihara, 2021), incorporating a statistically consistent LLR estimator, LSEL (Eq. 13). The LLR saturation problem, notably significant when the absolute value of the ground-truth LLR exceeds 100 nats, has also been addressed (Ebihara et al., 2023).

A distinctive feature of SPRT-TANDEM is its absence of a dedicated loss function for promoting earliness, despite its design for ECTS. This is because the precision in estimating the sufficient statistic (i.e., LLR) ensures the minimum required data sampling to achieve a predefined error rate. Thus, SPRT-TANDEM is trained using LSEL (Eq. 13) and multiplet cross-entropy loss (MCE, Def. I.1), without a specific loss function for earliness.

**Theorem I.1 (TANDEM formula).** *Assuming that $X^{(1,T)}$ are $N$-th order Markov series, $\lambda_{kl}(X^{(1,t)})$ can be approximated as:*

$$\lambda_{kl}(X^{(1,t)}) = \sum_{s=N+1}^{t} \log \frac{\pi_k(X^{(s-N,s)})}{\pi_l(X^{(s-N,s)})} - \sum_{s=N+2}^{t} \log \frac{\pi_k(X^{(s-N,s-1)})}{\pi_k(X^{(s-N,s-1)})} - \log \chi_{kl}, \quad (27)$$

where $\chi_{kl} = \log(p(y=k)/p(y=l))$ is a log class prior probability.

**Definition I.1 (MCE).**

$$L_{\mathrm{MCE}} := \frac{1}{M(T-N)} \sum_{i=1}^{M} \sum_{k=1}^{N+1} \sum_{t=k}^{T-(N+1-k)} \left( -\log \pi_{y_i}(X_i^{(t,t-k+1)}) \right). \quad (28)$$

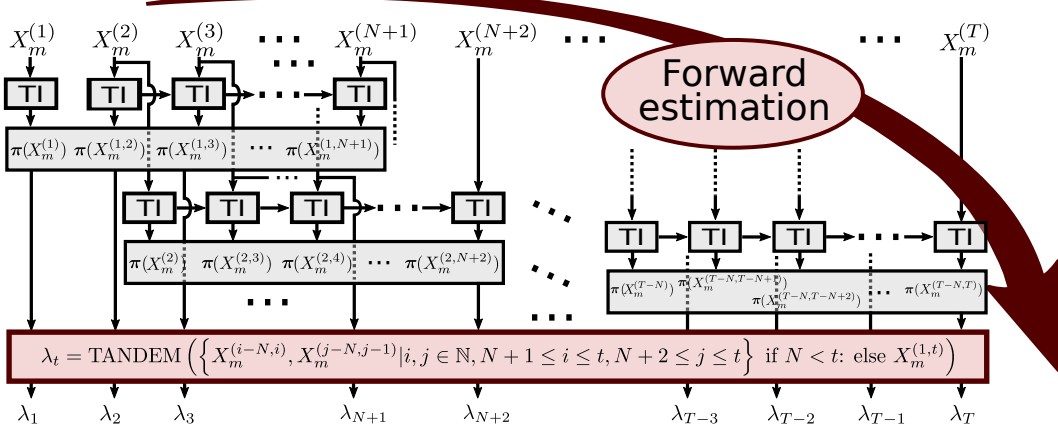

Figure 11: **LLR estimation with SPRT-TANDEM.** Feature vectors $x$, typically extracted by a feature vector extractor network, are sequentially fed into the temporal integrator (TI) network to output class posterior probabilities, $\boldsymbol{\pi} := (\pi_1, \ldots, \pi_K)$. The TANDEM formula (Eq. I.1, denoted as "TANDEM" in the figure) is used to convert these posteriors to LLRs, enabling an online sequential update of the estimation.

## J FIRMBOUND IS STATISTICALLY CONSISTENT

We provide the full assumptions, the formal statement, and the proof of Thm. 3.2.

### J.1 ASSUMPTIONS

Most of the necessary assumptions are given in the following Lems. J.1–J.4 and Thm. 2.1.

**Lemma J.1** (**CFL is statistically consistent** (Prop. 1 in (Siahkamari et al., 2022))). *See App. G for notations. With the appropriate choice of $\lambda$ which requires knowledge of the bound on $f$ and $n \geq d$, it holds that with probability at least $1 - \delta$ over the data, the estimator $\hat{f}$ of Eq. 18 has excess risk upper bounded by*

$$\mathbb{E}\left[|f(x) - \hat{f}(x)|^2\right] \leq O\left(\left(\frac{n}{d}\right)^{\frac{-2}{d+4}} \log\left(\frac{n}{d}\right) + \sqrt{\frac{\log(1/\delta)}{n}}\right). \tag{29}$$

See (Siahkamari et al., 2022) for the proof. Note that the bound limits to zero as the dataset size $n$ (denoted by $M$ in the main text) limits to infinity. Note also that $d$ used in this lemma corresponds to the number of classes $K$ in the main text.

**Lemma J.2** (CFL converges (Thm. 2 in (Siahkamari et al., 2022))). *See App. G for notations. Let $\{\hat{y}_i^t, \boldsymbol{a}_i^t\}$ be the output of Alg. 2 at the $t^{\text{th}}$ iteration, $\tilde{y}_i := \frac{1}{\mathscr{T}} \sum_{t=1}^{\mathscr{T}} \hat{y}_i^t$ and $\tilde{\boldsymbol{a}}_i := \frac{1}{\mathscr{T}} \sum_{t=1}^{\mathscr{T}} \boldsymbol{a}_i^t$. Denote $\tilde{f}_{\mathscr{T}}(\boldsymbol{x}) := \max_i \langle \tilde{\boldsymbol{a}}_i, \boldsymbol{x} - \boldsymbol{x}_i \rangle + \tilde{y}_i$. Assume $\max_{i,l} |x_{i,l}| \leq 1$ and $\mathrm{Var}(\{y_i\}_{i=1}^n) \leq 1$. If we choose*

$$\rho = \frac{\sqrt{d}\lambda^2}{n}, \quad \lambda \geq \frac{3}{\sqrt{2nd}}, \quad and \quad \mathscr{T} \geq n\sqrt{d},$$

*we have:*

$$\frac{1}{n}\sum_{i=1}^n \left(\tilde{f}_{\mathscr{T}}(\boldsymbol{x}_i) - y_i\right)^2 + \lambda\|\tilde{f}_{\mathscr{T}}\| \leq \min_{\hat{f} \in \mathcal{F}}\left(\frac{1}{n}\sum_{i=1}^n \left(\hat{f}(x_i) - y_i\right)^2 + \lambda\|\hat{f}\|\right) + \frac{6n\sqrt{d}}{\mathscr{T}+1}, \tag{30}$$

*where $\mathcal{F} := \{f : \mathbb{R}^d \to \mathbb{R} \mid f \text{ is convex}\}$.*

See (Siahkamari et al., 2022) for the proof. The inputs $\boldsymbol{x}$ and outputs $y$ in the lemma correspond to $(\pi_1(X^{(1,t)}), \ldots, \pi_K(X^{(1,t)}))$, and to $\tilde{G}_t(\mathscr{S}_t(X^{(1,t)}))$, respectively. $\pi_k(X^{(1,t)})$ for all $k \in [K]$ are obviously bounded by one, and thus, the assumption $\max_{i,l} |x_{i,l}| \leq 1$ is satisfied. Also, the assumption $\mathrm{Var}(\{y_i\}_{i=1}^n) \leq 1$ is satisfied because we only consider integrable functions, and the continuation risk $\tilde{G}_t$ is bounded. As a corollary of Lem. J.2, we have:

**Lemma J.3** (**Convergence rate of CFL** (Cor. 1 in (Siahkamari et al., 2022))). *See App. G for notations. The CFL algorithm used in* FIRMBOUND, *outlined in App. G, needs $\frac{6n\sqrt{d}}{\epsilon}$ iterations to achieve $\epsilon$ error. Each iteration requires $\mathcal{O}(n^2 d + nd^2)$ flops operations. Preprocessing costs $\mathcal{O}(nd^3)$. Therefore the total computational complexity is $\mathcal{O}\left(\frac{n^3 d^{1.5} + n^2 d^{2.5} + nd^3}{\epsilon}\right)$.*

Note that $n$ and $d$ used in this lemma correspond to $M$ and $K$ in the main text. See (Siahkamari et al., 2022) for the proof.

Next, let us define

$$L_{\text{LSEL}}[\lambda] := \frac{1}{KT} \sum_{k \in [K]} \sum_{t \in [T]} \int dX^{(1,t)} p(X^{(1,t)}|k) \log\left(1 + \sum_{l(\neq k)} e^{-\lambda_{kl}(X^{(1,t)})}\right). \tag{31}$$

Let $S := \{(X_i^{(1,T)}, y_i)\}_{i=1}^M \sim p(X^{(1,T)}, y)^M$ be a training dataset, where $M \in \mathbb{N}$ is the sample size. The empirical approximation of Eq. 31 is

$$\hat{L}_{\text{LSEL}}(\boldsymbol{w}; S) := \frac{1}{KT} \sum_{k \in [K]} \sum_{t \in [T]} \frac{1}{M_k} \sum_{i \in I_k} \log\left(1 + \sum_{l(\neq k)} e^{-\hat{\lambda}_{kl}(X_i^{(1,t)}; \boldsymbol{w})}\right). \tag{13}$$

$M_k$ and $I_k$ denote the sample size and index set of class $k$, respectively; i.e., $M_k = |I_k| = |\{i \in [M]|y_i = k\}$ and $\sum_k M_k = M$. Let $L(\boldsymbol{w})$ and $\hat{L}_S(\boldsymbol{w})$ denote $L_{\text{LSEL}}[\hat{\lambda}(\cdot; \boldsymbol{w})]$ and $\hat{L}_{\text{LSEL}}(\boldsymbol{w}; S)$, respectively. Let $\hat{\boldsymbol{w}}_S$ be the empirical risk minimizer of $\hat{L}_S$; namely, $\hat{\boldsymbol{w}}_S \in \arg\min_{\boldsymbol{w}} \hat{L}_S(\boldsymbol{w})$.

**Lemma J.4** (**LSEL is statistically consistent** (Thm. 3.1 in (Miyagawa & Ebihara, 2021))). *Let $\boldsymbol{W}^* := \left\{ \boldsymbol{w}^* \in \mathbb{R}^d \mid \hat{\lambda}(X^{(1,t)}; \boldsymbol{w}^*) = \lambda(X^{(1,t)}) \, (\forall t \in [T]) \right\}$ be the target parameter set. Assume, for simplicity of the proof, that each $\boldsymbol{w}^*$ is separated in $\boldsymbol{w}^*$; i.e., $\exists \delta > 0$ such that $\mathcal{B}(\boldsymbol{w}^*; \delta) \cap \mathcal{B}(\boldsymbol{w}^{*'}; \delta) = \emptyset$ for arbitrary $\boldsymbol{w}^*$ and $\boldsymbol{w}^{*'}$, where $\mathcal{B}(\boldsymbol{w}; \delta)$ denotes an open ball at center $\boldsymbol{w}$ with radius $\delta$. Assume the following three conditions:*

*(a) $\forall k, l \in [K], \forall t \in [T], p(X^{(1,t)} \mid k) = 0 \iff p(X^{(1,t)} \mid l) = 0.$*

*(b) $\sup_{\boldsymbol{w}} |\hat{L}_S(\boldsymbol{w}) - L(\boldsymbol{w})| \xrightarrow{\mathbb{P}} 0$ as $M \to \infty$; i.e., $\hat{L}_S(\boldsymbol{w})$ converges in probability uniformly over $\boldsymbol{w}$ to $L(\boldsymbol{w})$.*

*(c) For all $\boldsymbol{w}^* \in \boldsymbol{W}^*$, there exist $t \in [T]$, $k \in [K]$, and $l \in [K]$, such that the following $d \times d$ matrix is full-rank:*

$$\int dX^{(1,t)} p(X^{(1,t)} \mid k) \nabla_{\boldsymbol{w}^*} \hat{\lambda}_{kl}(X^{(1,t)}; \boldsymbol{w}^*) \nabla_{\boldsymbol{w}^*} \hat{\lambda}_{kl}(X^{(1,t)}; \boldsymbol{w}^*)^\top.$$

*Then, $\mathbb{P}(\hat{\boldsymbol{w}}_S \notin W^*) \xrightarrow{M \to \infty} 0$; i.e., $\hat{\boldsymbol{w}}_S$ converges in probability into $\boldsymbol{W}^*$.*

See (Miyagawa & Ebihara, 2021) for the proof. Assumption (a) ensures that LLRs $\lambda(X^{(1,t)}) := \{\lambda_{kl}(X^{(1,t)})\}_{k,l \in [K]}$ exists and is finite. Assumption (b) can be satisfied under the standard assumptions of the uniform law of large numbers (compactness, continuity, measurability, and dominance) (Jennrich, 1969; Newey & McFadden, 1986). Assumption (c) is a technical requirement, often assumed in the literature (Gutmann & Hyvärinen, 2012). We additionally assume that the neural network represented by $\boldsymbol{w}$ is so large that it can represent target LLRs, which can be satisfied according to the universal approximation theorem of neural networks.

## J.2 FORMAL STATEMENT

Now, we provide the formal statement of Thm. 3.2:

**Theorem J.1** (`FIRMBOUND` **is statistically consistent**). *Suppose that all the assumptions mentioned in App. J.1 are satisfied. Suppose that we have the sufficient statistics estimated with LSEL on a dataset with size $M$. Suppose also that we have the continuation risk estimated on with CFL the same dataset. Then, with arbitrary precision, we can solve the backward induction equation in Thm. 2.1, which yields the Bayes optimal terminal decision rule $d^*$ and stopping time $\tau^*$, with high probability over the data and as $M \to \infty$.*

## J.3 PROOF

We provide the proof of Thm. J.1.

*Proof.* We first show that the CFL combined with the density ratio estimation (DRE) with LSEL yields a consistent estimate of function $\tilde{G}_t$.

**Observation 1.** According to Lems. J.3 & J.2, for any dataset, the output function of the CFL algorithm can be arbitrarily close to any convex function if $\mathscr{T}, M \to \infty$ with $\mathscr{T} > \Omega(M\sqrt{K})$, where $\Omega(\cdot)$ here denotes a Landau symbol. Therefore, according to Lem. J.1, with high probability over the data, the output function of the CFL algorithm can be arbitrarily close to any convex function if $\mathscr{T}, M \to \infty$ with $\mathscr{T} > \Omega(M\sqrt{K})$.

**Observation 2.** According to Lem. J.4, the estimated LLRs $\hat{\lambda}_{kl}$ can be arbitrarily close to the true LLRs $\lambda_{kl}$ as $M \to \infty$.

**CFL with DRE is consistent.** Therefore, according to **Observation 1** & **2**, with high probability over the data, as $M \to \infty$, CFL with DRE can estimate any continuation risk $\tilde{G}_t$ at any $\mathscr{S}_t (= (\pi_1, \ldots, \pi_K))$ because $\tilde{G}_t$ is a continuous function of $\mathscr{S}_t$. That is, CFL with DRE is a statistically consistent estimator of $\tilde{G}_t$.

**FIRMBOUND is consistent.** Using the estimated continuation risk, we can solve the backward induction equation in Thm. 2.1 with arbitrary precision, which yields the Bayes optimal terminal decision rule $d^*$ and stopping time $\tau^*$, with high probability over the data and as $M \to \infty$. This means that FIRMBOUND (= CFL + DRE + backward induction) yields a statistically consistent estimator of the Bayes optimal algorithm in the sense of Thm. 2.1, minimizing AAPR. □

## K    EXPERIMENTAL DETAILS AND SUPPLEMENTARY RESULTS

Throughout the experiments, Optuna (Akiba et al., 2019) with the default algorithm, Tree-structured Parzen Estimator (TPE) (Bergstra et al., 2011), is used to find the best hyperparameter combinations from the predefined search space. TPE is a Bayesian optimization algorithm that models beliefs about the optimal hyperparameters using Parzen Estimation and optimizes the search process using a tree-like graph. The training procedure described below is common across all datasets unless specified otherwise.

### K.1    FIRMBOUND WITH CFL

Our custom code enables hyperparameter tuning at each time step, determining the *lambda* parameter (not to be confused with LLR $\lambda$; we maintain the original notation from Siahkamari et al. (2022) for consistency) used in the augmented Lagrangian algorithm. The concave conditional expectation is negated for optimizing CFL models. Adam (Kingma & Ba, 2014) is employed as the optimizer.

**Tuning.**    A total of 1000 data points (i.e., posteriors $\pi$ as the sufficient statistic) are randomly selected from the training dataset. Using Optuna, we search for the optimal *lambda* at each time step as follows: the initial value of *lambda* is log-uniformly selected from the range $[1e-3, 1e1]$. A 5-fold cross-validation, consisting of 3 epochs each, is conducted to evaluate the mean squared error between the predictions and observed data points. This tuning trial is repeated 30 times to ensure comprehensive parameter exploration.

**Fitting.**    A subset of 5000 data points is randomly selected from the training dataset. The optimal *lambda* parameter, identified from the tuning trials, is used to train the final CFL models over 3 epochs on training data, which will be used for future online ECTS. The evaluation of AAPR and SAT curves is conducted 5 times on test data to validate performance.

### K.2    FIRMBOUND WITH GP REGRESSION

Similar to CFL, GP regression models are trained at each time step $t$. Adam (Kingma & Ba, 2014) is utilized as the optimizer.

**Initialization.**    A Cholesky Variational Distribution is used to estimate the true posterior, initialized with 200 inducing points (i.e., sufficient statistics, either LLRs or posteriors) that are randomly selected from the training dataset. The GP model is initialized with a constant mean and a covariance module, the latter employing a Radial Basis Function (RBF) kernel. A Gaussian likelihood module is also initialized to evaluate the Evidence Lower Bound (ELBO).

**Fitting.**    The negative variational ELBO is computed and minimized across minibatches of size 2000. After 30 epochs of training on training data, the predictive distribution is evaluated on all sufficient statistics in the training data to assess the conditional expectation. The evaluation of AAPR and SAT curves is repeated 30 times on test data.

**Supplementary Results.**    Fig. 12 shows representative fitting results on the two-class sequential Gaussian dataset.

### K.3    DATASET PREPARATION

Following the methodologies described in Ebihara et al. (2021) and  Miyagawa & Ebihara (2021), we prepare feature vectors for the SiW and action recognition datasets UCF101/HMDB51, respectively. All pixel values are divided by 127.5 and then subtracted by 1 before feeding into the feature extractor. For the SiW videos, we use ResNet152 version 2 (He et al., 2016a;b) to produce a 512-dimensional feature vector (trainable parameters: 3.7M). For the UCF101 and HMDB51 videos, we employ the Pretrained Microsoft Vision Model ResNet50, which is used without fine-tuning to extract 2048-dimensional vector elements (trainable parameters: 23.5M). The train/test split for UCF101 and HMDB51 adheres to official splitting pattern #1. A validation set is derived from the training dataset

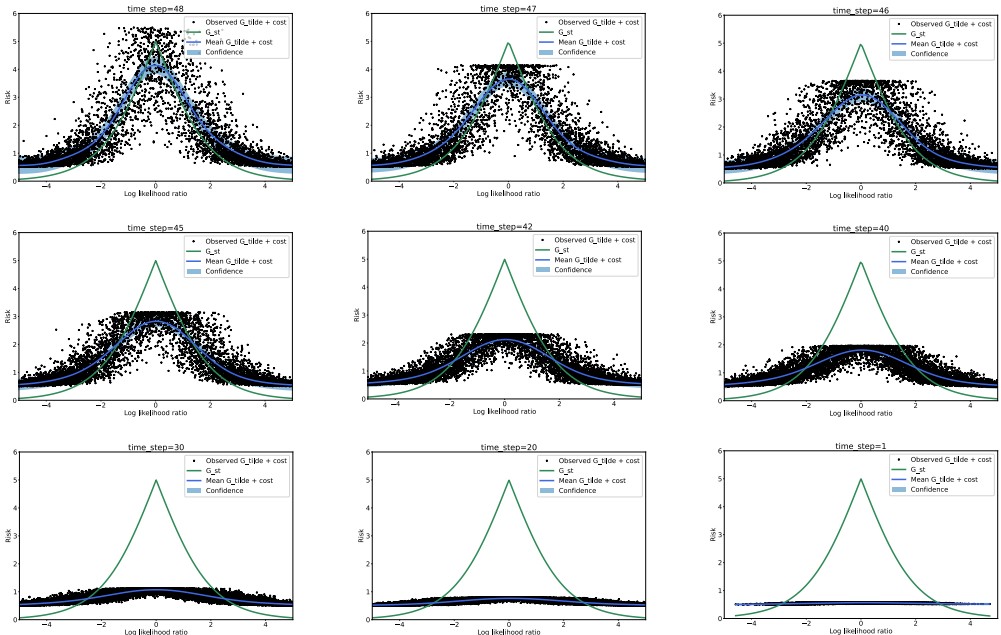

Figure 12: **Typical risk curves estimated with GP regression.** Two-class Gaussian distribution dataset is used to generate observed continuation risk $\tilde{G}$, on which GP models are trained to provide estimations of the conditional expectation.

while maintaining the original class frequency. All videos are clipped or repeated to standardize the time steps to 50 and 79, respectively.

Table 3: Extracted datasets.

| Orig. dataset | Train set | Val. set | Test set | Feat. dim | Time steps |
|---|---|---|---|---|---|
| SiW (Liu et al., 2018b) | 46,729 | 4,968 | 43,878 | 512 | 50 |
| HMDB51 (Kuehne et al., 2011) | 1,026 | 106 | 105 | 2,048 | 79 |
| UCF101 (Soomro et al., 2012) | 35,996 | 4,454 | 15,807 | 2,048 | 50 |
| FordA (Soomro et al., 2012) | 6,600 | 6,005 | 12,000 | 24 | 20 |

### K.4 TRAINING ECTS MODELS

For real-world datasets lacking ground-truth LLRs, we train the DRE model SPRT-TANDEM (Ebihara et al., 2021) to provide a statistically consistent estimator of LLRs. Additionally, ECTS baseline models, including LSTMms (Ma et al., 2016), EARLIEST (Hartvigsen et al., 2019), TCN-Transformer (Chen et al., 2022), and CALIMERA (Bilski & Jastrzębska, 2023) are trained.

Similar to FIRMBOUND, we utilize Optuna for hyperparameter optimization. The evaluation criterion is the averaged per-class error rate, or macro-averaged recall. A conservative $40\%$ percentile pruner is used for early stopping of unpromising parameter combinations. The training settings common across models and databases, along with detailed pruner settings, are provided in Tab. 4. The choice of optimizer includes Adam (Kingma & Ba, 2014), RMSprop (Graves, 2013), and Lion (Chen et al., 2023). An exception is CALIMERA, which are trained with fixed parameters. CALIMERA employs linear and ridge classifiers, which leverage closed-form solutions for parameter estimation, ensuring a deterministic and efficient optimization process that is less prone to the hyperparameter sensitivities often associated with deep learning models.

In subsequent analyses, the coefficient $\gamma$ demonstrates that batch size and learning rate can be scaled equivalently to maintain consistent training dynamics, as per the linear scaling law (Goyal et al., 2017).

Table 4: Common Hyperparameter Tuning Setup

| | |
|---|---|
| Number of iterations | Number of training data * Number of epochs / Batch size |
| Pruner type | 40% percentile |
| Pruner startup trials | Number of trials / 2 |
| Pruner warmup steps | Number of iterations / 2 |
| Pruner interval steps | Number of iterations / Number of epochs |

### K.4.1 TWO-CLASS GAUSSIAN DATSET

Table 5: SPRT-TANDEM on two-class Gaussian datset: parameter space.

| | Hyperparameter | Space | Optimal value |
|---|---|---|---|
| Fixed parameters | Batch size | $200 \times \gamma$ | N.A. (fixed) |
| | Epochs | $15 \times \gamma$ | N.A. (fixed) |
| | # Tuning trials | 200 | N.A. (fixed) |
| | # Repeated test trials | 60 | N.A. (fixed) |
| Searched hyperparameters | Learning rate | $[10^{-6}, 10^{-3}] \times \gamma$ | 0.0001 |
| | Weight decay | $[0.0, 10^{-5}]$ | 0.0005 |
| | Optimizer | {Adam, RMSprop, Lion} | Adam |
| | Order SPRT | $\{0, 1, \ldots, 10\}$ | 5 |
| | MCE weight | $[0.0, 1.0]$ | 1.0 |
| | LLR estim. loss weight | $[0.0, 1.0]$ | 0.8 |
| | FC activation | {B2Bsqrt, tanh, ReLU, GeLU} | ReLU |
| | Temporal integrator | {LSTM, Transformer} | Transformer |
| Backbone-specific parameters | num blocks | $[1, 3]$ | 1 |
| | num heads | $[2, 4]$ | 4 |
| | Dropout | $[0.0, 0.5]$ | 0.4 |
| | MLP_units | $[32, 64]$ | 64 |
| | FF_dim | $[32, 64]$ | 64 |

### K.4.2 THREE-CLASS GAUSSIAN DATASET

Table 6: SPRT-TANDEM on three-class Gaussian datset: parameter space.

| | Hyperparameter | Space | Optimal value |
|---|---|---|---|
| Fixed parameters | Batch size | $200 \times \gamma$ | N.A. (fixed) |
| | Epochs | $15 \times \gamma$ | N.A. (fixed) |
| | # Tuning trials | 200 | N.A. (fixed) |
| | # Repeated test trials | 30 | N.A. (fixed) |
| Searched hyperparameters | Learning rate | $[10^{-6}, 10^{-3}] \times \gamma$ | 0.0001 |
| | Weight decay | $[0.0, 10^{-5}]$ | 0.00025 |
| | Optimizer | {Adam, RMSprop, Lion} | Lion |
| | Order SPRT | $\{0, 1, \ldots, 10\}$ | 0 |
| | MCE weight | $[0.0, 1.0]$ | 0.7 |
| | LLR estim. loss weight | $[0.0, 1.0]$ | 0.1 |
| | FC activation | {B2Bsqrt, tanh, ReLU, GeLU} | ReLU |
| | Temporal integrator | {LSTM, Transformer} | Transformer |
| Backbone-specific parameters | num blocks | $[1, 3]$ | 1 |
| | num heads | $[2, 4]$ | 2 |
| | Dropout | $[0.0, 0.5]$ | 0.3 |
| | MLP_units | $[32, 64]$ | 64 |
| | FF_dim | $[32, 64]$ | 32 |

### K.4.3 SIW

Table 7: SPRT-TANDEM on SiW: parameter space.

|  | Hyperparameter | Space | Optimal value |
|---|---|---|---|
| Fixed parameters | Batch size | $83 \times \gamma$ | N.A. (fixed) |
|  | Epochs | 18 | N.A. (fixed) |
|  | # Tuning trials | 200 | N.A. (fixed) |
|  | # Repeated test trials | 200 | N.A. (fixed) |
| Searched hyperparameters | Learning rate | $[10^{-6}, 10^{-3}] \times \gamma$ | 0.0001 |
|  | Weight decay | $[0.0, 10^{-5}]$ | 0.0 |
|  | Optimizer | {Adam, RMSprop, Lion} | Adam |
|  | Order SPRT | $\{0, 1, \ldots, 10\}$ | 9 |
|  | MCE weight | $[0.0, 1.0]$ | 1.0 |
|  | LLR estim. loss weight | $[0.0, 1.0]$ | 1.0 |
|  | FC activation | {B2Bsqrt, tanh, ReLU, GeLU} | ReLU |
|  | Temporal integrator | {LSTM, Transformer} | LSTM |
| Backbone-specific parameters | LSTM output activation | {B2Bsqrt, tanh, GeLU} | B2Bsqrt |
|  | LSTM hidden dim. | $[32, 256]$ | 256 |

Table 8: LSTMms on SiW: parameter space.

|  | Hyperparameter | Space | Optimal value |
|---|---|---|---|
| Fixed parameters | Batch size | $100 \times \gamma$ | N.A. (fixed) |
|  | Epochs | 15 | N.A. (fixed) |
|  | # Tuning trials | 100 | N.A. (fixed) |
|  | # Repeated test trials | 26 | N.A. (fixed) |
| Searched hyperparameters | Learning rate | $[10^{-6}, 10^{-3}] \times \gamma$ | 0.0011 |
|  | Weight decay | $[0.0, 10^{-5}]$ | 0.001 |
|  | Optimizer | {Adam, RMSprop, Lion} | Lion |
|  | Cross entropy weight | $[0.0, 1.0]$ | 1.0 |
|  | Loss type | {LSTMm, LSTMs} | LSTMs |
|  | Loss weight | $[0.0, 1.0]$ | 1.0 |
|  | LSTM hidden dim. | $[32, 512]$ | 76 |

Table 9: EARLIEST (lambda=1e-1) on SiW: parameter space.

|  | Hyperparameter | Space | Optimal value |
|---|---|---|---|
| Fixed parameters | EARLIEST param. lambda | $1e - 1$ | N.A. (fixed) |
|  | Batch size | $256 \times \gamma$ | N.A. (fixed) |
|  | Epochs | 50 | N.A. (fixed) |
|  | # Tuning trials | 200 | N.A. (fixed) |
|  | # Repeated test trials | 30 | N.A. (fixed) |
| Searched hyperparameters | Learning rate | $[10^{-6}, 10^{-3}] \times \gamma$ | 0.000951 |
|  | Weight decay | $[0.0, 10^{-5}]$ | 0.0006 |
|  | Optimizer | {Adam, RMSprop, Lion} | Lion |
|  | LSTM hidden dim. | $[32, 256]$ | 16 |

Table 10: EARLIEST (lambda=1e-10) on SiW: parameter space.

|  | Hyperparameter | Space | Optimal value |
|---|---|---|---|
| Fixed parameters | EARLIEST param. lambda | $1e-10$ | N.A. (fixed) |
|  | Batch size | $256 \times \gamma$ | N.A. (fixed) |
|  | Epochs | 50 | N.A. (fixed) |
|  | # Tuning trials | 200 | N.A. (fixed) |
|  | # Repeated test trials | 30 | N.A. (fixed) |
| Searched hyperparameters | Learning rate | $[10^{-6}, 10^{-3}] \times \gamma$ | 0.000441 |
|  | Weight decay | $[0.0, 10^{-5}]$ | 0.001 |
|  | Optimizer | {Adam, RMSprop, Lion} | RMSprop |
|  | LSTM hidden dim. | [32, 256] | 16 |

Table 11: TCNT (alpha=0.3) on SiW: parameter space.

|  | Hyperparameter | Space | Optimal value |
|---|---|---|---|
| Fixed parameters | TCNT param. alpha | 0.3 | N.A. (fixed) |
|  | Batch size | $256 \times \gamma$ | N.A. (fixed) |
|  | Epochs | 20 | N.A. (fixed) |
|  | # Tuning trials | 100 | N.A. (fixed) |
|  | # Repeated test trials | 30 | N.A. (fixed) |
| Searched hyperparameters | Learning rate | $[10^{-6}, 10^{-3}] \times \gamma$ | 0.00425 |
|  | Weight decay | $[0.0, 10^{-5}]$ | 0.0003 |
|  | Dropout | [0.0, 0.5] | 0.3 |
|  | Optimizer | {Adam, RMSprop, Lion} | Lion |
|  | # Blocks | [1, 3] | 1 |
|  | # Num heads | [2, 4] | 4 |
|  | TCN channels | [256, 1024] | 256 |

Table 12: TCNT (alpha=0.5) on SiW: parameter space.

|  | Hyperparameter | Space | Optimal value |
|---|---|---|---|
| Fixed parameters | TCNT param. alpha | 0.5 | N.A. (fixed) |
|  | Batch size | $256 \times \gamma$ | N.A. (fixed) |
|  | Epochs | 20 | N.A. (fixed) |
|  | # Tuning trials | 100 | N.A. (fixed) |
|  | # Repeated test trials | 30 | N.A. (fixed) |
| Searched hyperparameters | Learning rate | $[10^{-6}, 10^{-3}] \times \gamma$ | 0.000002 |
|  | Weight decay | $[0.0, 10^{-5}]$ | 0.000 |
|  | Dropout | [0.0, 0.5] | 0.4 |
|  | Optimizer | {Adam, RMSprop, Lion} | Lion |
|  | # Blocks | [1, 3] | 1 |
|  | # Num heads | [2, 4] | 4 |
|  | TCN channels | [256, 512] | 32 |

Table 13: CALIMERA on SiW: parameter space.

|  | Hyperparameter | Space | Optimal value |
|---|---|---|---|
| Fixed parameters | Delay penalty | {0.1, 0.5, 1.0} | N.A. (fixed) |
|  | # Repeated test trials | 5 | N.A. (fixed) |

### K.4.4 HMDB51

Table 14: SPRT-TANDEM on HMDB51: parameter space.

|  | Hyperparameter | Space | Optimal value |
|---|---|---|---|
| Fixed parameters | Batch size | $128 \times \gamma$ | N.A. (fixed) |
|  | Epochs | 24 | N.A. (fixed) |
|  | # Tuning trials | 200 | N.A. (fixed) |
|  | # Repeated test trials | 5 | N.A. (fixed) |
| Searched hyperparameters | Learning rate | $[10^{-6}, 10^{-3}] \times \gamma$ | $10^{-4}$ |
|  | Weight decay | $[0.0, 10^{-5}]$ | $10^{-4}$ |
|  | Optimizer | {Adam, RMSprop, Lion} | Adam |
|  | Order SPRT | $\{0, 1, \ldots, 10\}$ | 4 |
|  | MCE weight | $[0.0, 1.0]$ | 0.1 |
|  | LLR estim. loss type | {LLLR, LSEL} | LSEL |
|  | LLR estim. loss weight | $[0.0, 1.0]$ | 1.0 |
|  | FC activation | {B2Bsqrt, tanh, ReLU, GeLU} | tanh |
|  | Temporal integrator | {LSTM, Transformer} | LSTM |
| Backbone-specific parameters | LSTM output activation | {B2Bsqrt, tanh, GeLU} | B2Bsqrt |
|  | LSTM hidden dim. | $[32, 256]$ | 256 |

Table 15: LSTMms on HMDB: parameter space.

|  | Hyperparameter | Space | Optimal value |
|---|---|---|---|
| Fixed parameters | Batch size | $100 \times \gamma$ | N.A. (fixed) |
|  | Epochs | 15 | N.A. (fixed) |
|  | # Tuning trials | 100 | N.A. (fixed) |
|  | # Repeated test trials | 20 | N.A. (fixed) |
| Searched hyperparameters | Learning rate | $[10^{-6}, 10^{-3}] \times \gamma$ | 0.000594 |
|  | Weight decay | $[0.0, 10^{-5}]$ | 0.0009 |
|  | Optimizer | {Adam, RMSprop, Lion} | Lion |
|  | Cross entropy weight | $[0.0, 1.0]$ | 0.7 |
|  | Loss type | {LSTMm, LSTMs} | LSTMm |
|  | Loss weight | $[0.0, 1.0]$ | 0.3 |
|  | LSTM hidden dim. | $[32, 512]$ | 282 |

Table 16: EARLIEST (lambda=1e-1) on HMDB51: parameter space.

|  | Hyperparameter | Space | Optimal value |
|---|---|---|---|
| Fixed parameters | EARLIEST param. lambda | $1e-1$ | N.A. (fixed) |
|  | Batch size | $256 \times \gamma$ | N.A. (fixed) |
|  | Epochs | 50 | N.A. (fixed) |
|  | # Tuning trials | 200 | N.A. (fixed) |
|  | # Repeated test trials | 30 | N.A. (fixed) |
| Searched hyperparameters | Learning rate | $[10^{-6}, 10^{-3}] \times \gamma$ | 0.000273 |
|  | Weight decay | $[0.0, 10^{-5}]$ | 0.0009 |
|  | Optimizer | {Adam, RMSprop, Lion} | RMSprop |
|  | LSTM hidden dim. | $[32, 256]$ | 159 |

Table 17: EARLIEST (lambda=1e-10) on HMDB51: parameter space.

|  | Hyperparameter | Space | Optimal value |
|---|---|---|---|
| Fixed parameters | EARLIEST param. lambda | $1e-10$ | N.A. (fixed) |
|  | Batch size | $256 \times \gamma$ | N.A. (fixed) |
|  | Epochs | 50 | N.A. (fixed) |
|  | # Tuning trials | 200 | N.A. (fixed) |
|  | # Repeated test trials | 30 | N.A. (fixed) |
| Searched hyperparameters | Learning rate | $[10^{-6}, 10^{-3}] \times \gamma$ | 0.000148 |
|  | Weight decay | $[0.0, 10^{-5}]$ | 0.000 |
|  | Optimizer | {Adam, RMSprop, Lion} | Lion |
|  | LSTM hidden dim. | $[32, 256]$ | 147 |

Table 18: TCNT (alpha=0.3) on HMDB51: parameter space.

|  | Hyperparameter | Space | Optimal value |
|---|---|---|---|
| Fixed parameters | TCNT param. alpha | 0.3 | N.A. (fixed) |
|  | Batch size | $256 \times \gamma$ | N.A. (fixed) |
|  | Epochs | 15 | N.A. (fixed) |
|  | # Tuning trials | 300 | N.A. (fixed) |
|  | # Repeated test trials | 10 | N.A. (fixed) |
| Searched hyperparameters | Learning rate | $[10^{-6}, 10^{-3}] \times \gamma$ | 0.000776 |
|  | Weight decay | $[0.0, 10^{-5}]$ | 0.000 |
|  | Dropout | $[0.0, 0.5]$ | 0.1 |
|  | Optimizer | {Adam, RMSprop, Lion} | Adam |
|  | # Blocks | $[1, 3]$ | 1 |
|  | # Num heads | $[2, 4]$ | 2 |
|  | TCN channels | $[256, 1024]$ | 1024 |

Table 19: TCNT (alpha=0.5) on HMDB51: parameter space.

|  | Hyperparameter | Space | Optimal value |
|---|---|---|---|
| Fixed parameters | TCNT param. alpha | 0.5 | N.A. (fixed) |
|  | Batch size | $256 \times \gamma$ | N.A. (fixed) |
|  | Epochs | 15 | N.A. (fixed) |
|  | # Tuning trials | 300 | N.A. (fixed) |
|  | # Repeated test trials | 10 | N.A. (fixed) |
| Searched hyperparameters | Learning rate | $[10^{-6}, 10^{-3}] \times \gamma$ | 0.000453 |
|  | Weight decay | $[0.0, 10^{-5}]$ | 0.010 |
|  | Dropout | $[0.0, 0.5]$ | 0.4 |
|  | Optimizer | {Adam, RMSprop, Lion} | Adam |
|  | # Blocks | $[1, 3]$ | 1 |
|  | # Num heads | $[2, 4]$ | 4 |
|  | TCN channels | $[256, 1024]$ | 1024 |

Table 20: CALIMERA on HMDB51: parameter space.

|  | Hyperparameter | Space | Optimal value |
|---|---|---|---|
| Fixed parameters | Delay penalty | $\{0.1, 0.5, 1.0\}$ | N.A. (fixed) |
|  | # Repeated test trials | 5 | N.A. (fixed) |

### K.4.5 UCF101

Table 21: SPRT-TANDEM on UCF101: parameter space.

|  | Hyperparameter | Space | Optimal value |
|---|---|---|---|
| Fixed parameters | Batch size | $16 \times \gamma$ | N.A. (fixed) |
|  | Epochs | 25 | N.A. (fixed) |
|  | # Tuning trials | 200 | N.A. (fixed) |
|  | # Repeated test trials | 14 | N.A. (fixed) |
| Searched hyperparameters | Learning rate | $[10^{-6}, 10^{-3}] \times \gamma$ | 0.000027 |
|  | Weight decay | $[0.0, 10^{-5}]$ | 0.0006 |
|  | Optimizer | {Adam, RMSprop, Lion} | Lion |
|  | Order SPRT | {0, 1, ..., 10} | 6 |
|  | MCE weight | [0.0, 1.0] | 0.2 |
|  | LLR estim. loss type | {LSEL, LLLR} | LSEL |
|  | LLR estim. loss weight | [0.0, 1.0] | 0.4 |
|  | FC activation | {B2Bsqrt, tanh, ReLU, GeLU} | tanh |
|  | Temporal integrator | {LSTM, Transformer} | Transformer |
| Backbone-specific parameters | # Blocks | [1, 3] | 1 |
|  | # Heads | [2, 4] | 4 |
|  | Dropout | [0.0, 0.5] | 0.1 |
|  | MLP_units | [256, 416] | 288 |
|  | FF_dim | [256, 416] | 256 |

Table 22: LSTMms on UCF101: parameter space.

|  | Hyperparameter | Space | Optimal value |
|---|---|---|---|
| Fixed parameters | Batch size | $100 \times \gamma$ | N.A. (fixed) |
|  | Epochs | 15 | N.A. (fixed) |
|  | # Tuning trials | 100 | N.A. (fixed) |
|  | # Repeated test trials | 11 | N.A. (fixed) |
| Searched hyperparameters | Learning rate | $[10^{-6}, 10^{-3}] \times \gamma$ | 0.000184 |
|  | Weight decay | $[0.0, 10^{-5}]$ | 0.008 |
|  | Optimizer | {Adam, RMSprop, Lion} | RMSprop |
|  | Cross entropy weight | [0.0, 1.0] | 0.5 |
|  | Loss type | {LSTMm, LSTMs} | LSTMs |
|  | Loss weight | [0.0, 1.0] | 0.4 |
|  | LSTM hidden dim. | [32, 512] | 362 |

Table 23: EARLIEST (lambda=1e-1) on UCF101: parameter space.

|  | Hyperparameter | Space | Optimal value |
|---|---|---|---|
| Fixed parameters | EARLIEST param. lambda | $1e-1$ | N.A. (fixed) |
|  | Batch size | $256 \times \gamma$ | N.A. (fixed) |
|  | Epochs | 50 | N.A. (fixed) |
|  | # Tuning trials | 200 | N.A. (fixed) |
|  | # Repeated test trials | 30 | N.A. (fixed) |
| Searched hyperparameters | Learning rate | $[10^{-6}, 10^{-3}] \times \gamma$ | 0.000026 |
|  | Weight decay | $[0.0, 10^{-5}]$ | 0.0005 |
|  | Optimizer | {Adam, RMSprop, Lion} | Lion |
|  | LSTM hidden dim. | [32, 256] | 238 |

Table 24: EARLIEST (lambda=1e-10) on UCF101: parameter space.

|  | Hyperparameter | Space | Optimal value |
|---|---|---|---|
| Fixed parameters | EARLIEST param. lambda | $1e-10$ | N.A. (fixed) |
|  | Batch size | $256 \times \gamma$ | N.A. (fixed) |
|  | Epochs | 50 | N.A. (fixed) |
|  | # Tuning trials | 200 | N.A. (fixed) |
|  | # Repeated test trials | 30 | N.A. (fixed) |
| Searched hyperparameters | Learning rate | $[10^{-6}, 10^{-3}] \times \gamma$ | 0.000758 |
|  | Weight decay | $[0.0, 10^{-5}]$ | 0.0006 |
|  | Optimizer | {Adam, RMSprop, Lion} | RMSprop |
|  | LSTM hidden dim. | [32, 256] | 196 |

Table 25: TCNT (alpha=0.3) on UCF101: parameter space.

|  | Hyperparameter | Space | Optimal value |
|---|---|---|---|
| Fixed parameters | TCNT param. alpha | 0.3 | N.A. (fixed) |
|  | Batch size | $256 \times \gamma$ | N.A. (fixed) |
|  | Epochs | 15 | N.A. (fixed) |
|  | # Tuning trials | 300 | N.A. (fixed) |
|  | # Repeated test trials | 15 | N.A. (fixed) |
| Searched hyperparameters | Learning rate | $[10^{-6}, 10^{-3}] \times \gamma$ | 0.000585 |
|  | Weight decay | $[0.0, 10^{-5}]$ | 0.001 |
|  | Dropout | [0.0, 0.5] | 0.1 |
|  | Optimizer | {Adam, RMSprop, Lion} | Adam |
|  | # Blocks | [1, 3] | 1 |
|  | # Num heads | [2, 4] | 4 |
|  | TCN channels | [256, 1024] | 512 |

Table 26: TCNT (alpha=0.5) on UCF101: parameter space.

|  | Hyperparameter | Space | Optimal value |
|---|---|---|---|
| Fixed parameters | TCNT param. alpha | 0.5 | N.A. (fixed) |
|  | Batch size | $256 \times \gamma$ | N.A. (fixed) |
|  | Epochs | 15 | N.A. (fixed) |
|  | # Tuning trials | 300 | N.A. (fixed) |
|  | # Repeated test trials | 15 | N.A. (fixed) |
| Searched hyperparameters | Learning rate | $[10^{-6}, 10^{-3}] \times \gamma$ | 0.001106 |
|  | Weight decay | $[0.0, 10^{-5}]$ | 0.001 |
|  | Dropout | [0.0, 0.5] | 0.2 |
|  | Optimizer | {Adam, RMSprop, Lion} | Adam |
|  | # Blocks | [1, 3] | 1 |
|  | # Heads | [2, 4] | 4 |
|  | TCN channels | [256, 1024] | 512 |

Table 27: CALIMERA on UCF: parameter space.

|  | Hyperparameter | Space | Optimal value |
|---|---|---|---|
| Fixed parameters | Delay penalty | {0.1, 0.5, 1.0} | N.A. (fixed) |
|  | # Repeated test trials | 5 | N.A. (fixed) |

### K.5 COMPUTING INFRASTRUCTURE

All experiments were carried out using custom Python scripts on NVIDIA GeForce RTX 2080 Ti graphics cards. For mathematical computations, Numpy (Harris et al., 2020) and Scipy (Virtanen et al., 2020) were employed. Machine learning frameworks used include PyTorch 2.0.0 (Paszke et al., 2019) and TensorFlow 2.8.0 (Abadi et al., 2015). Gaussian process regression was performed using stochastic variational inference in GPyTorch (Gardner et al., 2018).

## L    ON HYPERPARAMETER SENSITIVITY OF FIRMBOUND

Our algorithm either requires minimal hyperparameter tuning or can be easily tuned on the training dataset. Below is a list of major hyperparameters for the two approaches:

**Convex Function Learning (CFL)**

- Lambda
- Number of training data for tuning
- Number of training data for fitting
- Tuning trials
- Epochs

**Gaussian Processes (GP)**

- Kernel type
- Number of inducing points
- Batch size
- Learning rate
- Optimizer
- Epochs

The most critical hyperparameter is the lambda parameter (do not confuse with the LLR or the baseline model EARLIEST's hyperparameter) in CFL, which controls the flexibility of the fitting curves. As described in the Sec. 4, we keep the other hyperparameter settings consistent across all datasets, including i.i.d., non-i.i.d., artificial, and real-world. Our experiments show that FIRMBOUND reliably minimizes the average a posteriori risk (AAPR) to delineate the Pareto front.

As an additional experiment for hyperparameter sensitivity, we tested GP approach with varying kernel and number of inducing points using the two-class Gaussian dataset (Fig. 13). The number of inducing points is varied from the default 200 to 50 and 1000, while keeping the original Radial Basis Function (RBF) kernel. Alternatively, the number of inducing points is fixed at 200, and the Matérn kernel is used instead of RBF. Matérn kernel is a generalization of RBF kernel with a parameter $\nu$ controlls its smoothness. As $\nu$ approaches to infinity, Matérn kernel converges to the RBF kernel. In Fig. 13, two values, 1.5 and 2.5, are used as the parameter $\nu$. Cost parameter is $c = \bar{L}/T$. The results are robust against any of the above hyperparameters, while Matérn kernel slightly off from the optimality (also see Fig. 5a and 6a of the main manuscript).

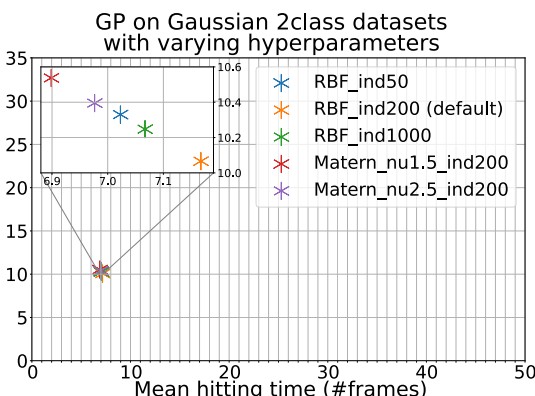

Figure 13: GP's hyperparameter sensitivity test on the two-class Gaussian dataset. The number of inducing points is varied from 50 to 1000, with the Radial Basis Function (RBF) kernel, or fixed at 200 using the Matérn kernel with $\nu$ values of 1.5 and 2.5. Note that $\nu \to \infty$ converges to the RBF kernel. $c = \bar{L}/T$.

# M   AAPR ON BASELINE MODELS

In this section, we demonstrate that ill-calibrated ECTS models can misleadingly exhibit small AAPRs. LSTMms, EARLIEST, and TCN-Transformer models were trained on the two-class Gaussian dataset and evaluated using two performance criteria: AAPR and the SAT curve. As shown in Fig 14a, while LSTMms achieves a lower AAPR than FIRMBOUND and SPRT-TANDEM, which maintain well-calibrated statistics, it is outperformed by them in terms of the SAT curve (i.e., ECTS results). This discrepancy arises from overconfidence, which inflates the statistic beyond the calibrated level. Consequently, AAPR alone does not reliably predict SAT performance when using an ill-calibrated statistic. Notably, FIRMBOUND records the minimal AAPR across all models.

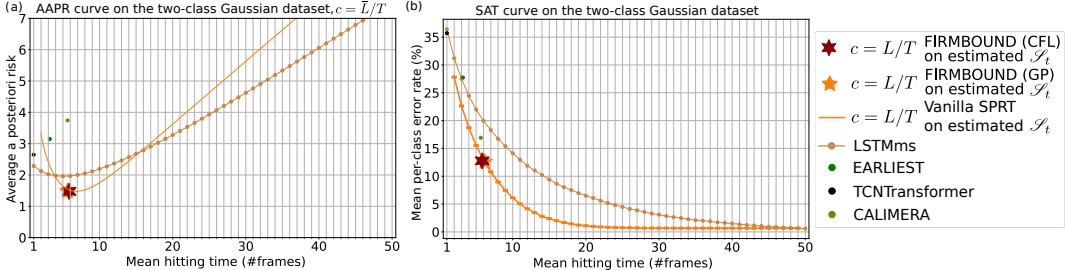

Figure 14: **AAPR and SAT curve with other ECTS algorithms.** ECTS algorithms, LSTMms, EARLIEST, and TCN-Transformer are used to compare the two evaluation criteria on the two-class sequential Gaussian dataset.

## N    PARAMETER SPACE OF $L$ AND $c$

Here, we prove that the possible parameter search space of coefficients $\bar{L}_k = L$ (for all $k \in [K]$) and $c$ of APR is confined, and thus, we only need to consider a ratio of $L$ and $c$. First, given that the continuation risk $\tilde{G}_t(\pi)$ is concave and $\tilde{G}_t(\pi = 0) = \tilde{G}_t(\pi = 1) = c$, the maximum value of the stopping risk $G_t^{\text{st}}(\pi)$ needs to be larger than $c$ in order to have more than one intersection (i.e., threshold):

$$\max\{G_t^{\text{st}}(\pi)\} = L\left(1 - \frac{1}{K}\right)$$
$$> c, \tag{32}$$

where $K$ is the number of classes.

Second, the intersections of the two risk functions $\tilde{G}_t(\pi)$ and $G_t^{\text{st}}(\pi)$ remain invariant under the scaling transformation of $L$ and $c$ by a factor $\alpha \in \mathbb{R}_{\geq 0}$. Specifically, at $t = T$,

$$G_T^{\text{st}}(\pi; \alpha L) = \min_k \left\{\alpha L \left(1 - \pi_k(X_m^{(1,T)})\right)\right\}$$
$$= \alpha \min_k \left\{L \left(1 - \pi_k(X_m^{(1,T)})\right)\right\}$$
$$= \alpha G_T^{\text{st}}(\pi; L, c) \tag{33}$$

thus,

$$G_T^{\min}(\pi; \alpha L, \alpha c) = G_T^{\text{st}}(\pi; \alpha L)$$
$$= \alpha G_T^{\min}(\pi; L, c) \tag{34}$$

then at $t = T - 1$,

$$\tilde{G}_{T-1}(\pi; \alpha L, \alpha c) = \mathbb{E}[G_T^{\min}(\pi; \alpha L, \alpha c) | \pi_k(X_m^{(1,T)})] + \alpha c$$
$$= \mathbb{E}[\alpha G_T^{\min}(\pi; L, c) | \pi_k(X_m^{(1,T)})] + \alpha c$$
$$= \alpha \left(\mathbb{E}[G_T^{\min}(\pi; L, c) | \pi_k(X_m^{(1,T)})] + c\right)$$
$$= \alpha \tilde{G}_{T-1}(\pi; L, c) \tag{35}$$

$$G_{T-1}^{\min}(\pi; \alpha L, \alpha c) = \min\left\{G_{T-1}^{\text{st}}(\pi; \alpha L), \tilde{G}_{T-1}(\pi; \alpha L, \alpha c)\right\}$$
$$= \min\left\{G_{T-1}^{\alpha\text{st}}(\pi; L), \alpha \tilde{G}_{T-1}(\pi; L, c)\right\}$$
$$= \alpha G_{T-1}^{\min}(\pi; L, c) \tag{36}$$

holds true for a scaling factor $\alpha \in \mathbb{R}$. By induction, the above linearity holds true for general $t \leq T$. Given that the threshold is defined as the intersection of $G^{\text{st}}$ and $\tilde{G}$, scaling $L$ and $c$ by a constant $\alpha$ does not alter the threshold.

To equalize the magnitudes of the terms in APR (Eq. 3), we set $c_{\text{def}} = L/T$ as the default in our experiments. This choice consistently balances the speed-accuracy tradeoff and minimizes APR, as elaborated in Sec. 4.

## O    ABLATION STUDY DETAILS

As stated in the main manuscript, vanilla SPRT with static threshold, whether applied to true LLRs or estimated LLRs, is crucial for our ablation studies. Figures 5 and 6 demonstrate that SPRT with a static threshold can lead to either a larger APR or a suboptimal speed-accuracy tradeoff. In this supplementary section on ablation studies, we detail the other two conditions tested: random stopping times and artificial tapering thresholds.

**Random stopping times.**    To establish a chance-level baseline, we randomly generate integers of size $M$ within the range $[1, \ldots, T]$ to use as stopping times. This experiment is repeated five times, with the computed AAPR and SAT points plotted in Fig. 7. These points typically fall in the middle of the figures, delineating the chance levels.

**Artificial tapering thresholds.**    Optimal stopping theory suggests that the optimal threshold computed with backward induction typically descends monotonically as it approaches the finite horizon (Tartakovsky et al., 2014). This insight motivates us to create artificial decision thresholds as economical alternatives. The following power function with $\kappa \in -1.5, 0, 1.5$ is used to generate concave, linear, and convex curves, respectively (Fig. 15):

$$f(t; A, T, \kappa) = A \left(1 - \frac{t}{T}\right)^{e^{\kappa}} \tag{37}$$

Resulting AAPR and SAT are plotted in Fig. 7. The magnitude $A$ is set to $a, 2/a, 0$ where $a = \max_m\{\lambda(X_m^{(1,T)})\}$, whose result corresponding to the three points in Fig. 7.

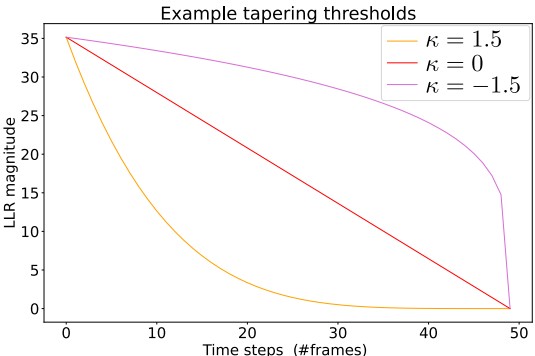

Figure 15: **Tapering thresholds generated with the power function.** According to Eq. 37, concave, linear, and convex tapering threshold are generated with $\kappa \in \{-1.5, 0, 1.5\}$, respectively.

## P    DUMPED OSCILLATING LOG-LIKELIHOOD RATIO FUNCTION

To simulate non-i.i.d., non-monotonic LLR trajectories, we generate binary class LLRs according to
Eq. 38:

$$\Lambda(t) = \gamma\left(1 - \left(1 - \frac{t}{T}\right)^{\exp(\kappa)}\right) + A\exp(-\beta t)\sin(\omega t) + \mathcal{N}(0, \sigma), \tag{38}$$

where $\gamma \in \{-1, 1\}$ corresponds to class targets that the trajectories asymptotically approach. $A$,
$\beta$, and $\omega$ denote the oscillation amplitude, damping coefficient, and angular frequency of the wave,
respectively. $\sigma$ indicates the noise level. Example trajectories are depicted in Fig. 16.

The parameters and their respective prior distributions used in this study are detailed in Tab. 28. The
notation $\mathcal{N}(\mu, \sigma)$ represents a Gaussian (normal) distribution with mean $\mu$ and standard deviation $\sigma$.
The notation $\mathcal{U}(a, b)$ denotes a uniform distribution sampled within the interval $[a, b]$.

Table 28: Parameter space of DOL dataset.

| Parameter | Distribution |
|-----------|--------------|
| A | Gaussian $\mathcal{N}(\mu = 2, \sigma = 2)$ |
| $\beta$ | Uniform $\mathcal{U}(0.02, 0.2)$ |
| $\omega$ | Uniform $\mathcal{U}(-2, 3)$ |
| $\kappa$ | Uniform $\mathcal{U}(-2.5, 0)$ |
| $\sigma$ | Gaussian $\mathcal{N}(\mu = 0.0, \sigma = 1.0)$ |

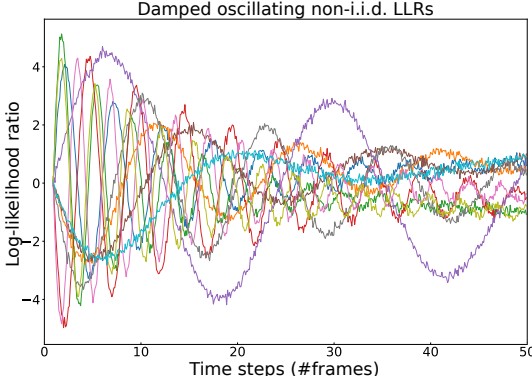

Figure 16: **Tapering thresholds generated with the power function (Eq. 38).** Note that the
trajectories are generated at a higher sampling rate just for visualization purposes. In the experiment,
we sample points at each time step $t \in 1, \ldots, T$.

## Q    SUPPLEMENTARY EXPERIMENT ON THE UCR FORDA DATASET

To test FIRMBOUND's risk minimization capability on continuous signals, we conduct additional experiments on the UCR FordA dataset. FordA is a time series binary classification dataset with 500 samples. Each time series is sliced into non-overlapping segments of 100 time steps. Then each 100-step segment was further processed using a sliding window approach with a window size of 24 and a stride of 4. This resulted in multiple windows per segment:

- The number of windows $W$ generated from each segment is calculated as:

$$W = 1 + \left\lfloor \frac{100 - 24}{4} \right\rfloor = 20$$

- Therefore, each 100-step segment was transformed into 20 windows, each of length 24.

The resulting data are reshaped into a 3-dimensional array with dimensions $(M \times T \times 24)$, where $M$ is the number of original time series (6,600 and 18,005 for training and test dataset, respectively), $T = 20$ is time steps, or the number of windows per segment, and 24 is the feature dimension, or the window length.

The result shows that FIRMBOUND effectively find minima of AAPR to optimize the speed-accuracy tradeoff, as shown in Fig. 17.

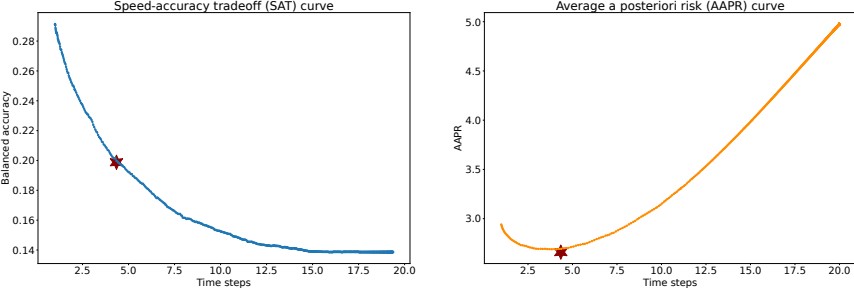

Figure 17: AAPR and SAT curves on UCR FordA dataset. FIRMBOUND effectively find minima of AAPR to optimize the speed-accuracy tradeoff.

## SUPPLEMENTARY REFERENCES

M. Abadi, A. Agarwal, P. Barham, E. Brevdo, Z. Chen, C. Citro, G. S. Corrado, A. Davis, J. Dean, M. Devin, S. Ghemawat, I. Goodfellow, A. Harp, G. Irving, M. Isard, Y. Jia, R. Jozefowicz, L. Kaiser, M. Kudlur, J. Levenberg, D. Mané, R. Monga, S. Moore, D. Murray, C. Olah, M. Schuster, J. Shlens, B. Steiner, I. Sutskever, K. Talwar, P. Tucker, V. Vanhoucke, V. Vasudevan, F. Viégas, O. Vinyals, P. Warden, M. Wattenberg, M. Wicke, Y. Yu, and X. Zheng. TensorFlow: Large-scale machine learning on heterogeneous systems, 2015. Software available from tensorflow.org.

S. Ahmad and A. J. Yu. Active sensing as bayes-optimal sequential decision-making. In *Proceedings of the Twenty-Ninth Conference on Uncertainty in Artificial Intelligence*, UAI'13, pp. 12–21, Arlington, Virginia, USA, 2013. AUAI Press.

P. Armitage. Sequential analysis with more than two alternative hypotheses, and its relation to discriminant function analysis. *Journal of the Royal Statistical Society. Series B (Methodological)*, 12(1):137–144, 1950.

H. Asano. Sequential bayesian experimental designs via reinforcement learning. *arXiv preprint arXiv:2202.07472*, 2022.

C. Atkinson, B. McCane, L. Szymanski, and A. V. Robins. Pseudo-rehearsal: Achieving deep reinforcement learning without catastrophic forgetting. *Neurocomputing*, 428:291–307, 2018.

D. M. Bashtannyk and R. J. Hyndman. Bandwidth selection for kernel conditional density estimation. *Computational Statistics & Data Analysis*, 36:279–298, 2001.

C. W. Baum and V. V. Veeravalli. A sequential procedure for multihypothesis testing. *IEEE Transactions on Information Theory*, 40(6):1994–2007, Nov 1994.

J. Bjorck, C. P. Gomes, and K. Q. Weinberger. Is high variance unavoidable in RL? a case study in continuous control. *ArXiv*, abs/2110.11222, 2021.

T. Blau, E. V. Bonilla, I. Chades, and A. Dezfouli. Optimizing sequential experimental design with deep reinforcement learning. In *International Conference on Machine Learning*, pp. 2107–2128. PMLR, 2022.

Z. Botev, J. Grotowski, and D. Kroese. Kernel density estimation via diffusion. *The Annals of Statistics*, 38, 11 2010.

A. Brockwell and J. J. B. Kadane. A gridding method for bayesian sequential decision problems. *Journal of Computational and Graphical Statistics*, 12:566 – 584, 2003.

J. Q. Candela and C. E. Rasmussen. A unifying view of sparse approximate Gaussian process regression. *J. Mach. Learn. Res.*, 6:1939–1959, 2005.

E. Cetin, P. J. Ball, S. Roberts, and O. Çeliktutan. Stabilizing off-policy deep reinforcement learning from pixels. In *International Conference on Machine Learning*, 2022.

X. Chen, C. Liang, D. Huang, E. Real, K. Wang, H. Pham, X. Dong, T. Luong, C.-J. Hsieh, Y. Lu, and Q. V. Le. Symbolic discovery of optimization algorithms. In A. Oh, T. Naumann, A. Globerson, K. Saenko, M. Hardt, and S. Levine (eds.), *Advances in Neural Information Processing Systems*, volume 36, pp. 49205–49233. Curran Associates, Inc., 2023.

H. Chernoff. Sequential design of experiments. *The Annals of Mathematical Statistics*, 30(3): 755–770, 1959.

A. K. Churchland, R. Kiani, and M. N. Shadlen. Decision-making with multiple alternatives. *Nature Neuroscience*, 11:693–702, 2008.

D. A. Cohn, Z. Ghahramani, and M. I. Jordan. Active learning with statistical models. *ArXiv*, cs.AI/9603104, 1996.

M. P. Deisenroth and J. W. Ng. Distributed Gaussian processes. In *International Conference on Machine Learning*, 2015.

D. K. Dennis, C. Pabbaraju, H. V. Simhadri, and P. Jain. Multiple instance learning for efficient sequential data classification on resource-constrained devices. In *Proceedings of the 32nd International Conference on Neural Information Processing Systems*, NIPS'18, pp. 10976–10987, Red Hook, NY, USA, 2018. Curran Associates Inc.

K. Doya. Modulators of decision making. *Nat. Neurosci.*, 11(4):410–416, Apr 2008.

V. P. Dragalin, A. G. Tartakovsky, and V. V. Veeravalli. Multihypothesis sequential probability ratio tests. i. asymptotic optimality. *IEEE Transactions on Information Theory*, 45(7):2448–2461, November 1999.

V. Dragalin. Asymptotic solution of a problem of detecting a signal from $k$ channels. *Russian Mathematical Surveys*, 42(3):213, 1987.

V. Dragalin and A. Novikov. Adaptive sequential tests for composite hypotheses. *Survey of Applied and Industrial Mathematics*, 6:387–398, 1999.

J. Drugowitsch, R. Moreno-Bote, A. K. Churchland, M. N. Shadlen, and A. Pouget. The cost of accumulating evidence in perceptual decision making. *The Journal of Neuroscience*, 32:3612 – 3628, 2012.

T. Duan, J. Zhao, S. Zhang, J. Tao, and P. Wang. Representation learning of tangled key-value sequence data for early classification. In *2024 40th IEEE International Conference on Data Engineering*, 2024.

T. S. Ferguson. Optimal stopping and applications, 2006.

P. Frazier and A. J. Yu. Sequential hypothesis testing under stochastic deadlines. In J. Platt, D. Koller, Y. Singer, and S. Roweis (eds.), *Advances in Neural Information Processing Systems*, volume 20. Curran Associates, Inc., 2007.

J. P. Gallivan, C. S. Chapman, D. M. Wolpert, and J. R. Flanagan. Decision-making in sensorimotor control. *Nat. Rev. Neurosci.*, 19(9):519–534, 09 2018.

J. I. Gold and M. N. Shadlen. The neural basis of decision making. *Annu. Rev. Neurosci.*, 30:535–574, 2007.

P. Goyal, P. Dollár, R. B. Girshick, P. Noordhuis, L. Wesolowski, A. Kyrola, A. Tulloch, Y. Jia, and K. He. Accurate, large minibatch SGD: training imagenet in 1 hour. *CoRR*, abs/1706.02677, 2017.

A. Graves. Generating sequences with recurrent neural networks. *arXiv preprint arXiv:1308.0850*, 2013.

M. U. Gutmann and A. Hyvärinen. Noise-contrastive estimation of unnormalized statistical models, with applications to natural image statistics. *The journal of machine learning research*, 13(1): 307–361, 2012.

T. D. Hanks, R. Kiani, and M. N. Shadlen. A neural mechanism of speed-accuracy tradeoff in macaque area lip. *eLife*, 3, 2014.

C. R. Harris, K. J. Millman, S. J. van der Walt, R. Gommers, P. Virtanen, D. Cournapeau, E. Wieser, J. Taylor, S. Berg, N. J. Smith, R. Kern, M. Picus, S. Hoyer, M. H. van Kerkwijk, M. Brett, A. Haldane, J. F. Del Río, M. Wiebe, P. Peterson, P. Gérard-Marchant, K. Sheppard, T. Reddy, W. Weckesser, H. Abbasi, C. Gohlke, and T. E. Oliphant. Array programming with NumPy. *Nature*, 585(7825):357–362, 09 2020.

T. Hartvigsen, W. Gerych, J. Thadajarassiri, X. Kong, and E. A. Rundensteiner. Stop&hop: Early classification of irregular time series. In *CIKM*, 2021.

J. Hensman, A. G. de G. Matthews, and Z. Ghahramani. Scalable variational Gaussian process classification. In *International Conference on Artificial Intelligence and Statistics*, 2014.

S. Hochreiter and J. Schmidhuber. Long short-term memory. *Neural Comput.*, 9(8):1735–1780, 1997.

N. Houlsby, F. Huszár, Z. Ghahramani, and M. Lengyel. Bayesian active learning for classification and preference learning. *ArXiv*, abs/1112.5745, 2011.

A. Irle and N. Schmitz. On the optimality of the sprt for processes with continuous time parameter. *Statistics: A Journal of Theoretical and Applied Statistics*, 15(1):91–104, 1984.

H. Ismail Fawaz, G. Forestier, J. Weber, L. Idoumghar, and P.-A. Muller. Deep learning for time series classification: a review. *Data Mining and Knowledge Discovery*, 33(4):917–963, July 2019. ISSN 1573-756X.

D. Jarrett and M. van der Schaar. Inverse active sensing: Modeling and understanding timely decision-making. *ArXiv*, abs/2006.14141, 2020.

R. I. Jennrich. Asymptotic properties of non-linear least squares estimators. *Ann. Math. Statist.*, 40 (2):633–643, 04 1969.

J. B. Kadane and P. K. Vlachos. Hybrid methods for calculating optimal few-stage sequential strategies: Data monitoring for a clinical trial. *Statistics and Computing*, 12:147–152, 2002.

J. Kiefer and J. Sacks. Asymptotically optimum sequential inference and design. *The Annals of Mathematical Statistics*, pp. 705–750, 1963.

D. P. Kingma and J. Ba. Adam: A method for stochastic optimization. *arXiv preprint arXiv:1412.6980*, 2014.

S. Kira, T. Yang, and M. N. Shadlen. A neural implementation of wald's sequential probability rato test. *Neuron*, 85(4):861–873, February 2015.

S. Kira, A. Zylberberg, and M. N. Shadlen. Incorporation of a cost of deliberation time in perceptual decision making. *bioRxiv*, 2024.

S. Kleinegesse, C. Drovandi, and M. Gutmann. Sequential bayesian experimental design for implicit models via mutual information. *Bayesian Analysis*, -1, 07 2020.

D. Kroese, T. Taimre, and Z. Botev. *Handbook of Monte Carlo Methods*. Wiley Series in Probability and Statistics. Wiley, 2011. ISBN 9780470177938.

A. Kumar, A. Gupta, and S. Levine. DisCor: Corrective feedback in reinforcement learning via distribution correction. In *Proceedings of the 33rd International Conference on Neural Information Processing Systems*, 2020.

T. L. Lai. Asymptotic optimality of invariant sequential probability ratio tests. *The Annals of Statistics*, pp. 318–333, 1981.

K. W. Latimer, J. L. Yates, M. L. Meister, A. C. Huk, and J. W. Pillow. Single-trial spike trains in parietal cortex reveal discrete steps during decision-making. *Science*, 349(6244):184–187, Jul 2015.

D. D. Lewis and W. A. Gale. A sequential algorithm for training text classifiers. In *Annual International ACM SIGIR Conference on Research and Development in Information Retrieval*, 1994.

G. Lorden. Nearly-optimal sequential tests for finitely many parameter values. *Annals of Statistics*, 5: 1–21, 01 1977.

J. Lv, Y. Chu, J. Hu, P. Li, and X. Hu. Second-order confidence network for early classification of time series. *ACM Transactions on Intelligent Systems and Technology*, 2023.

C. Martinez, E. Ramasso, G. Perrin, and M. Rombaut. Adaptive early classification of temporal sequences using deep reinforcement learning. *Knowledge-Based Systems*, 190:105290, 2020. ISSN 0950-7051.

T. Miyagawa and A. F. Ebihara. The power of log-sum-exp: Sequential density ratio matrix estimation for speed-accuracy optimization. In *Proceedings of the 38th International Conference on Machine Learning*, pp. 7792–7804, 2021.

U. Mori, A. Mendiburu, S. Dasgupta, and J. A. Lozano. Early classification of time series by simultaneously optimizing the accuracy and earliness. *IEEE Transactions on Neural Networks and Learning Systems*, 29(10):4569–4578, Oct 2018. ISSN 2162-237X.

U. Mori, A. Mendiburu, S. Dasgupta, and J. A. Lozano. Early classification of time series from a cost minimization point of view. In *Proceedings of the NIPS Time Series Workshop*, 2015.

P. Müller, D. A. Berry, A. P. Grieve, M. K. Smith, and M. Krams. Simulation-based sequential bayesian design. *Journal of Statistical Planning and Inference*, 137:3140–3150, 2007.

M. Naghshvar and T. Javidi. Active sequential hypothesis testing. *The Annals of Statistics*, 41(6): 2703–2738, 2013. ISSN 00905364, 21688966.

J. Najemnik and W. S. Geisler. Optimal eye movement strategies in visual search. *Nature*, 434: 387–391, 2005.

W. Newey and D. McFadden. Large sample estimation and hypothesis testing. In R. F. Engle and D. McFadden (eds.), *Handbook of Econometrics*, volume 4, chapter 36, pp. 2111–2245. Elsevier, 1 edition, 1986.

E. Nikishin, P. Izmailov, B. Athiwaratkun, D. Podoprikhin, T. Garipov, P. Shvechikov, D. Vetrov, and A. G. Wilson. Improving stability in deep reinforcement learning with weight averaging. In *Uncertainty in artificial intelligence workshop on uncertainty in Deep learning*, 2018.

G. Okazawa, C. E. Hatch, A. Mancoo, C. K. Machens, and R. Kiani. The geometry of the representation of decision variable and stimulus difficulty in the parietal cortex. *bioRxiv*, 2021.

A. Paszke, S. Gross, F. Massa, A. Lerer, J. Bradbury, G. Chanan, T. Killeen, Z. Lin, N. Gimelshein, L. Antiga, A. Desmaison, A. Kopf, E. Yang, Z. DeVito, M. Raison, A. Tejani, S. Chilamkurthy, B. Steiner, L. Fang, J. Bai, and S. Chintala. Pytorch: An imperative style, high-performance deep learning library. In *Advances in Neural Information Processing Systems 32*, pp. 8024–8035. Curran Associates, Inc., 2019.

E. Paulson. A sequential decision procedure for choosing one of $k$ hypotheses concerning the unknown mean of a normal distribution. *The Annals of Mathematical Statistics*, pp. 549–554, 1963.

I. V. Pavlov. Sequential procedure of testing composite hypotheses with applications to the kiefer–weiss problem. *Theory of Probability & Its Applications*, 35(2):280–292, 1991.

I. Pavlov. Sequential decision rule for the case of many complex hypotheses. *ENG. CYBER.*, (6): 19–22, 1984.

B. Rhodes, K. Xu, and M. U. Gutmann. Telescoping density-ratio estimation. In *NeurIPS*, volume 33, pp. 4905–4916, 2020.

C. Robert and G. Casella. *Monte Carlo statistical methods*. Springer Verlag, 2004.

J. D. Roitman and M. N. Shadlen. Response of neurons in the lateral intraparietal area during a combined visual discrimination reaction time task. *J. Neurosci.*, 22(21):9475–9489, Nov 2002.

M. Rosenblatt. Conditional probability density and regression estimators. In P. R. Krishnaiah (ed.), *Multivariate analysis, II*, pp. 25–31. Academic Press, New York, 1969. (Dayton, OH, 17–22 June 1968). MR:254987.

D. Rossell, P. Müller, and G. L. Rosner. Screening designs for drug development. *Biostatistics*, 8 3: 595–608, 2007.

D. Scott. *Multivariate Density Estimation: Theory, Practice, and Visualization*. A Wiley-interscience publication. Wiley, 1992. ISBN 9780471547709.

O. Sener and S. Savarese. Active learning for convolutional neural networks: A core-set approach. *arXiv: Machine Learning*, 2017.

H. S. Seung, M. Opper, and H. Sompolinsky. Query by committee. In *Annual Conference Computational Learning Theory*, 1992.

A. Siahkamari, D. A. E. Acar, C. Liao, K. L. Geyer, V. Saligrama, and B. Kulis. Faster algorithms for learning convex functions. In K. Chaudhuri, S. Jegelka, L. Song, C. Szepesvari, G. Niu, and S. Sabato (eds.), *Proceedings of the 39th International Conference on Machine Learning*, volume 162 of *Proceedings of Machine Learning Research*, pp. 20176–20194. PMLR, 17–23 Jul 2022.

B. Silverman. *Density Estimation for Statistics and Data Analysis*. Chapman & Hall/CRC Monographs on Statistics & Applied Probability. Taylor & Francis, 1986. ISBN 9780412246203.

G. Simons. Lower bounds for average sample number of sequential multihypothesis tests. *The Annals of Mathematical Statistics*, pp. 1343–1364, 1967.

M. Sobel and A. Wald. A sequential decision procedure for choosing one of three hypotheses concerning the unknown mean of a normal distribution. *Ann. Math. Statist.*, 20(4):502–522, 12 1949.

J. Sochman and J. Matas. Waldboost - learning for time constrained sequential detection. In *2005 IEEE Computer Society Conference on Computer Vision and Pattern Recognition (CVPR'05)*, volume 2, pp. 150–156 vol. 2, June 2005.

R. Sullivan, J. K. Terry, B. Black, and J. P. Dickerson. Cliff diving: Exploring reward surfaces in reinforcement learning environments. *ArXiv*, abs/2205.07015, 2022.

C. Sun, H. Li, M. Song, and linda Qiao. A ranking-based cross-entropy loss for early classification of time series. *IEEE transactions on neural networks and learning systems*, PP, 2023.

T. Suzuki, H. Kataoka, Y. Aoki, and Y. Satoh. Anticipating traffic accidents with adaptive loss and large-scale incident db. In *Proceedings of the IEEE Conference on Computer Vision and Pattern Recognition*, pp. 3521–3529, 2018.

A. Tartakovsky. Asymptotic optimality of certain multihypothesis sequential tests: Non-i.i.d. case. *Stat. Inference Stoch. Process.*, 1:265–295, 1998.

A. Tartakovsky, I. Nikiforov, and M. Basseville. *Sequential Analysis: Hypothesis Testing and Changepoint Detection*. Chapman & Hall/CRC, 1st edition, 2014.

M. Tec, Y. Duan, and P. Müller. A comparative tutorial of bayesian sequential design and reinforcement learning. *The American Statistician*, 77(2):223–233, 2023.

D. Teng and E. Ertin. Wald-kernel: A method for learning sequential detectors. In *2016 IEEE Statistical Signal Processing Workshop (SSP)*, pp. 1–5, June 2016.

P. Virtanen, R. Gommers, T. E. Oliphant, M. Haberland, T. Reddy, D. Cournapeau, E. Burovski, P. Peterson, W. Weckesser, J. Bright, S. J. van der Walt, M. Brett, J. Wilson, K. Jarrod Millman, N. Mayorov, A. R. J. Nelson, E. Jones, R. Kern, E. Larson, C. Carey, İ. Polat, Y. Feng, E. W. Moore, J. Vand erPlas, D. Laxalde, J. Perktold, R. Cimrman, I. Henriksen, E. A. Quintero, C. R. Harris, A. M. Archibald, A. H. Ribeiro, F. Pedregosa, P. van Mulbregt, and S. . . . Contributors. SciPy 1.0: Fundamental Algorithms for Scientific Computing in Python. *Nature Methods*, 17: 261–272, 2020.

A. Wald and J. Wolfowitz. Optimum character of the sequential probability ratio test. *Ann. Math. Statist.*, 19(3):326–339, 09 1948.

A. Wald and J. Wolfowitz. Bayes solutions of sequential decision problems. *The Annals of Mathematical Statistics*, 21(1):82–99, 1950. ISSN 00034851.

B. Wang and X. Wang. Bandwidth selection for weighted kernel density estimation. *arXiv*, 2007.

Y. Wang, Q. Zhang, L. Ying, and C. Zhou. Deep reinforcement learning for early diagnosis of lung cancer. In M. J. Wooldridge, J. G. Dy, and S. Natarajan (eds.), *AAAI*, pp. 22410–22419. AAAI Press, 2024.

Z. Wang and D. W. Scott. Nonparametric density estimation for high-dimensional data—algorithms and applications. *Wiley Interdisciplinary Reviews: Computational Statistics*, 11, 2019.

J. Weng, X. Jiang, W.-L. Zheng, and J. Yuan. Early action recognition with category exclusion using policy-based reinforcement learning. *IEEE TCSVT*, 30:4626–4638, 2020.

A. G. Wilson and H. Nickisch. Kernel interpolation for scalable structured Gaussian processes (KISS-GP). In *International Conference on Machine Learning*, 2015.

Z. Xing, J. Pei, and P. S. Yu. Early classification on time series. *Knowledge and Information Systems*, 31(1):105–127, April 2012.

P.-E. Zafar, Y. Achenchabe, A. Bondu, A. Cornuéjols, and V. Lemaire. Early Classification of Time Series: Cost-based multiclass Algorithms. In *2021 IEEE 8th International Conference on Data Science and Advanced Analytics (DSAA)*, pp. 1–10, Porto, Portugal, October 2021. IEEE.

