# OpenReview forum: "Learning the Optimal Stopping for Early Classification within Finite Horizons via Sequential Probability Ratio Test"
_ICLR.cc/2025/Conference — ICLR 2025 Poster_

### Official Review · Reviewer_HH7v · 2024-11-03

**Soundness:** 3
**Presentation:** 2
**Contribution:** 3
**Rating:** 8
**Confidence:** 2

**Summary:**

The paper studies the problem of optimal stopping time for early classification of time-series. The problem is to identify the best time to stop and identify the correct class of a time-series as the time-progresses. It is known that the optimal method for this problem with a finite time-horizon is the backward induction equation that is computationally and sample complexity wise intractable. The paper makes two contribution (i) reducing the complexity of the backward induction method by observing that conditional expectation in backwards induction is a convex function learning under noise problem which has standard techniques. (ii) Further assuming the conditional expectation sequence follows a gaussian process thus reducing it to GP regression learning. They show their method achieves close to optimal speed accuracy trade-off for the problem against four baselines on synthetic and real world datasets.

**Strengths:**

1. The paper tackles an interesting and technically hard problem.

2. The ideas and the algorithm seems sound and practically implementable.

3. The empirical experiments are thorough and has real world datasets.

4. The paper explains the ideas in a nice sequence even though some parts are hard to parse.

**Weaknesses:**

1. Some parts of the paper are too dense and hard to parse. For instance can the authors expand on the intuition behind the various terms of the backward induction equation in Theorem 2.1.

2. Why does FIRMBOUND not improve upon vanilla SPRT in terms of performance gains on the real datasets?

**Questions:**

Asked above.

---

> ### Author Response · Authors · 2024-11-13
>
> Thank you for your thoughtful and positive feedback on our paper. We greatly appreciate your time and effort in reviewing our work. Please find our detailed responses to your comments below.
>
> # Intuition Behind Theorem 2.1 (Backward Induction Equation)
> In Appendix F, we provided a brief intuitive explanation under the section titled “F.1 Intuitive Understanding of FIRMBOUND.” Below, we expand on this explanation to provide more detail. In the updated manuscript, we have incorporated this expanded explanation and added a reference to Appendix F to help readers build a deeper understanding.
>
> **Theorem 2.1 can be interpreted as a recursive decision-making process that minimizes Bayes risk at each time step**. The Sequential Probability Ratio Test (SPRT) at each step must either (i) continue sampling to refine the sufficient statistic, or (ii) make a final classification decision with the current sufficient statistic. Each choice—(i) continuing or (ii) stopping—incurs a form of risk: the continuation risk, $\tilde{G}_t$ (Eq. (5)), and the stopping risk, $G^{\mathrm{st}}_t$ (Eq. (6)). At each time step, the lower of these two values defines the minimum risk, $G^{\mathrm{min}}_t$ (Eq. (7)), up to the classification deadline or finite horizon.
>
> The key computational challenge lies in estimating Eq. (5), which calculates the expected future minimum risk, conditioned on the current sufficient statistic $\mathscr{S_t}$. Our approach overcomes this by formulating it as a noisy convex regression problem. The recurrence is computed backward from the terminal condition at $t = T$, where only the stopping risk applies. Readers familiar with reinforcement learning may find it helpful to view the backward induction as analogous to the Bellman equation, where future risk minimization is the objective.
>
> # Why FIRMBOUND Shows Limited Performance Gains Over Vanilla SPRT on Real Datasets?
> To clarify, we do observe performance improvements, as measured by mean macro-averaged recall, on some real-world datasets (e.g., SiW and UCF101), as shown in Fig. 6d and 6f (insets). However, as noted, these gains are modest and may not reach statistical significance.
>
> **The limited effect size arises due to the relatively monotonic trajectory of the sufficient statistic in these datasets**, as we briefly discussed in Appendix F.6. As illustrated in Fig. 1, the red trajectory demonstrates a monotonic increase, reaching either the optimal threshold (yellow curve) or the static threshold (gray dashed line) without fluctuation. In such cases, lowering the static threshold to achieve the same hitting time as the optimal threshold does not lead to classification errors (gray dash-dot line). This results in “sliding” along the Pareto front of the vanilla SPRT without advancing to a new Pareto-optimal front (i.e., no improvement in macro-averaged recall). Conversely, the Damped Oscillating LLR (DOL) dataset (Fig. 6c) exhibits greater fluctuations, leading to more pronounced performance gains with FIRMBOUND. The blue trajectory in Fig. 1 highlights the impact of fluctuating sufficient statistics on error rates.
>
> **This minimal performance gain does not diminish FIRMBOUND’s practical value**. The real-world datasets examined (SiW, HMDB51, UCF101, and FordA) involve relatively small domain gaps and fewer fluctuations, producing stable, monotonic trajectories that limit opportunities for improvement over static thresholds. However, in more adversarial real-world scenarios, such as those with dim or variable lighting, we would expect the trajectory to fluctuate similarly to the DOL dataset, where FIRMBOUND shows robust performance gains. Furthermore, as noted in Section 4, FIRMBOUND consistently reduces the variance of hitting times across all datasets. Along with its capability to handle i.i.d., non-i.i.d., and multiclass data, this variance reduction demonstrates FIRMBOUND’s potential for reliable decision making. In future work, we aim to explore FIRMBOUND’s performance under more dynamically adversarial conditions, which we anticipate will reveal greater gains similar to those seen in our DOL experiments. We updated Appendix F.5 to include the future research direction under dynamic environments.

---

> > ### Author Response · Authors · 2024-11-22
> >
> > Dear Reviewer `HH7v`,
> >
> > With the discussion period nearing its end, I wanted to kindly ask you to engage the discussion. We have carefully addressed all of your comments and concerns. We believe that our responses clarify the key points but if there are additional questions or points requiring clarification, we would be happy to address them promptly. We kindly ask that you consider our responses in your reevaluation of the manuscript, and we sincerely appreciate your time and effort in reviewing our work.
> >
> > Best regards,
> > Authors

---

### Official Review · Reviewer_81so · 2024-11-03

**Soundness:** 4
**Presentation:** 4
**Contribution:** 3
**Rating:** 6
**Confidence:** 3

**Summary:**

The paper introduces FIRMBOUND, a framework for early classification of time series tasks that builds on the Sequential Probability Ratio Test (SPRT) to address computational challenges in estimating the solution to backward induction from training data. FIRMBOUND uses density ratio estimation and convex function learning to provide consistent estimators for the sufficient statistic and conditional expectation, enabling to minimize Bayes risk for optimality. The paper then presents a faster alternative using Gaussian process regression, which reduces training time while retaining low deployment overhead, albeit with potential compromise in statistical consistency. Experiments demonstrate that FIRMBOUND achieves optimalities in the sense of Bayes risk and speed-accuracy tradeoff.

**Strengths:**

1. Convex function learning could give a consistent estimator of the backward induction functions. The Gaussian process regression gives a computationally more efficient approach for  CFL.
2. The minimizer of the Log-Sum-Exp Loss gives a consistent estimation of the log density ratio.
3. Combining the two approaches, FIRMBOUND with CFL is consistent and Bayes optimal for ECTS.
4. The paper performed experiments across synthetic data and real world data, demonstrating that FIRMBOUND achieves optimalities in the sense of Bayes risk and speed-accuracy tradeoff.

In general, this paper provides a method that solves an important task and works well in practice.

**Weaknesses:**

1. The procedure requires a two-step estimation: first estimate the sufficient statistics (log density ratio) with finite samples; then, based on the estimated sufficient statistics, estimate the backward induction function. The error made in the first step could make the second step much harder. Figure 5 seem to show that estimated LLRs indeed perform worse than true LLRs.

**Questions:**

NA

---

> ### Author Response · Authors · 2024-11-14
>
> Thank you for your thoughtful and positive feedback on our paper. We greatly appreciate your time and effort in reviewing our work. Please find our detailed responses to your comments below.
>
> # Two-step Estimation and Error Propagation
> We acknowledge that a two-step estimation can indeed lead to error propagation; **this is precisely why FIRMBOUND was designed with theoretical consistency in mind**. FIRMBOUND achieves consistent estimation for both the conditional expectation in backward induction and the log-likelihood ratio. We provide theoretical proof of this "double consistency" and our extensive experiments demonstrate that FIRMBOUND achieves Pareto-optimal performance and reduces hitting-time variance across diverse datasets, from i.i.d. binary to non-i.i.d. multiclass data.
>
> In comparison, other two-step estimation algorithms, such as CALIMERA, fall short. Like FIRMBOUND, CALIMERA involves two-step estimation, training a classifier to set the prediction score, followed by training a stopping module. While CALIMERA predicts future risk to determine hitting time [1], its performance shows higher classification error rates and greater hyperparameter sensitivity, as highlighted in Fig. 6.
>
> One-step, end-to-end algorithms, including LSTMms, EARLIEST, and TCN-Transformer, are also outperformed by FIRMBOUND. While these methods are simpler to train and avoid error propagation by design, they lack theoretical guarantees for estimation accuracy. Furthermore, their simultaneous training of the classifier and stopping module is computationally intensive, which may contribute to higher error rates and greater hyperparameter sensitivity, as discussed in our paper.
>
> **Overall, our results support the adoption of a two-step algorithm, as justified by both theoretical guarantees and empirical findings**. By ensuring double consistency, FIRMBOUND minimizes estimation errors and achieves robust performance across diverse scenarios. We added the above discussion to Appendix F.6.
>
> [1] Bilski and Jastrzebska. CALIMERA: A new early time series classification method. Information Processing & Management, 2023.
>
> # FIRMBOUND on Finite Samples
> Thank you for raising this important point. While FIRMBOUND ensures statistical consistency, its practicality could be questioned if performance degrades significantly with a reduction in dataset size. To address this concern, we conducted additional experiments with reduced dataset sizes. Please see the newly added section, "Performance under Small Datasets," in Appendix F. Using the three-class Gaussian dataset, we trained FIRMBOUND with datasets up to two orders of magnitude smaller than the original size, while using the same test dataset. Specifically, dataset sizes of $M = 600$ and $M = 6000$ were compared to the original size of $M = 60000$. Hyperparameter settings were kept fixed, and experiments were repeated five times to compute error bars.
>
> The results show that even with a dataset one order of magnitude smaller ($M = 6000$), **FIRMBOUND demonstrates almost negligible performance differences compared to the original dataset** in terms of mean hitting time, AAPR, and mean per-class error rate. However, with the smallest dataset ($M = 600$), AAPR and mean per-class error rates increase, indicating that the dataset is too small to accurately estimate the sufficient statistics. Despite this, the mean hitting time remains close to the original, highlighting FIRMBOUND's stability and its ability to make best-effort decisions even under highly limited data conditions.

---

> ### Author Response · Authors · 2024-11-16
>
> # Reduced Performance on Estimated LLRs Compared to the True LLRs (Fig. 5)
> We agree that Fig. 5a (for the two-class Gaussian dataset) shows a discrepancy in the minimal averaged posterior risk (AAPR) locations between true and estimated LLRs. Interestingly, this trend is negligible in Fig. 5b (for the three-class Gaussian dataset), where AAPR locations for true and estimated LLRs align more closely. One possible explanation is the density chasm problem, a known issue specific to density ratio estimation on "easy" problems, which increases the absolute value of the density ratio and could contribute to these errors [2]. A countermeasure to the density chasm problem, telescoping density ratio estimation, was proposed in [2], and employing this approach may help mitigate the error on simple datasets.
>
> These issues are likely specific to simple datasets and are less prevalent in complex real-world datasets with arbitrary class counts. It is important to note that while statistical consistency guarantees minimization of estimation error, it does not address approximation or optimization errors, which may be the main contributing factors here. Although we defer a detailed analysis and implementation of potential countermeasures to future work, we have added a section in Appendix F to discuss this issue as an open area for further investigation. We appreciate you bringing this to our attention.
>
> [2] Rhodes+, Telescoping density-ratio estimation. NeurIPS, 2020.

---

> > ### Author Response · Authors · 2024-11-22
> >
> > Dear Reviewer `81so`,
> >
> > With the discussion period nearing its end, I wanted to kindly ask you to engage the discussion. We have carefully addressed all of your comments and concerns. We believe that our responses clarify the key points but if there are additional questions or points requiring clarification, we would be happy to address them promptly. We kindly ask that you consider our responses in your reevaluation of the manuscript, and we sincerely appreciate your time and effort in reviewing our work.
> >
> > Best regards,
> > Authors

---

### Official Review · Reviewer_2NLt · 2024-11-03

**Soundness:** 3
**Presentation:** 3
**Contribution:** 3
**Rating:** 6
**Confidence:** 2

**Summary:**

This paper presents a framework to enhance early classification in time-sensitive applications by adapting the Sequential Probability Ratio Test (SPRT) for scenarios with finite horizons. The contribution is motivated by applications where decisions must be made within limited timeframes. The paper draws from ideas in Convex Function Learning (CFL), Density Ratio Estimation (DRE), and Gaussian Process (GP) Regression—and develops a framework for achieving optimal stopping in early classification tasks. Notably, CFL is used to estimate continuation risk efficiently, providing theoretically consistent estimations; GP regression is used as a faster, though slightly less statistically consistent, alternative; and integrates Density Ratio Estimation (DRE) algorithm for accurate log-likelihood ratio (LLR) estimates. The authors focus on Bayes-optimal solutions for early classification, which are supported by consistency results for CFL and DRE estimations.

**Strengths:**

CFL is chosen specifically for its ability to offer statistically consistent estimates of the continuation risk function, which is concave in nature. This consistency is achieved by leveraging convex regression to estimate the conditional expectations that are key to solving the backward induction. The authors support this choice by citing prior work.

The DRE algorithm, specifically the SPRT-TANDEM variant, enhances LLR estimation accuracy by directly estimating the density ratios, avoiding the need to estimate individual probability densities. This approach reduces degrees of freedom, resulting in more stable and accurate LLR estimates across various data distributions (both i.i.d. and non-i.i.d.). The resulting approach handles both synthetic and real-world datasets more flexibly.

GP regression is presented as a faster but less consistent alternative, primarily for cases where computational speed outweighs the need for the strict statistical consistency.

**Weaknesses:**

1. The paper would benefit from more detailed empirical comparisons with traditional methods to demonstrate performance gains in practice.

2. Assessment/comparison of DRE’s advantage with respect to other estimation remains a question.

3. More discussion is needed on scenarios where GP regression’s speed justifies its consistency loss. Overall leaving tradeoff in consistency with GP regression is underexplored.

4. The overall approach becomes multi-component approach – adding implementation complexity and training across datasets. Specialized tuning (e.g., ADMM for CFL) may raise setup complexity.
Some potentially related work (in terms of ideas/concepts) include:

“Analyzing the discrepancy principle for kernelized spectral filter learning algorithms,” Alain Celisse and Martin Wahl, (2020) J. Mach. Learn. Res.

“WALD-Kernel: A method for learning sequential detectors,” Diyan Teng and Emre Ertin (2016) IEEE Statistical Signal Processing Workshop (SSP)

“Bayesian Optimization Meets Bayesian Optimal Stopping,” Zhongxiang Dai and Haibin Yu and K. H. Low and Patrick Jaillet (2019). Proposes unifying GP-UCB with Bayesian Optimal Stopping (BO-BOS). Focuses on hyperparameter tuning efficiency rather than early classification tasks.

**Questions:**

While the paper contributes to the question of how Convex Function Learning (CFL) be effectively integrated with Density Ratio Estimation (DRE and Gaussian Process Regression (GPR) for synergistic benefits in early classification tasks, what would be authors’ recommendation on evaluation of their combined advantage and limitations in practical settings?  This overall question relates to items 1-4 in the “weaknesses” section above.

Under what conditions does the proposed multi-component framework maintain statistical consistency in small sample settings typically found in real-time applications? Although some statistical properties are explored, there is still uncertainty about how these methods perform and remain consistent under stringent data constraints.

Finally, what practical obstacles exist in deploying these methods at scale in time-sensitive applications?

---

> ### Author Response · Authors · 2024-11-13
>
> Thank you for your thoughtful and positive feedback on our paper. We greatly appreciate your time and effort in reviewing our work. Please find our detailed responses to your comments below.
>
> # Synergistic Benefits of FIRMBOUND’s Components
> The design of FIRMBOUND’s components was carefully chosen to ensure consistent estimation of both the conditional expectation in backward induction and the log-likelihood ratio (LLR). First, the Sequential Probability Ratio Test (SPRT) is Bayes optimal in minimizing the expected sample size (hitting time) for a given error rate. **Since SPRT uses the log-density ratio (or log-likelihood ratio) as its sufficient statistic, Density Ratio Estimation (DRE) is an indispensable component** (addressing Weakness 2). Any alternative scoring algorithm would compromise SPRT’s optimality. Second, Convex Function Learning (CFL) is essential for enabling online evaluation of the backward induction equation (Theorem 2.1). As detailed in Appendix D, “Computational Complexity of FIRMBOUND and Sampling Method,” conventional methods for estimating the conditional expectation, such as Monte Carlo integration, are computationally infeasible in high dimensions due to the curse of dimensionality. **CFL provides a practical and efficient solution, making it feasible to estimate the conditional expectation directly and in real-time**.
>
> This “double consistency”—achieved through the integration of DRE and CFL—is central to FIRMBOUND’s ability to function as a deployable, Bayes-optimal, real-world, online early classification of time series (ECTS) algorithm.
>
> # Justification to FIRMBOUND’s Multi-component Framework
> **The doubly consistent estimation required by FIRMBOUND necessitates a multi-component framework**. While this design may introduce additional complexity and the potential for error propagation across components, FIRMBOUND guarantees the minimization of estimation errors, particularly in large datasets.
>
> **Consistent estimation itself represents a significant advancement in early classification of time series (ECTS)**. Most existing ECTS methods rely on empirical heuristics, whether they follow a two-component approach (e.g., CALIMERA) or a one-component, end-to-end framework (e.g., LSTMms, TCN-Transformer, EARLIEST). Our experiments demonstrate that these heuristic-based approaches are consistently outperformed by FIRMBOUND, emphasizing the practical benefits derived from our theoretically grounded, multi-component framework (addressing Weakness 4).
>
> Therefore, we do not believe further empirical comparisons are necessary, as they are not backed by theoretical guarantees and would not alter the paper’s narrative or conclusions (addressing Weakness 1). However, if you have specific methods in mind, we would welcome suggestions for consideration.
>
> We added the above discussion to Appendix F.6 "Frequently Asked Questions."
>
> # Performance Under Stringent Data Constraints
> Thank you for suggesting this important experiment. While FIRMBOUND ensures statistical consistency, its practicality could be questioned if performance degrades significantly with reduced dataset sizes. To address this concern, we updated Appendix F with a new section, “Performance under Small Datasets.” Using the three-class Gaussian dataset, we trained FIRMBOUND with datasets up to two orders of magnitude smaller than the original size while keeping the same test dataset. Specifically, dataset sizes of $M = 600$ and $M = 6000$ were compared to the original size of $M = 60000$. Hyperparameter settings were kept fixed, and experiments were repeated five times to compute error bars.
>
> The results (Fig. 10, p.29) show that even with a dataset one order of magnitude smaller ($M = 6000$), **FIRMBOUND demonstrates almost negligible differences in performance compared to the original dataset** in terms of mean hitting time, AAPR, and mean per-class error rate. Notably, the real-world HMDB51 dataset has a similar order of magnitude ($M = 5277$), further showcasing FIRMBOUND’s robustness in real-world scenarios. However, with the smallest dataset ($M = 600$), AAPR and mean per-class error rates increase significantly, indicating that the dataset size is insufficient to accurately estimate the sufficient statistics. Despite this, the mean hitting time remains close to the original, highlighting FIRMBOUND’s stability and its ability to make reliable best-effort decisions even under highly limited data conditions.

---

> ### Author Response · Authors · 2024-11-16
>
> # On the Suggested References
> Thank you for pointing out potentially related works. We appreciate the effort in identifying connections and have carefully considered each reference. Below, we provide a clarification on how these works relate to our paper:
>
> “Analyzing the discrepancy principle for kernelized spectral filter learning algorithms” (Celisse and Wahl, 2020): This paper focuses on early stopping during the training process of iterative algorithms, such as gradient descent. While early stopping and early classification share some conceptual similarities in decision-making under constraints, the goals and methods differ significantly. Our paper focuses on early classification of time series with a finite horizon, rather than halting training processes.
>
> “WALD-Kernel: A method for learning sequential detectors” (Teng and Ertin, 2016): This work focuses on binary classification in i.i.d. sequences and assumes infinite horizons, which limits its applicability to real-world, non-i.i.d. scenarios addressed in our paper. While it could potentially serve as an additional baseline, its performance would be upper-bounded by the FIRMBOUND algorithm, which employs consistent DRE methods capable of handling non-i.i.d. datasets. Nonetheless, given the shared interest in sequential decision-making, we have cited this paper in the updated manuscript to acknowledge its relevance to the broader context of our study.
>
> “Bayesian Optimization Meets Bayesian Optimal Stopping” (Dai et al., 2019): This paper focuses on hyperparameter optimization and Bayesian optimization, specifically improving efficiency in hyperparameter tuning. Although it involves Bayesian concepts, it does not address early classification of time series or finite-horizon decision-making, which are the central topics of our work.
>
> We hope this clarifies the distinction between these works and the problem addressed in our paper. Nonetheless, we appreciate these references, as they highlight diverse areas of research that could inspire future extensions or applications of our framework.
>
> # Remaining Practical Obstacles
> Our work focuses on removing practical obstacles to applying optimal early classification in real-world scenarios. However, as discussed in Appendix F, Section F.5, several limitations remain. One notable issue is the domain gap. The theoretical framework underpinning FIRMBOUND does not account for potential domain shifts between training and test datasets. To mitigate domain shifts, one approach is to use large-scale foundation models for feature extraction, which can improve robustness to changes in data distribution. For example, in the UCF101 dataset, we use DINOv2 (Darcet et al., 2024), a large-scale vision transformer trained on extensive image datasets, as the feature extractor for SPRT-TANDEM. Our experiments show that DINOv2 significantly reduces the domain gap compared to ResNet50.
>
> While addressing domain gaps remains an active area of research in machine learning, we defer further exploration to future work, as the proposed approach effectively demonstrates FIRMBOUND’s contributions within the scope of this study.
>
>
> # On the Loss of Consistency Using Gaussian Process Regression
> Thank you for highlighting this important point (addressing Weakness 3). We have added the following discussion to Appendix F.
>
> To clarify, it is possible to construct a consistent estimator using Gaussian Processes (GP) under certain conditions. The GP regressor can be a consistent estimator if, with an appropriate choice of the kernel, $\epsilon_m^{(t)}$ follows a Gaussian distribution for all $t \in [T]$ and $m \in [M]$, and $\\{ \tilde{G}\_t(\mathscr{S}\_{t, m}) \\}_{m=1}^{M}$ is a Gaussian process for all $t \in [T]$. However, these conditions are difficult to ensure under arbitrary circumstances. As a result, we introduced Convex Function Learning (CFL) as a more general and robust alternative.
>
> Nonetheless, our experiments show that the GP regressor performs competitively with CFL, which is the theoretically optimal approach. This makes GP a practical choice in scenarios with limited computational resources. It is worth noting that GP regression is particularly efficient during the *training* stage. During testing or deployment, both CFL and GP regression are sufficiently fast to support real-time decision-making. Therefore, GP regression is an effective option in environments with constrained training resources, such as edge computing settings.

---

> > ### Author Response · Authors · 2024-11-22
> >
> > Dear Reviewer `2NLt`,
> >
> > With the discussion period nearing its end, I wanted to kindly ask you to engage the discussion. We have carefully addressed all of your comments and concerns. We believe that our responses clarify the key points but if there are additional questions or points requiring clarification, we would be happy to address them promptly. We kindly ask that you consider our responses in your reevaluation of the manuscript, and we sincerely appreciate your time and effort in reviewing our work.
> >
> > Best regards,
> > Authors

---

> > > ### Comment · Reviewer_2NLt · 2024-11-25
> > >
> > > Dear Authors,
> > >
> > > Many thanks for taking the time to respond to my questions.

---

> > > > ### Author Response · Authors · 2024-11-30
> > > >
> > > > Thank you for your feedback. If you have any further questions, we’re happy to address them during the discussion period, which remains open until Dec. 2nd.

---

### Official Review · Reviewer_mx42 · 2024-11-06

**Soundness:** 3
**Presentation:** 3
**Contribution:** 3
**Rating:** 6
**Confidence:** 2

**Summary:**

This paper considers sequential probability ratio test (SPRT) under finite horizon. Bayes optimality characterizes backward induction equation, given in Theorem 2.1 in the paper. The contribution of the paper is to provide a statistical method that implements this formulation. This task involves estimating the conditional expectation function as well as density ratio estimation. The former exploits concavity in the backward induction equation and gives an option of Gaussian process (GP) regression for speed-up. Theorem 3.2 indicates that the proposed method is statistically consistent. An extensive set of experiments provides numerical evidence that the propose method works better than vanilla SPRT with static threshold.

**Strengths:**

- SPRT under finite horizon seems very relevant in practice.
- The proposed method is well grounded in Bayes optimality.
- The numerical results are extensive.
- The figures look impressive, although information contained in each of them is very dense.

**Weaknesses:**

- The paper is very lengthy if we include the appendices (60 pages in the current format). Purely based on the length, one might argue that it might be better to publish this paper in JMLR or a journal outlet.
- Theorem 3.2 informally states the consistency of the proposed method and Theorem J.1 in Appendix J gives a formal statement. However, assumptions are given in the forms of lemmas and thus they are not low-level or easy-to-understand conditions. In fact, the lemmas are theoretical results from Siahkamari et al. (2022, ICML) titled "Faster algorithms for learning convex functions". In view of that, it is not very innovative in terms of proving the consistency of the proposed method.

**Questions:**

- It would be helpful to clarify the main contribution of this paper. It seems to me that this paper is a convex combination of three existing results: (i) Bayes optimality via backward induction equation, (ii) convex function learning, and (iii) density ratio estimation. Would it be fair to say that the current paper is novel in the sense that it combines the three above but each of them is already developed by previous works?
- It is intriguing that Gaussian process (GP) regression gives a substantial speed-up. Would it be possible to establish consistency using GP?
- There is no theoretical result regarding the convergence rates or minimax lower bound. It might be good to indicate them as possible future research topics.

---

> ### Author Response · Authors · 2024-11-13
>
> Thank you for your thoughtful and positive feedback on our paper. We greatly appreciate your time and effort in reviewing our work. Please find our detailed responses to your comments below.
>
> # On the Combination of Existing Results
> We respectfully disagree with the suggestion that our paper combines existing results without introducing innovation. **The primary novelty of our work lies in reformulating the estimation of optimal decision boundaries as a noisy convex regression problem**, enabling the consistent estimation of both the conditional expectation in backward induction and the log-likelihood ratio. We provide a proof to this "double consistency" and the approach is empirically validated in extensive experiments, where FIRMBOUND achieves Pareto-optimal performance and reduces hitting-time variance across diverse datasets, from i.i.d. binary to non-i.i.d. multiclass data.
>
> **Consistent estimation itself marks a significant advancement in early classification of time series (ECTS)**. Most prior ECTS methods rely on empirical heuristics, such as combining misclassification penalties with time costs (e.g., LSTMms, TCN-Transformer, and [1]). Our experiments demonstrate that these heuristic-based algorithms are outperformed by FIRMBOUND, underscoring the practical impact of our theoretical approach.
>
> While we build on prior work (as is often the case in research), we believe this integration addresses a novel and impactful problem in ECTS, demonstrating both theoretical and empirical advancements. We kindly ask the reviewer to consider the overall contribution, the novelty of application, and effectiveness of our method. As per the ICLR 2025 Reviewer Guidelines, we aim to provide “new knowledge and sufficient value to the community” (https://iclr.cc/Conferences/2025/ReviewerGuide). Although from a different conference, NeurIPS 2024 Guidelines similarly recognize that a “novel combination of well-known techniques can be valuable” (https://neurips.cc/Conferences/2024/ReviewerGuidelines). They emphasize that a well-executed combination of existing methods addressing new challenges can offer substantial contributions to the field.
>
> Additionally, combining established algorithms is often non-trivial due to differences in assumptions, mathematical structures, and implementation requirements. For example, while the SPRT is known to be Bayes optimal in minimizing the expected sample size (hitting time) at a given error rate [2], combining it with components like video cameras and feature extractors does not automatically guarantee SPRT’s optimality. In real-world applications, where the sufficient statistic (log-likelihood ratio or LLR) is unknown, SPRT’s theoretical properties cannot be directly leveraged.
>
> Even when components are theoretically combinable, achieving optimal performance and conducting rigorous analysis can be challenging. For instance, combining SPRT with density ratio estimation (DRE) [3] to estimate the LLR is non-trivial because sequential data in practice is often non-i.i.d., unlike many applications of SPRT that assume i.i.d. data [4]. It has taken over 70 years since SPRT's development to effectively realize its optimality in practical settings [5], and even so, its optimality remains unproven for a crucial condition—finite horizons. FIRMBOUND aims to overcome this remaining challenge.
>
> Our particular combination of methods was carefully designed to achieve double consistency, addressing both theoretical and practical challenges in ECTS. Determining if these methods could work together effectively, produce a reliable solution, and outperform existing approaches was by no means straightforward. As demonstrated in Appendix D, “Computational Complexity of FIRMBOUND and Sampling Method,” conventional methods are prohibitively slow in high-dimensional contexts. CFL enables practical, direct estimation of the conditional expectation in the backward induction, with sufficient statistics (e.g., a likelihood ratio in complex non-i.i.d. streams) supported by DRE. Together, these elements make FIRMBOUND both theoretically sound and computationally feasible.
>
> Finally, while additional innovations could be incorporated (e.g., differentiable SPRT in the training loop to further optimize hitting time), we aim to provide the simplest, most effective solution for achieving a doubly consistent ECTS framework, consistent with best practices (https://www.cs.ucr.edu/~eamonn/Keogh_SIGKDD09_tutorial.pdf).
>
> [1] Mori+, Early classification of time series from a cost minimization point of view. NIPS Time Series Workshop, 2015.
> [2] Wald. Sequential tests of statistical hypotheses. Ann. Math. Statist., 1945.
> [3] Sugiyama+, Density Ratio Estimation in Machine Learning. Cambridge University Press, 2012.
> [4] Teng and Ertin. Wald-kernel: A method for learning sequential detectors. IEEE SSP, 2016.
> [5] Ebihara+, Sequential density ratio estimation for simultaneous optimization of speed and accuracy. ICLR, 2021.

---

> ### Author Response · Authors · 2024-11-14
>
> # Would it be Possible to Establish Consistency Using GP?
> Yes, it is possible to construct a consistent estimator using Gaussian Processes (GP) under certain conditions. The GP regressor can be a consistent estimator if, with an appropriate choice of the kernel, $\epsilon_m^{(t)}$ follows a Gaussian distribution for all $t \in [T]$ and $m \in [M]$, and $\\{ \tilde{G}\_t(\mathscr{S}\_{t, m}) \\}\_{m=1}^{M}$ is a Gaussian process for all $t \in [T]$. However, these conditions are difficult to guarantee under arbitrary circumstances. Thus, we introduced CFL as a more general and robust solution. We updated Appendix F.6 to include the above discussion.
>
> # Future Theoretical Directions
> Thank you for your insightful suggestion about the convergence rate and minimax lower bound.
> The convergence rate is presumably given by the sum of LSEL's and CFL's.
> The latter is given in the paper, but the former requires an additional extensive analysis because Lemma J.4 (consistency of LSEL), on which our consistency proof relies, is an asymptotics of the probability that the estimated parameters deviates from the optimal parameter set.
> For the same reason, our proof cannot adapt to minimax bounds straightforwardly.
> These discussions would warrant a separate, focused study.
> Please see also Appendix F, which is dedicated to future theoretical directions, along with additional open questions.
>
>
> # On the Length of the Paper
> We do not believe the paper length should be considered a weakness. **Our priority is comprehensiveness, particularly to support reproducibility in machine learning**. Much of the Appendix is devoted to hyperparameter settings and additional experiments, reflecting our commitment to transparent research practices.
>
> FIRMBOUND is a multidisciplinary project, involving concepts from statistics, machine learning, reinforcement learning, ECTS, CFL, GP regression, and neuroscience. Consequently, we provide extensive background information and detailed related work. Furthermore, we’ve encountered reviewer feedback of the past papers that emphasized the need for rigorous mathematical foundations, which prompted us to make our theoretical explanations self-contained in the Appendix.

---

> > ### Author Response · Authors · 2024-11-22
> >
> > Dear Reviewer `mx42`,
> >
> > With the discussion period nearing its end, I wanted to kindly ask you to engage the discussion. We have carefully addressed all of your comments and concerns. We believe that our responses clarify the key points but if there are additional questions or points requiring clarification, we would be happy to address them promptly. We kindly ask that you consider our responses in your reevaluation of the manuscript, and we sincerely appreciate your time and effort in reviewing our work.
> >
> > Best regards,
> > Authors

---

> > > ### Comment · Reviewer_mx42 · 2024-11-26
> > >
> > > Thank you very much for your responses. I was wondering if you addressed my earlier comment regarding one of the weaknesses, namely,
> > >
> > > - Theorem 3.2 informally states the consistency of the proposed method and Theorem J.1 in Appendix J gives a formal statement. However, assumptions are given in the forms of lemmas and thus they are not low-level or easy-to-understand conditions. In fact, the lemmas are theoretical results from Siahkamari et al. (2022, ICML) titled "Faster algorithms for learning convex functions". In view of that, it is not very innovative in terms of proving the consistency of the proposed method.
> > >
> > > Perhaps I missed this point, but would you kindly provide a response to this question?

---

> > > > ### Author Response · Authors · 2024-11-28
> > > >
> > > > Thank you for your follow-up question. We appreciate the opportunity to provide further clarification.
> > > >
> > > > To ensure complete clarity, we would like to confirm that Theorem J.1, which establishes the consistency of FIRMBOUND, is not solely based on the work of Siahkamari et al. (2022). Theorem J.1 integrates two distinct components:
> > > >
> > > > - Consistent estimation of the backward induction equation, derived from Lemmas J.1–J.3 of Siahkamari et al.
> > > > - Consistent estimation of the sufficient statistic (log-density ratio), which relies on results from Miyagawa and Ebihara (2021, Lemma J.4).
> > > >
> > > > Together, these components enable FIRMBOUND to achieve “double consistency,” a key requirement for the early classification of time series (ECTS). It is important to note that log-density ratio estimation is conceptually independent of convex optimization. Prior to FIRMBOUND, there was no explicit connection between convex function learning and density ratio estimation within the ECTS framework. The integration of these two distinct methodologies is central to FIRMBOUND’s novelty and contribution.
> > > >
> > > >
> > > >
> > > > > Theorem 3.2 informally states the consistency of the proposed method and Theorem J.1 in Appendix J gives a formal statement. However, assumptions are given in the forms of lemmas and thus they are not low-level or easy-to-understand conditions.
> > > >
> > > > To clarify, we summarize and rephrase our assumptions of Theorem 3.2 (= Theorem J.1), which are provided in Appendix J.
> > > >
> > > > ### Setup
> > > >
> > > > In Appendix G, we review the 2-block ADMM algorithm (Siahkamari et al., 2022), which is used for solving the convex regression problem (Eq. 12).
> > > > We solve the following noisy convex regression problem with regularization:
> > > > $
> > > > \hat{f} \triangleq \underset{f}{\arg \min } \frac{1}{n} \sum\_{i=1}^n\left(y\_i-f\left(\boldsymbol{x}\_i\right)\right)^2+\bar{\lambda}\|f\|,
> > > > $
> > > > where $\boldsymbol{x}\_i \in \mathbb{R}^d$, and $\bar{\lambda}$ is a hyperparameter affecting convergence. Note that regression labels $y_i$ are noisy; i.e., they have bounded, centered, independent, random discrepancies from the true label.
> > > >
> > > > - Assumption: the noise $\epsilon_i := y_i - f(x_i)$ is centered, independent, random noise. $|\epsilon_i|$ is bounded by a real number $\xi > 0$.
> > > >
> > > > The ground-truth function $f: \mathbb{R}^d \rightarrow \mathbb{R}$ is assumed to satisfy the following conditions:
> > > >
> > > > - Assumption: $f$ is convex and Lipshitz. $|f|$ is bounded by $\xi$.
> > > >
> > > > The 2-block ADMM solves this problem by using piecewise linear functions:
> > > > $
> > > > \min\_{\hat{y}\_i, a\_i} \frac{1}{n} \sum\_{i=1}^n\left(\hat{y}\_i-y\_i\right)^2+\bar{\lambda }\sum\_{l=1}^d \max\_{i=1}^n\left|a_{i, l}\right|
> > > > $
> > > >
> > > > $
> > > > \text { s.t. } \hat{y}_i-\hat{y}_j-\left\langle a_i, \boldsymbol{x}_i-\boldsymbol{x}_j\right\rangle \leq 0 \quad i, j \in[n] \times[n] .
> > > > $
> > > >
> > > > Then, we estimate $f(\boldsymbol{x})$ via
> > > >
> > > > $
> > > > \hat{f}(\boldsymbol{x}) \triangleq \max\_i\left\langle a\_i, \boldsymbol{x}-\boldsymbol{x}\_i\right\rangle+\hat{y}\_i .
> > > > $
> > > >
> > > > The 2-block ADMM is summarized in Algorithms 1 & 2 and Updates 1-4 in Appendix G (with no additional assumptions).
> > > >
> > > > Finally, please bear the following correspondence in mind:
> > > >
> > > > - Note: $y\_i = f(x\_i) + \epsilon\_i$ corresponds to $\mathscr{G}\_m^{(t+1)}+c=\tilde{G}\_t\left(\mathscr{S}\_{t, m}\right)+\epsilon\_m^{(t)}$ (Eq. (11) in the main text). Thus, $n$ and $d$ are denoted by $M$ (the number of training examples) and $K$ (the number of classes) in the main text, respectively.
> > > >
> > > > ### Assumptions in Lemma J.1
> > > >
> > > > Consider the following assumptions:
> > > >
> > > > - Assumption: The hyperparameter $\bar{\lambda}$ is appropriately chosen, which requires knowledge of the bound on $f$ (note that no explicit form is given in Siahkamari et al. (2022) and references therein).
> > > > - Assumption: It holds that $n \geq d$.
> > > >   - That is, the number of training examples is greater than or equal to the number of classes.
> > > >
> > > > Then, Lemma J.1 indicates that CFL is a consistent estimator of $f$; i.e., $\hat{f}$ approaches $f$ as $M \rightarrow \infty$.

---

> > > > > ### Author Response · Authors · 2024-11-28
> > > > >
> > > > > ### Assumptions in Lemma J.2
> > > > >
> > > > > Consider the following assumptions:
> > > > >
> > > > > - Assumption: $\max\_{i, l}\left|x\_{i, l}\right| \leq 1$, and $\mathrm{Var}\left(\left\\{y\_i\right\\}\_{i=1}^n\right) \leq 1$.
> > > > >   - That is, the max of the sufficient statistics and the variance of the observed continuation risks are bounded.
> > > > > - Assumption: $\rho=\frac{\sqrt{d} \bar{\lambda}^2}{n}, \bar{\lambda} \geq \frac{3}{\sqrt{2 n d}},$ $\mathscr{T} \geq n \sqrt{d}$, where $\rho$ is a hyperparameter in Algorithm 1 and 2, and $\mathscr{T}$ is the number of iterartions of the 2-block ADMM.
> > > > >   - That is, the hyperparameters are appropriately set, and the number of iterations is sufficiently large.
> > > > >
> > > > > As noted above, the inputs $\boldsymbol{x}$ and outputs $y$ in the lemma correspond to $(\pi_1(X^{(1,t)}), \ldots, \pi_K(X^{(1,t)}))$, and to $\tilde{G}\_t(\mathscr{S}\_t(X^{(1,t)}))$, respectively.
> > > > > $\pi\_k(X^{(1,t)})$ is the posterior of class $k$ and obviously bounded by one. Thus, the assumption $\max\_{i,l} |x\_{i,l}| \leq 1$ is satisfied.
> > > > > Also, the assumption $\mathrm{Var}(\\{ y\_i \\}\_{i=1}^n) \leq 1$ is satisfied because we only consider integrable functions, and the continuation risk $\tilde{G}\_t$ is bounded.
> > > > >
> > > > > Under these assumptions, Lemma J.2 ensures the convergence of the CFL algorithm.
> > > > >
> > > > > ### Assumptions in Lemma J.3
> > > > > Lemma J.3 does not require additional assumptions.
> > > > >
> > > > > ### Assumptions in Lemma J.4
> > > > >
> > > > > Lemma J.4 indicates that the minimizer of LSEL is a consistent estimator of log-likelihood ratios, which requires:
> > > > >
> > > > > - Assumption: The log-likelihood ratios $\log p(X^{(1,t)} | y=k)/p(X^{(1,t)} | y=l)$ with $k, l \in [K]$ exist and are finite.
> > > > > - Assumption: The standard assumptions of the uniform law of large numbers, namely, compactness, continuity, measurability, and dominance, given LSEL (${L}\_\mathrm{LSEL}$) and its empirical approximation ($\hat{L}\_\mathrm{LSEL}$).
> > > > > - Assumption: A technical requirement on the density $p$ and the estimator $\hat{\lambda}\_{\boldsymbol{w}}$, often assumed in the literature (Gutmann & Hyvärinen, 2012). For all $\boldsymbol{w}^*$ in the optimal parameter set, there exist $t \in[T], k \in[K]$, and $l \in[K]$, such that the following $d \times d$ matrix is full-rank: $\int d X^{(1, t)} p\left(X^{(1, t)} \mid k\right) \nabla\_{\boldsymbol{w}^*} \hat{\lambda}\_{k l}\left(X^{(1, t)} ; \boldsymbol{w}^*\right) \nabla\_{\boldsymbol{w}^*} \hat{\lambda}\_{k l}\left(X^{(1, t)} ; \boldsymbol{w}^*\right)^{\top}$.
> > > > >
> > > > > To ensure our estimator $\hat{\lambda}\_{\boldsymbol{w}}$ (a neural network) can approximate the target log-likelihood ratios, we additionally assume that the neural network represented by $\boldsymbol{w}$ is sufficiently large (the universal approximation theorem of neural networks).
> > > > >
> > > > > - Assumption: The parameter set $W$ of the neural network estimator is so large that it can represent the target log-likelihood ratios.
> > > > >
> > > > > This concludes the full list of assumptions for the consistency proof of FIRMBOUND.

---

> > > > > > ### Author Response · Authors · 2024-11-28
> > > > > >
> > > > > > > In fact, the lemmas are theoretical results from Siahkamari et al. (2022, ICML) titled "Faster algorithms for learning convex functions". In view of that, it is not very innovative in terms of proving the consistency of the proposed method.
> > > > > >
> > > > > > As noted in our rebuttal section, “On the Combination of Existing Results,” consistent estimation represents a significant advancement in the field of ECTS. Most prior approaches rely on heuristic methods rather than theoretically grounded techniques.
> > > > > >
> > > > > > While the combination of convex function learning (CFL) and log-likelihood ratio estimation (LSEL) may appear straightforward in hindsight, arriving at this solution *was* far from trivial. Through extensive discussions and preliminary experiments, we determined that implementing the backward induction equation in practice—arguably the most challenging aspect of our problem—was unexpectedly feasible using this modular structure. By combining CFL and LSEL, we not only simplified the implementation but also enabled a concise proof of consistency, adapting insights from Siahkamari et al. (2022) and Miyagawa and Ebihara (2021) to the unique context of ECTS. This integration is the core innovation of FIRMBOUND.
> > > > > >
> > > > > > Furthermore, our experiments demonstrate that FIRMBOUND consistently outperforms heuristic-based methods across diverse real-world datasets, including i.i.d., non-i.i.d., binary, and multiclass settings, all under finite horizons. This underscores the practical relevance of our framework, as you kindly noted in your Strengths section (thank you).
> > > > > >
> > > > > > As the discussion period has been extended until Dec. 2nd, please do not hesitate to reach out if further clarification is needed. We sincerely appreciate your thoughtful feedback and the time you have dedicated to reviewing our work.

---

> > > > > > > ### Comment · Reviewer_mx42 · 2024-12-03
> > > > > > >
> > > > > > > Thank you very much for your thorough explanations. After careful consideration, I will be maintaining my original grading.

---

> > > > > > > > ### Author Response · Authors · 2024-12-03
> > > > > > > >
> > > > > > > > Thank you for acknowledging our detailed responses. We appreciate your time and effort in reviewing our work. If any concerns remain, we would be happy to address them during the remainder of the discussion period.

---

### Author Response · Authors · 2024-11-13

We thank the reviewers for their thorough reading of our paper and their recognition of its contributions. We are pleased to see that the reviewers appreciated the theoretical foundations of our work, particularly the statistically consistent estimation methods for both the backward induction and log density ratio [`mx42`, `2NLt`, `81so`]. We are also encouraged by the reviewers' acknowledgment of FIRMBOUND as a Bayes-optimal approach for early classification of time series (ECTS) [`81so`], with practical relevance [`mx42`, `81so`].

The reviewers highlighted the technical difficulty of optimizing ECTS [`HH7v`] and affirmed that our algorithm is both sound and implementable [`HH7v`]. We appreciate the acknowledgment of our extensive experiments [`mx42`, `HH7v`], which demonstrate robust performance on i.i.d. and non-i.i.d. datasets [`2NLt`], including both synthetic and real-world cases [`2NLt`, `81so`, `HH7v`]. Additionally, we are glad the figures were found to be visually effective [`mx42`].

---

> ### Author Response · Authors · 2024-11-17
>
> **We have provided detailed, point-by-point replies below and updated the PDF as needed.** We hope that these responses address the raised concerns and kindly invite the reviewers to consider them in their reevaluation of the manuscript.

---

### Meta-Review · Area_Chair_DRSt · 2024-12-17

**Metareview:**

The paper develops a framework for early classification of time series (in the finite horizon setting) based on sequential probability ratio test (SPRT). It combines ideas from Bayes optimality via backward induction, density ratio estimation, and convex function learning to deal with the computational challenges of determining the optimal stopping rule. GP regression is proposed as a faster alternative but is less statistically consistent.

Overall, the reviewers believe the paper addresses an important problem and works well in practice. There were several issues raised by the reviewers concerning the (1) length of the paper and its suitability for conference presentation; (2) the novelty of the paper wrt previous theoretical results; (3) the potential need for more comparisons with previous works and additional experiments dealing with more stringent data constraints.  Despite the lack of a more thorough engagement from the reviewers and the relatively low confidence across all of them, I believe the authors have addressed all these concerns satisfactorily. Therefore, I recommend acceptance.

**Additional Comments On Reviewer Discussion:**

Despite the lack of a more thorough engagement from the reviewers and the relatively low confidence across all of them, I believe the authors have addressed all these concerns satisfactorily. Therefore, I recommend acceptance.

---

### Decision · Program_Chairs · 2025-01-22

Accept (Poster)